# THE PHASE TRANSITION PHENOMENON OF SHUFFLED REGRESSION

## ABSTRACT

We study the phase transition phenomenon inherent in the shuffled (permuted) regression problem, which has found numerous applications in databases, privacy, data analysis, etc. For the permuted regression task: $\mathbf{Y} = \mathbf{\Pi}^{\natural}\mathbf{X}\mathbf{B}^{\natural}$, the goal is to recover the permutation matrix $\mathbf{\Pi}^{\natural}$ as well as the coefficient matrix $\mathbf{B}^{\natural}$. It has been empirically observed in prior studies that when recovering $\mathbf{\Pi}^{\natural}$, there exists a phase transition phenomenon: the error rate drops to zero rapidly once the parameters reach certain thresholds. In this study, we aim to precisely identify the locations of the phase transition points by leveraging techniques from *message passing* (MP).

In our analysis, we first transform the permutation recovery problem into a probabilistic graphical model. Then, we leverage the analytical tools rooted in the message passing (MP) algorithm and derive an equation to track the convergence of the MP algorithm. By linking this equation to the branching random walk process, we are able to characterize the impact of the *signal-to-noise-ratio* (snr) on the permutation recovery. Depending on whether the signal is given or not, we separately investigate the oracle case and the non-oracle case. The bottleneck in identifying the phase transition regimes lies in deriving closed-form formulas for the corresponding critical points, but only in rare scenarios can one obtain such precise expressions. To tackle this challenge, we propose the Gaussian approximation method, which allows us to obtain the closed-form formulas in almost all scenarios. In the oracle case, our method can fairly accurately predict the phase transition snr. In the non-oracle case, our proposed algorithm can predict the maximum allowed number of permuted rows and uncover its dependency on the sample number.

Numerical experiments reveal that the observed phase transition points are well aligned with our theoretical predictions. Our study will motivate exploiting MP algorithms (and related techniques) as an effective tool for permuted regression problems, which have found applications in machine learning, privacy, and databases.

## 1 INTRODUCTION

In this paper, we consider the following permuted (shuffled) linear regression problem:

$$\mathbf{Y} = \mathbf{\Pi}^{\natural}\mathbf{X}\mathbf{B}^{\natural} + \sigma\mathbf{W}, \tag{1}$$

where $\mathbf{Y} \in \mathbb{R}^{n \times m}$ denotes the matrix of observations, $\mathbf{\Pi}^{\natural} \in \{0,1\}^{n \times n}$ is the permutation matrix, $\mathbf{X} \in \mathbb{R}^{n \times p}$ is the design matrix, $\mathbf{B}^{\natural} \in \mathbb{R}^{p \times m}$ is the matrix of signals (regressors), $\mathbf{W} \in \mathbb{R}^{n \times m}$ denotes the additive noise matrix (with unit variance), and $\sigma^2$ is the noise variance. The task is to recover both the signal matrix $\mathbf{B}^{\natural}$ and the permutation matrix $\mathbf{\Pi}^{\natural}$. The research on this challenging permuted regression problem dates back at least to 1970s under the name "broken sample problem" (DeGroot et al., 1971; Goel, 1975; DeGroot & Goel, 1976; 1980; Bai & Hsing, 2005). Recent years have witnessed a revival of this problem due to its broad spectrum of applications in (e.g.,) privacy protection, data integration, etc. (Unnikrishnan et al., 2015; Pananjady et al., 2018; Slawski & Ben-David, 2019; Pananjady et al., 2017; Slawski et al., 2020; Zhang & Li, 2020).

Specifically, this paper will focus on studying the "phase transition" phenomenon in recovering the whole permutation matrix $\mathbf{\Pi}^{\natural}$: the error rate for the permutation recovery sharply drops to zero

once the parameters reach certain thresholds. In particular, we leverage techniques in the *message passing* (MP) algorithm literature to identify the precise positions of the phase transition thresholds. The bottleneck in identifying the phase transition regimes lies in deriving closed-form formulas for the corresponding critical points. This is a highly challenging task because only in rare scenarios can one obtain such precise expressions. To tackle the difficulty, we propose the Gaussian approximation method which allows us to obtain the closed-form formula in almost all scenarios. We should mention that, in previous studies (Slawski et al., 2020; Pananjady et al., 2017; Zhang et al., 2022; Zhang & Li, 2020), this phase transition phenomenon was empirically observed.

**Related work.** The problem we study simultaneously touches two distinct areas of research: (A) permutation recovery, and (B) *message passing* (MP). In the literature of permuted linear regression, essentially all existing works used the same setting (1). Pananjady et al. (2018); Slawski & Ben-David (2019) consider the single observation model (i.e., $m = 1$) and prove that the *signal-to-noise-ratio* (snr) for the correct permutation recovery is $\mathbb{O}_P(n^c)$, where $c > 0$ is some positive constant. Slawski et al. (2020); Zhang & Li (2020); Zhang et al. (2022) investigate the multiple observations model (i.e., $m > 1$) and suggest that the snr requirement can be significantly decreased, from $\mathbb{O}_P(n^c)$ to $\mathbb{O}_P(n^{c/m})$. In particular, Zhang & Li (2020) develop an estimator which we will leverage and analyze for studying the phase transition phenomenon. Compared with the above work, our analysis can identify the precise locations of the phase transition thresholds. In this February, there comes a paper (Lufkin et al., 2024) considering the same problem as ours but in a much simpler setting (single measurement with $m = 1$). Compared with their work, our framework can easily reproduce their predicted phase transition points, answer the questions they treat as open, and predict the phase transition points in a unified framework. Another line of related research comes from the field of statistical physics. For example, using the replica method, Mézard & Parisi (1985; 1986) study the *linear assignment problem* (LAP), i.e., $\min_{\mathbf{\Pi}} \sum_{i,j} \mathbf{\Pi}_{ij} \mathbf{E}_{ij}$ where $\mathbf{\Pi}$ denotes a permutation matrix and $\mathbf{E}_{ij}$ is i.i.d random variable uniformly distributed in $[0, 1]$. Martin et al. (2005) then generalize LAP to multi-index matching and presented an investigation based on MP algorithm. Recently, Caracciolo et al. (2017); Malatesta et al. (2019) extend the distribution of $\mathbf{E}_{ij}$ to a broader class. However, all the above works exhibit no phase transition. Chertkov et al. (2010) extend it to the particle tracking problem and observe a phase transition phenomenon. Later, Semerjian et al. (2020) modify it to fit the graph matching problem, which paves way for our work in studying the permuted linear regression problem.

**Our contributions.** We propose the first framework to identify the precise locations of phase transition thresholds associated with permuted linear regression. In the oracle case where $\mathbf{B}^\natural$ is known, our scheme is able to determine the phase transition snr. In the non-oracle case where $\mathbf{B}^\natural$ is not given, our method will predict the maximum allowed number of permuted rows and uncover its dependence on the ratio $p/n$. In our analysis, we identify the precise positions of the phase transition points in the large-system limit, e.g., $n$, $m$, $p$ all approach to infinity with $m/n \to \tau_m$, $p/n \to \tau_p$. Interestingly, numerical results well match predictions even when $n, m, p$ are in the hundreds.

Here, we would also like to briefly mention the technical challenges. Compared with the previous works (Mezard & Montanari, 2009; Talagrand, 2010; Linusson & Wästlund, 2004; Mézard & Parisi, 1987; 1986; Parisi & Ratiéville, 2002; Semerjian et al., 2020), where the edge weights are relatively simple, our edge weights usually involve high-order interactions across Gaussian random variables and are densely correlated. To tackle this issue, our proposed approximation method to compute the phase transition thresholds consists of three parts: 1) performing Gaussian approximation; 2) modifying the leave-one-out technique; and 3) performing size correction. A detailed explanation can be found in Section 4. Hopefully, our approximation method will serve independent technical interests for researchers in the machine learning community.

**Notations.** $a \xrightarrow{\text{a.s.}} b$ denotes $a$ converges almost surely to $b$. $\mathcal{P}_n$ denotes the set of all possible permutation matrices: $\mathcal{P}_n \triangleq \{\mathbf{\Pi} \in \{0,1\}^{n \times n}, \sum_i \mathbf{\Pi}_{ij} = 1, \sum_j \mathbf{\Pi}_{ij} = 1\}$. The *signal-to-noise-ratio* is snr $= \frac{\|\mathbf{B}^\natural\|_{\mathrm{F}}^2}{m \cdot \sigma^2}$, where $\|\cdot\|_{\mathrm{F}}$ is the Frobenius norm and $\sigma^2$ is the variance of the sensing noise.

## 2 Permutation Recovery Using the Message Passing Algorithm

Inspired by Mezard & Montanari (2009); Chertkov et al. (2010); Semerjian et al. (2020), we leverage tools from the statistical physics to identify the locations of the phase transition threshold. We start

this section with a brief review of the *linear assignment problem* (LAP), which reads as

$$\widehat{\mathbf{\Pi}} = \text{argmin}_{\mathbf{\Pi} \in \mathcal{P}_n} \langle \mathbf{\Pi}, \mathbf{E} \rangle, \tag{2}$$

where $\mathbf{E} \in \mathbb{R}^{n \times n}$ is a fixed matrix and $\mathcal{P}_n$ denotes the set of all possible permutation matrices.

In our work, we first establish the link between the LAP and the permuted linear recovery, to be more specific, formulating the permutation recovery of (1) in the form of (2). Next, we predict the phase transition points by studying the matrix $\mathbf{E}$, which is our major contribution.

We follow the approach in Mezard & Montanari (2009); Semerjian et al. (2020) and introduce a probability measure over the permutation matrix $\mathbf{\Pi}$, which is written as

$$\mu(\mathbf{\Pi}) = (1/z) \prod_i \mathbb{1}\left(1 - \sum_j \mathbf{\Pi}_{ij}\right) \prod_j \mathbb{1}\left(1 - \sum_i \mathbf{\Pi}_{ij}\right) \times \exp\left(-\beta \sum_{i,j} \mathbf{\Pi}_{ij} \mathbf{E}_{ij}\right), \tag{3}$$

where $\mathbb{1}(\cdot)$ is the indicator function, $Z$ is the normalization constant of the probability measure $\mu(\mathbf{\Pi})$, and $\beta > 0$ is an auxiliary parameter. It is easy to verify the following two properties, e.g., 1) ML estimator in (2) can be rewritten as $\widehat{\mathbf{\Pi}} = \text{argmax}_{\mathbf{\Pi}} \mu(\mathbf{\Pi})$[1]; and 2) the probability measure $\mu(\mathbf{\Pi})$ concentrates on $\widehat{\mathbf{\Pi}}$ when letting $\beta \to \infty$.

Then, we study the impact of $\{\mathbf{E}_{ij}\}$ on the reconstructed permutation $\widehat{\mathbf{\Pi}}$ with the *message passing* (MP) algorithm. First, we associate a probabilistic graphical model with the probability measure defined in (3). Then, we rewrite the solution in (2) in the language of the MP algorithm. Finally, we derive an equation (4) to track the convergence of the MP algorithm. By exploiting relation of (4) to the *branching random walk* (BRW) process, we can identify the phase transition points corresponding to the LAP in (2). Since this procedure is fairly standard and space is limited, we defer the detailed derivation to the Appendix.

## 2.1 IDENTIFICATION OF THE PHASE TRANSITION THRESHOLD

Following the approach of Semerjian et al. (2020), we split the pairs $(i, j)$ into two groups, depending on whether $\mathbf{\Pi}_{ij}^\natural$ equals zero. Within each group, we model the edge weights as independent and identically distributed random variables. For the set $\{(i,j) \mid \mathbf{\Pi}_{ij}^\natural = 1\}$, we denote the random variable corresponding to the edge weight $\mathbf{E}_{ij}$ by $\Omega$, and for all other edge weights we use $\widehat{\Omega}$. [2]

After linking the solution $\widehat{\mathbf{\Pi}} = \text{argmax}_{\mathbf{\Pi}} \mu(\mathbf{\Pi})$ with the message passing algorithm, we adopt an approach in the spirit of *density evolution* and *state evolution*, classical techniques used to analyze the convergence of message passing and approximate message passing algorithms, respectively (Chung, 2000; Richardson & Urbanke, 2001; 2008; Donoho et al., 2009; Maleki, 2010; Bayati & Montanari, 2011; Rangan, 2011). This allows us to study the phase transition point, i.e., the condition under which $\widehat{\mathbf{\Pi}} = \mathbf{\Pi}^\natural$. Consider the following iterative updates of the random variables $H$ and $\widehat{H}$

$$\widehat{H}^{(t+1)} = \min\left(\Omega - H^{(t)}, H^{'(t)}\right), \quad H^{(t+1)} = \min_{1 \le i \le n-1} \widehat{\Omega}_i - \widehat{H}_i^{(t)}, \tag{4}$$

where $(\cdot)^{(t)}$ denotes the update in the $t$-th iteration, $H^{'}$ is an independent copy of $H$, $\{H_i^{(t)}\}_{1 \le i \le n-1}$ and $\{\widehat{\Omega}_i\}_{1 \le i \le n-1}$ denote the i.i.d. copies of random variables $H_{(\cdot)}^{(t)}$ and $\widehat{\Omega}_{(\cdot)}$.

**Proposition 1 (Informal).** *The phase transition point, under which $\mathbb{P}(\widehat{\mathbf{\Pi}} \ne \mathbf{\Pi}^\natural)$, corresponds to the point in which the distributions of $H$ and $\widehat{H}$ begin to diverge to infinity.*

It's noteworthy that the incorrect recovery, i.e, $\mathbb{P}(\widehat{\mathbf{\Pi}} \ne \mathbf{\Pi}^\natural)$, does not mean the reconstructed correspondences are simultaneously incorrect. Numerical experiments also confirm this claim.

---

[1] Notice that the requirement $\mathbf{\Pi} \in \mathcal{P}_n$ is incorporated in $\mu(\mathbf{\Pi})$ implicitly and thus we do not need an explicit constraint.

[2] We conjecture that the distribution difference in the edges' weights is a necessary component in capturing the phase transition. On one hand, according to Mézard & Parisi (1986; 1987); Parisi & Ratiéville (2002); Linusson & Wästlund (2004); Mezard & Montanari (2009); Talagrand (2010), there is no phase transition phenomenon in LAP if the edges' weights, i.e., $\mathbf{E}_{ij}$, are assumed to be i.i.d uniformly distributed in $[0, 1]$. On the other hand, Semerjian et al. (2020) show a phase transition phenomenon when assuming the weights $\mathbf{E}_{ij}$ follow different distributions among the edges associated with the ground-truth correspondence $\mathbf{\Pi}_{ij}^\natural = 1$ and the rest edges.

**Relation to *branching random walk* (BRW) process.** Conditional on the event that the permutation can be perfectly reconstructed, we have $H + \widehat{H}' > \Omega$ (c.f. (19)) and can simplify (4) as

$$H^{(t+1)} = \min_{1 \leq i \leq n-1} H_i^{(t)} + \Xi_i, \tag{5}$$

where $\Xi$ is defined as the difference between $\widehat{\Omega}$ and $\Omega$, which is written as $\Xi \triangleq \widehat{\Omega} - \Omega$, and $\{H_i^{(t)}\}_{1 \leq i \leq n-1}$ and $\{\Xi_i\}_{1 \leq i \leq n-1}$ denote the i.i.d. copies of random variables $H_{(\cdot)}^{(t)}$ and $\Xi_{(\cdot)}$.

Adopting the same viewpoint of Semerjian et al. (2020), we treat (5) as a *branching random walk* (BRW) process, which enjoys the following property.

**Theorem 1** (Hammersley (1974); Kingman (1975); Semerjian et al. (2020)). *Consider the* recursive *distributional equation* $K^{(t+1)} = \min_{1 \leq i \leq n} K_i^{(t)} + \Xi_i$, *where* $K_i^{(t)}$ *and* $\Xi_i$ *are i.i.d copies of random variables* $K_{(\cdot)}^{(t)}$ *and* $\Xi_{(\cdot)}$, *we have* $\frac{K^{(t+1)}}{t} \xrightarrow{\text{a.s.}} -\inf_{\theta > 0} \frac{1}{\theta} \log \left[ \sum_{i=1}^n \mathbb{E} e^{-\theta \Xi_i} \right]$, *conditional on the event that* $\lim_{t \to \infty} K^{(t)} \neq \infty$.

With Theorem 1, we can compute phase transition point for the correct (full) permutation recovery, i.e., $H + H' > \Omega$, by letting $\inf_{\theta > 0} \frac{1}{\theta} \log \left[ \sum_{i=1}^n \mathbb{E} e^{-\theta \Xi_i} \right] = 0$, since otherwise the condition in (19) will be violated (see a detailed explanation in Appendix). In practice, directly computing the infimum of $\inf_{\theta > 0} \frac{1}{\theta} \log \left[ \sum_{i=1}^n \mathbb{E} e^{-\theta \Xi_i} \right]$ is only possible for limited scenarios. The next section proposes an approximate computation method for the phase transition points, which is capable of covering a broader class of scenarios.

## 3 ANALYSIS OF THE PHASE TRANSITION POINTS

Recall that, in this paper, we consider the following linear regression problem with permuted labels

$$\mathbf{Y} = \mathbf{\Pi}^\natural \mathbf{X} \mathbf{B}^\natural + \sigma \mathbf{W},$$

where $\mathbf{Y} \in \mathbb{R}^{n \times m}$ represents the matrix of observations, $\mathbf{\Pi}^\natural \in \mathcal{P}_n$ denotes the permutation matrix to be reconstructed, $\mathbf{X} \in \mathbb{R}^{n \times p}$ is the sensing matrix with each entry $\mathbf{X}_{ij}$ following the i.i.d standard normal distribution, $\mathbf{B}^\natural \in \mathbb{R}^{p \times m}$ is the matrix of signals, and $\mathbf{W} \in \mathbb{R}^{n \times m}$ represents the additive noise matrix and its entries $\mathbf{W}_{ij}$ are i.i.d standard normal random variables. In addition, we denote $h$ as the number of permuted rows corresponding to the permutation matrix $\mathbf{\Pi}^\natural$.

In this work, we focus on studying the "phase transition" phenomenon in recovering $\mathbf{\Pi}^\natural$ from the pair $(\mathbf{Y}, \mathbf{X})$. That is, the error rate for the permutation recovery sharply drops to zero once certain parameters reach the thresholds. In particular, our analysis will identify the precise positions of the phase transition points in the large-system limit, i.e., $n$, $m$, $p$, and $h$ all approach to infinity with $m/n \to \tau_m$, $p/n \to \tau_p$, $h/n \to \tau_h$. We will separately study the phase transition phenomenon in 1) the oracle case where $\mathbf{B}^\natural$ is given as a prior, and 2) the non-oracle case where $\mathbf{B}^\natural$ is unknown.

In this section, we consider the oracle scenario, as a warm-up example. To reconstruct the permutation matrix $\mathbf{\Pi}^\natural$, we adopt the following *maximum-likelihood* (ML) estimator:

$$\widehat{\mathbf{\Pi}}^{\text{oracle}} = \text{argmin}_{\mathbf{\Pi} \in \mathcal{P}_n} \left\langle \mathbf{\Pi}, -\mathbf{Y} \mathbf{B}^{\natural\top} \mathbf{X}^\top \right\rangle. \tag{6}$$

Denoting the variable $\mathbf{E}_{ij}^{\text{oracle}}$ as $-\mathbf{X}_{\pi^\natural(i)}^\top \mathbf{B}^\natural \mathbf{B}^{\natural\top} \mathbf{X}_j - \sigma \mathbf{W}_i^\top \mathbf{B}^{\natural\top} \mathbf{X}_j$, $(1 \leq i, j \leq n)$, we can transform the objective function in (6) as the canonical form of LAP, i.e., $\sum_{i,j} \mathbf{\Pi}_{ij} \mathbf{E}_{ij}^{\text{oracle}}$.

### 3.1 THE PHASE TRANSITION THRESHOLD FOR THE ORACLE CASE

In the oracle case where $\mathbf{B}^\natural$ is known, we define the following random variable $\Xi$:

$$\Xi = \boldsymbol{x}^\top \mathbf{B}^\natural \mathbf{B}^{\natural\top} (\boldsymbol{x} - \boldsymbol{y}) + \sigma \boldsymbol{w} \mathbf{B}^{\natural\top} (\boldsymbol{x} - \boldsymbol{y}), \tag{7}$$

where $\boldsymbol{x}$ and $\boldsymbol{y}$ follow the distribution $\mathsf{N}(\mathbf{0}, \mathbf{I}_{p \times p})$, and $\boldsymbol{w}$ follows the distribution $\mathsf{N}(\mathbf{0}, \mathbf{I}_{m \times m})$.

**Assumption 1.** *We ignore the weak correlation across the* $\mathbf{E}_{ij}^{\text{oracle}}$ *and view the corresponding* $\Xi_i$ *as i.i.d. copies of* (7).

Numerical experiments show that we can safely adopt Assumption 1 without much sacrifice in the prediction accuracy, see Table 1 and 2. Recalling Theorem 1, we predict the critical points by letting

$$\inf_{\theta>0} {}^{1}\!/\!\theta \cdot \log\left(\sum_{i=1}^{n} \mathbb{E}e^{-\theta\Xi_i}\right) = \inf_{\theta>0} {}^{1}\!/\!\theta \cdot \left(\log n + \log\mathbb{E}e^{-\theta\Xi}\right) = 0. \tag{8}$$

The computation procedure consists of two stages:

- **Step I.** We compute the optimal $\theta_*$, which is written as $\theta_* = \mathrm{argmin}_{\theta>0} {}^{1}\!/\!\theta \cdot \left(\log n + \log\mathbb{E}e^{-\theta\Xi_i}\right)$.
- **Step II.** We plug the optimal $\theta^*$ into (8) and obtain the phase transition snr accordingly.

The following context illustrates the computation details.

**Step I: Determine $\theta_*$.** The key in determining $\theta_*$ lies in the computation of $\mathbb{E}e^{-\theta\Xi}$, which is summarized in the following proposition.

**Lemma 1.** *For the random variable $\Xi$ defined in* (7)*, we can write its expectation as*

$$\mathbb{E}e^{-\theta\Xi} = \prod_{i=1}^{\mathrm{rank}(\mathbf{B}^{\natural})} \left[1 + 2\theta\lambda_i^2 - \theta^2\lambda_i^2\left(\lambda_i^2 + 2\sigma^2\right)\right]^{-\frac{1}{2}}, \tag{9}$$

*provided that*

$$\theta^2\sigma^2\lambda_i^2 < 1 \text{ and } \theta^2\lambda_i^2\left(\lambda_i^2 + 2\sigma^2\right) \le 1 + 2\theta\lambda_i^2 \tag{10}$$

*hold for all singular values $\lambda_i$ of $\mathbf{B}^{\natural}$, $1 \le i \le \mathrm{rank}(\mathbf{B}^{\natural})$.*

**Remark 1.** *When the conditions in* (10) *is violated, we have the expectation $\mathbb{E}e^{-\theta\Xi}$ diverge to infinity, which suggests the optimal $\theta_*$ for $\inf_{\theta>0} \log\left(n\cdot\mathbb{E}e^{-\theta\Xi}\right)/\theta$ cannot be achieved.*

With (9), we can compute the optimal $\theta_*$ by setting the gradient $\frac{\partial\left[\log(n\cdot\mathbb{E}e^{-\theta\Xi})/\theta\right]}{\partial\theta} = 0$. However, a closed-form of the exact solution for $\theta^*$ is out of reach. As a mitigation, we resort to approximating $\log\mathbb{E}e^{-\theta\Xi}$ by its lower-bound, which reads as

$$\log\mathbb{E}e^{-\theta\Xi} \ge \frac{\theta^2}{2}\left(\left\|\mathbf{B}^{\natural\top}\mathbf{B}^{\natural}\right\|_{\mathrm{F}}^2 + 2\sigma^2\left\|\mathbf{B}^{\natural}\right\|_{\mathrm{F}}^2\right) - \theta\left\|\mathbf{B}^{\natural}\right\|_{\mathrm{F}}^2.$$

The corresponding minimum value $\widetilde{\theta}_*$ is thus obtained by minimizing the lower-bound, which is written as $\widetilde{\theta}_* = 2\log n / \left(\left\|\mathbf{B}^{\natural\top}\mathbf{B}^{\natural}\right\|_{\mathrm{F}}^2 + 2\sigma^2\left\|\mathbf{B}^{\natural}\right\|_{\mathrm{F}}^2\right)$.

**Step II: Compute the phase transition snr.** We predict the phase transition point $\mathsf{snr}_{\mathrm{oracle}}$ by letting the lower bound being zero, which can be written as

$$\frac{\log n}{\theta^*} - \left\|\mathbf{B}^{\natural}\right\|_{\mathrm{F}}^2 + \frac{\theta^*}{2}\left(\left\|\mathbf{B}^{\natural\top}\mathbf{B}^{\natural}\right\|_{\mathrm{F}}^2 + 2\sigma^2\left\|\mathbf{B}^{\natural}\right\|_{\mathrm{F}}^2\right) = 0.$$

With standard algebraic manipulations, we have

**Proposition 2.** *The predicted phase transition for the oracle case in* (6) *can be computed as*

$$2(\log n)\mathsf{snr}_{\mathrm{oracle}} \cdot \left\|\mathbf{B}^{\natural\top}/\|\mathbf{B}^{\natural}\|_{\mathrm{F}} \cdot \mathbf{B}^{\natural}/\|\mathbf{B}^{\natural}\|_{\mathrm{F}}\right\|_{\mathrm{F}}^2 + 4\log n/m = \mathsf{snr}_{\mathrm{oracle}}. \tag{11}$$

We then evaluate the accuracy of our predicted phase transition threshold by comparing the predicted values with the numerical values.

## 3.2 Numerical Method for Phase Transition Points

Noticing that the correct recovery rate is in monotonic non-decreasing relation with the snr, that is, the error rate $\mathbb{P}(\widehat{\mathbf{\Pi}} \ne \mathbf{\Pi}^{\natural})$ for a larger snr is at least equal to, if not less than, that for a smaller snr, we propose a binary-search-based method to compute the phase transition points. The detailed description is in Algorithm 1.

**Complexity analysis**. For a given precision threshold $\varepsilon$, each iteration (Line 2 to Line 12 in Algorithm 1) takes $O(\log\frac{1}{\varepsilon})$ rounds to converge and it runs the permutation recovery algorithm $T_{\mathrm{iter}}$ times in each round. That is, the computational complexity of phase transition point for each configuration is $O(T_{\mathrm{iter}} \cdot \log\frac{1}{\varepsilon})$.

**Experiment result**. The results are shown in Table 1, from which we can conclude the phase transition threshold snr can be predicted to a good extent. In addition, we observe that the gap between the theoretical values and the numerical values keeps shrinking as $m$ increases.

---

**Algorithm 1** Numerical method to compute the phase transition points.

---

1: **Initialization.** Set the initial search range for snr as $[l, r]$. Define the precision threshold $\varepsilon$, the error rate Err (we pick it as 20) and iterations $T_{\text{iter}}$.
2: **while** $|l - r| > \varepsilon$ **do**
3:     Set $\text{snr}_{\text{middle}} = \frac{l+r}{2}$.
4:     Given $\text{snr}_{\text{middle}}$, we run (2) for $T_{\text{iter}}$ times.
5:     Compute the error rate of full permutation recovery, namely, $\mathbb{P}(\widehat{\mathbf{\Pi}} \neq \mathbf{\Pi}^\natural)$.
6:
7:     **if** the error rate is below Err **then**
8:         $r \to \text{snr}_{\text{middle}} - \varepsilon$,   # we have $\text{snr}_{\text{middle}}$ be greater than the phase transition point
9:     **else**
10:         $l \to \text{snr}_{\text{middle}} + \varepsilon$.  # we have $\text{snr}_{\text{middle}}$ be no greater than the phase transition point
11:     **end if**
12: **end while**
13: **Output.** Return the phase transition point $\text{snr}_{\text{middle}}$.

---

Table 1:   Comparison between the predicted value of the phase transition threshold $\text{snr}_{\text{oracle}}$ in Proposition 2 and its simulated value when $n = 500$. **P** denotes the predicted value while **S** denotes the simulated value (i.e., mean $\pm$ std). **S** corresponds to the snr when the error rate drops below 0.05. A detailed description of the numerical method can be found in the appendix (code also included).

| $m$ | 20 | 30 | 40 | 50 | 60 | 70 |
|---|---|---|---|---|---|---|
| **P** | 3.283 | 1.415 | 0.902 | 0.662 | 0.523 | 0.432 |
| **S** | $2.529 \pm 0.079$ | $1.290 \pm 0.054$ | $0.872 \pm 0.034$ | $0.649 \pm 0.012$ | $0.515 \pm 0.016$ | $0.429 \pm 0.015$ |

| $m$ | 100 | 110 | 120 | 130 | 140 | 150 |
|---|---|---|---|---|---|---|
| **P** | 0.284 | 0.255 | 0.231 | 0.211 | 0.195 | 0.181 |
| **S** | $0.282 \pm 0.008$ | $0.256 \pm 0.006$ | $0.232 \pm 0.006$ | $0.212 \pm 0.004$ | $0.196 \pm 0.006$ | $0.183 \pm 0.005$ |

### 3.3 Gaussian approximation of the phase transition threshold

From the above analysis, we can see that deriving a closed-form expression of the infimum value $\theta$ of $\log(n\mathbb{E}e^{-\theta\Xi})/\theta$ can be difficult. In fact, in certain scenarios, even obtaining a closed-form expression of $\mathbb{E}e^{-\theta\Xi}$ is difficult. To handle such challenge, we propose to approximate random variable $\Xi$ by a Gaussian $\mathsf{N}(\mathbb{E}\Xi, \text{Var}\Xi)$, namely,

$$\mathbb{E}e^{-\theta\Xi} \approx \exp\left(-\theta\mathbb{E}\Xi + \frac{\theta^2}{2}\text{Var}\Xi\right). \tag{12}$$

With this approximation, we can express $\theta_* \triangleq \inf \log(n \cdot \mathbb{E}e^{-\theta\Xi})/\theta$ in a closed form, which is $\sqrt{2\log n/\text{Var}\Xi}$.

**Theorem 2.** *For the random variable $\Xi$ defined in (7), its mean and variance can be computed as*

$$\mathbb{E}\Xi = \left\|\mathbf{B}^\natural\right\|_{\text{F}}^2, \quad \text{Var}\Xi = 3\left\|\mathbf{B}^\natural\mathbf{B}^{\natural\top}\right\|_{\text{F}}^2 + 2\sigma^2\left\|\mathbf{B}^\natural\right\|_{\text{F}}^2. \tag{13}$$

Then, we can predict the phase transition point as follows.

**Proposition 3.** *With Gaussian approximation, we can predict the critical point corresponding to the phase transition in (8) as*

$$2(\log n) \cdot \text{Var}\Xi = (\mathbb{E}\Xi)^2, \tag{14}$$

*where $\mathbb{E}\Xi$ and $\text{Var}\Xi$ can be found in Theorem 2.*

**Example 1** (Scaled identity matrix). *We consider the scenario where $\mathbf{B}^\natural = \lambda\mathbf{I}_{m\times m}$. Then, we have $\mathbf{B}^\natural/\|\mathbf{B}^\natural\|_{\text{F}} = m^{-1/2}\mathbf{I}$. The phase transition threshold $\text{snr}_{\text{oracle}}$ in (11) is then $4\log n/(m-2\log n)$, and the phase transition threshold $\widetilde{\text{snr}}_{\text{oracle}}$ in (14) as $4\log n/(m-6\log n)$. This solution is almost identical to (11) in the limit as $\text{snr}_{\text{oracle}} \approx \widetilde{\text{snr}}_{\text{oracle}} \approx 4\log n/m \simeq n^{\frac{4}{m}} - 1$.*

Moreover, we should mention that 1) our approximation method applies to a general matrix $\mathbf{B}^{\natural}$, not limited to a scaled identity matrix; and 2) our approximation method can also predict the phase transition thresholds to a good extent when the entries $\mathbf{X}_{ij}$ are sub-Gaussian. The numerical experiments are given in Table 2, from which we can conclude that the predicted values are well aligned with the simulation results.

Table 2: Comparison between the predicted value of the phase transition threshold $\widetilde{\mathsf{snr}}_{\text{oracle}}$ in Proposition 3 and its simulated value when $n = 600$. In (**Case 1**), half of singular values are with $\lambda$ and the other half are with $\lambda/2$; while in (**Case 2**), half of the singular values are with $\lambda$ and the other half are with $^{(3 \cdot \lambda)}/_4$. **Gauss** refers to $\mathbf{X}_{ij} \overset{\text{i.i.d}}{\sim} \mathsf{N}(0, 1)$ and **Unif** refers to $\mathbf{X}_{ij} \overset{\text{i.i.d}}{\sim} \mathsf{Unif}[-1, 1]$. **P** denotes the predicted value and **S** denotes the simulated value (i.e., mean $\pm$ std). **S** corresponds to the snr when the error rate drops below 0.05. Averaged over 20 repetitions.

| $m$ | | 100 | 110 | 120 | 130 | 140 | 150 |
|---|---|---|---|---|---|---|---|
| (**Case 1**) **P** | | 0.297 | 0.266 | 0.241 | 0.220 | 0.203 | 0.188 |
| (**Gauss**) **S** | | $0.307 \pm 0.009$ | $0.275 \pm 0.005$ | $0.246 \pm 0.006$ | $0.227 \pm 0.007$ | $0.210 \pm 0.005$ | $0.194 \pm 0.004$ |
| (**Unif**) **S** | | $0.294 \pm 0.008$ | $0.266 \pm 0.005$ | $0.239 \pm 0.008$ | $0.216 \pm 0.004$ | $0.201 \pm 0.005$ | $0.189 \pm 0.006$ |
| (**Case 2**) **P** | | 0.310 | 0.276 | 0.249 | 0.227 | 0.209 | 0.193 |
| (**Gauss**) **S** | | $0.294 \pm 0.008$ | $0.266 \pm 0.006$ | $0.241 \pm 0.005$ | $0.220 \pm 0.004$ | $0.204 \pm 0.006$ | $0.190 \pm 0.003$ |
| (**Unif**) **S** | | $0.287 \pm 0.007$ | $0.255 \pm .0043$ | $0.234 \pm 0.007$ | $0.213 \pm 0.005$ | $0.197 \pm 0.003$ | $0.185 \pm 0.005$ |

### 3.4 RELATED WORK COMPARISON

In the end, we want to discuss one closely related work (Lufkin et al., 2024), which studies the same topic but in a much simpler setting (i.e., single measurement with $m = 1$). Compared with their work, our framework can easily produce results that the rigorous method in Lufkin et al. (2024) regards as an open problem. For example, the results (on the oracle case) which we treat as a warm-up example, are unsolved and specifically mentioned in the last paragraph. In particular, our framework can derive their proposed conjectured phase transition $\mathsf{snr} = n^{4/m}$. Moreover, our framework can also tell when the conjecture holds, i.e., $\mathbf{B}^{\natural}$ is an identity matrix, and our predictions' accuracy has been extensively verified by numerical experiments.

Other advantages of our work include 1) our ability to more accurately pinpoint phase transition points (their work can only obtain the lower bound of the phase transition point, while our work can predict the precise location); 2) our applicability to a wider array of cases (we cover the case when $m > 1$ while their method only works for $m = 1$); and 3) our consolidation into a more cohesive framework to predict the phase transition point.

## 4 EXTENSION TO NON-ORACLE CASE

Having analyzed the oracle case in the previous section, we now extend the analysis to the non-oracle case, where the value of $\mathbf{B}^{\natural}$ is not given. Different from the oracle case, the ML estimator reduces to a *quadratic assignment problem* (QAP) as opposed to LAP. As a mitigation, we adopt the estimator in Zhang & Li (2020), which reconstructs the permutation matrix within the LAP framework, i.e.,

$$\widehat{\mathbf{\Pi}}^{\text{non-oracle}} = \operatorname{argmin}_{\mathbf{\Pi} \in \mathcal{P}_n} \left\langle \mathbf{\Pi}, -\mathbf{Y}\mathbf{Y}^{\top}\mathbf{X}\mathbf{X}^{\top} \right\rangle. \tag{15}$$

We expect this estimator can yield good insights of the permuted linear regression since 1) this estimator can reach the statistical optimality in a broad range of parameters; and 2) estimator exhibits a phase transition phenomenon, a similar pattern as the oracle case. The technical details of the above claims can be found in Zhang & Li (2020).

Following the same procedure as in Section 3, we identify the phase transition threshold snr with Theorem 1. First, we rewrite the random variable $\Xi$ as

$$\Xi \cong \mathbf{Y}_i \mathbf{Y}^{\top} \mathbf{X} \left( \mathbf{X}_{\pi^{\natural}(i)} - \mathbf{X}_j \right)^{\top}, \tag{16}$$

where $i$ and $j$ are uniformly distributed among the set $\{1, 2, \cdots, n\}$. Afterwards, we adopt the Gaussian approximation scheme illustrated in Subsection 3.3 and determine the phase transition points by first computing $\mathbb{E}\Xi$ and $\mathsf{Var}\Xi$, respectively.

**Theorem 3.** *For the random variable $\Xi$ defined in* (16), *its mean $\mathbb{E}\Xi$ and variance $\mathsf{Var}\Xi$ are*

$$\mathbb{E}\Xi \simeq n\left(1 - \tau_h\right)\left[\left(1 + \tau_p\right)\left\|\!\left\|\mathbf{B}^\natural\right\|\!\right\|_{\mathrm{F}}^2 + m\tau_p\sigma^2\right],$$

$$\mathrm{Var}\Xi \simeq n^2\tau_h\left(1 - \tau_h\right)\tau_p^2\left[\left\|\!\left\|\mathbf{B}^\natural\right\|\!\right\|_{\mathrm{F}}^2 + m\sigma^2\right]^2 + n^2\left[2\tau_p + 3\left(1 - \tau_h\right)^2\right]\left\|\!\left\|\mathbf{B}^{\natural\top}\mathbf{B}^\natural\right\|\!\right\|_{\mathrm{F}}^2$$

$$+ n^2\left[6\tau_p\left(1 - \tau_h\right)^2 + \left(3 - \tau_h\right)\tau_p^2\right]\left\|\!\left\|\mathbf{B}^{\natural\top}\mathbf{B}^\natural\right\|\!\right\|_{\mathrm{F}}^2,$$

*respectively, where the definitions of $\tau_p$ and $\tau_h$ can be found in Section 3.*

The proof of Theorem 3 is quite complicated, involving Wick's theorem, Stein's lemma, the conditional technique, and the leave-one-out technique, etc. For clarity, we provide only an outline of the proof strategy here.

*Proof outline.* We decompose the random variable $\Xi$ as $\Xi = \Xi_1 + \sigma\left(\Xi_2 + \Xi_3\right) + \sigma^2\Xi_4$, where $\Xi_i$ $(1 \le i \le 4)$ are respectively defined as

$$\Xi_1 \triangleq \mathbf{X}_{\pi^\natural(i)}^\top\mathbf{B}^\natural\mathbf{B}^{\natural\top}\mathbf{X}^\top\mathbf{\Pi}^{\natural\top}\mathbf{X}(\mathbf{X}_{\pi^\natural(i)} - \mathbf{X}_j), \qquad \Xi_2 \triangleq \mathbf{X}_{\pi^\natural(i)}^\top\mathbf{B}^\natural\mathbf{W}^\top\mathbf{X}(\mathbf{X}_{\pi^\natural(i)} - \mathbf{X}_j),$$

$$\Xi_3 \triangleq \mathbf{W}_i^\top\mathbf{B}^{\natural\top}\mathbf{X}^\top\mathbf{\Pi}^{\natural\top}\mathbf{X}(\mathbf{X}_{\pi^\natural(i)} - \mathbf{X}_j), \qquad \Xi_4 \triangleq \mathbf{W}_i^\top\mathbf{W}^\top\mathbf{X}(\mathbf{X}_{\pi^\natural(i)} - \mathbf{X}_j).$$

Unlike the oracle case, obtaining a closed-form expression of $\mathbb{E}e^{-\theta\Xi}$ would be too difficult. Hence, we adopt the Gaussian approximation method as presented in Section 3.3. The task then transforms to computing the expectation and variance of $\Xi$.

**Computation of the mean $\mathbb{E}\Xi$.** For the computation of the mean $\mathbb{E}\Xi$, we can verify that $\mathbb{E}\Xi_2$ and $\mathbb{E}\Xi_3$ are both zero, due to the independence between $\mathbf{X}$ and $\mathbf{W}$. For $\mathbb{E}\Xi_1$ and $\mathbb{E}\Xi_4$, we adopt Wick's theorem to obtain

$$\mathbb{E}\Xi_1 = n\left(1 - \tau_h\right)\left(1 + \tau_p\right)\left[1 + o_{\mathrm{P}}\left(1\right)\right]\left\|\!\left\|\mathbf{B}^\natural\right\|\!\right\|_{\mathrm{F}}^2, \quad \mathbb{E}\Xi_4 = nm\tau_p\left(1 - \tau_h\right)\left[1 + o_{\mathrm{P}}\left(1\right)\right].$$

**Computation of the variance $\mathrm{Var}\Xi$.** Since $\mathrm{Var}\Xi = \mathbb{E}\Xi^2 - \left(\mathbb{E}\Xi\right)^2$, we just need to compute $\mathbb{E}\Xi^2$, which can be expanded into the following six terms

$$\mathbb{E}\Xi^2 = \mathbb{E}\Xi_1^2 + \sigma^2\mathbb{E}\Xi_2^2 + \sigma^2\mathbb{E}\Xi_3^2 + \sigma^4\mathbb{E}\Xi_4^2 + 2\sigma^2\mathbb{E}\Xi_1\Xi_4 + 2\sigma^2\mathbb{E}\Xi_2\Xi_3.$$

The computation of above terms turns out to be quite complex due to the high order Gaussian random variables. For example, the term $\mathbb{E}\Xi_1^2$ involves the eighth-order Gaussian moments, the terms $\mathbb{E}\Xi_2^2, \mathbb{E}\Xi_3^2, \mathbb{E}\Xi_1\Xi_4$ and $\mathbb{E}\Xi_2\Xi_3$ all involve the sixth-order Gaussian variables, etc. To handle the difficulties in computing $\mathbb{E}\Xi^2$, we propose the following computation procedure, which can be roughly divided into 3 phases.

- **Phase I: Leave-one-out decomposition.** The major technical difficulty comes from the correlation between the product $\mathbf{X}^\top\mathbf{\Pi}^\natural\mathbf{X}$ and the difference $\mathbf{X}_{\pi^\natural(i)} - \mathbf{X}_j$. We decouple this correlation by first rewriting the matrix $\mathbf{X}^\top\mathbf{\Pi}^\natural\mathbf{X}$ as the sum $\sum_\ell \mathbf{X}_\ell\mathbf{X}_{\pi^\natural(\ell)}^\top$. Then we collect all terms $\mathbf{X}_\ell\mathbf{X}_{\pi^\natural(\ell)}^\top$ independent of $\mathbf{X}_{\pi^\natural(i)}$ and $\mathbf{X}_j$ in the matrix $\mathbf{\Sigma}$ and leave the remaining terms to the matrix $\mathbf{\Delta}$, i.e., $\mathbf{\Delta} \triangleq \mathbf{X}^\top\mathbf{\Pi}^\natural\mathbf{X} - \mathbf{\Sigma}$. This decomposition shares the same spirit as the leave-one-out technique (Karoui, 2013; Bai & Silverstein, 2010; Karoui, 2018; Sur et al., 2019). Then, we divide all terms in $\mathbb{E}\Xi^2$ into 3 categories: 1) terms only containing matrix $\mathbf{\Sigma}$; 2) terms containing both $\mathbf{\Sigma}$ and $\mathbf{\Delta}$; and 3) terms only containing $\mathbf{\Delta}$.

- **Phase II: Conditional technique.** Concerning the terms in the first two categories, which covers the majority of terms, we can exploit the independence among the rows in the sensing matrix $\mathbf{X}$. With the conditional technique, we can reduce the order of Gaussian moments by separately taking the expectation w.r.t $\mathbf{\Sigma}$ and w.r.t vectors $\mathbf{X}_{\pi^\natural(i)}$ and $\mathbf{X}_j$.

- **Phase III: Direct computation.** For the few terms in the third category (i.e., terms only containing $\mathbf{\Delta}$), we compute the high-order Gaussian moments by exhausting all terms and iterative applying of Wick's Theorem and Stein's Lemma, which can reduce the higher-order Gaussian moments to lower-orders.

For the interested readers, the technical details can be found in Section $D$.

### 4.1 AN ILLUSTRATING EXAMPLE

Afterwards, we predict the phase transition points. Unlike the oracle case, we notice the edge weights $\mathbf{E}_{ij}$ are strongly correlated, especially when $j = \pi^\natural(j)$, which corresponds to the non-permuted rows. To factor out these dependencies, we only take the permuted rows into account and correct the sample size from $n$ to $\tau_h n$.

**Proposition 4.** *The predicted* $\mathsf{snr}_{\text{non-oracle}}$ *for the non-oracle case in* (15) *can be computed by solving*

$$2\log(n\tau_h)\text{Var}\Xi = (\mathbb{E}\Xi)^2\,,$$

*where $\mathbb{E}\Xi$ and $\text{Var}\Xi$ are in Theorem 3.*

To illustrate the prediction accuracy, we consider the case where $\mathbf{B}^\natural$'s singular values are of the same order, i.e., $\frac{\lambda_i(\mathbf{B}^\natural)}{\lambda_j(\mathbf{B}^\natural)} = O(1)$, $1 \le i, j \le m$, where $\lambda_i(\cdot)$ denotes the $i$-th singular value. Then, we obtain the $\mathsf{snr}_{\text{non-oracle}}$, which is written as

$$\mathsf{snr}_{\text{non-oracle}} \approx \eta_1/\eta_2. \tag{17}$$

Here, $\eta_1$ and $\eta_2$ are defined as

$$\eta_1 \triangleq 2\tau_h\tau_p^2\log(n\tau_h) - \tau_p(\tau_p+1)(1-\tau_h) + \tau_p\sqrt{2(1-\tau_h)\tau_h\cdot\log(n\tau_h)},$$

$$\eta_2 \triangleq (1-\tau_h)(\tau_p+1)^2 - 2\tau_h\tau_p^2\log(n\tau_h).$$

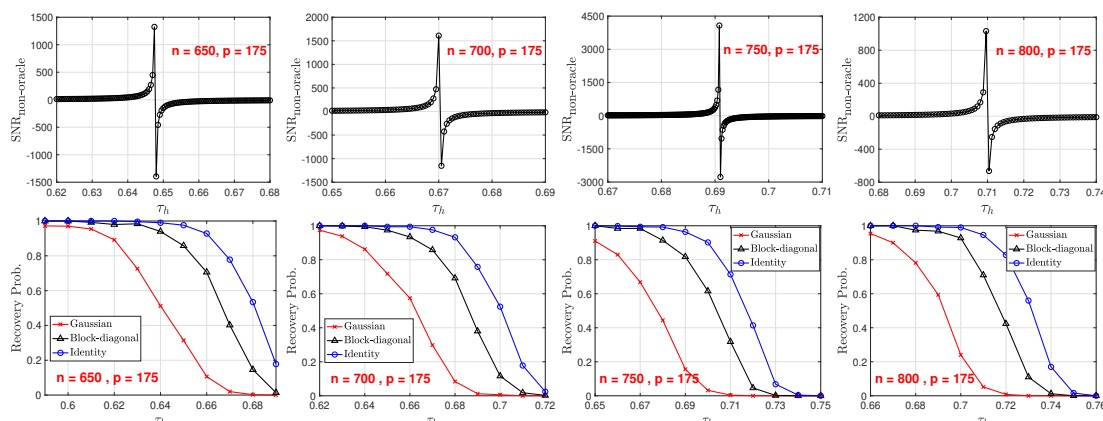

Figure 1: (**Upper panel**) Predicted phase transition points $\mathsf{snr}_{\text{non-oralce}}$. (**Lower panel**) Plot of the recovery rate under the noiseless setting, i.e., $\mathsf{snr} = \infty$. **Gaussian**: $\mathbf{B}^\natural_{ij} \overset{\text{i.i.d}}{\sim} \mathsf{N}(0,1)$; **Identity**: $\mathbf{B}^\natural = \mathbf{I}_{p\times p}$; **Block-diagonal**: $\mathbf{B}^\natural = \text{diag}\{1,\cdots,1,0.5,\cdots,0.5\}$. We observe that the correct recovery rates drop sharply within the regions of our predicted value.

Note that the predicted $\mathsf{snr}_{\text{non-oracle}}$ varies for different $\tau_h$ and $\tau_p$. Viewing $\mathsf{snr}_{\text{non-oracle}}$ as a function of $\tau_h$, we observe a singularity point of $\tau_h$, which corresponds to the case when $\eta_2 = 0$. This suggests a potential phase transition phenomenon w.r.t. $\tau_h$. To validate the predicted phenomenon, we consider the noiseless case, i.e., $\mathsf{snr} = \infty$, and reconstruct the permutation matrix $\mathbf{\Pi}^\natural$ with (2). Numerical experiments in Figure 1 confirm our prediction.

Due to the space limit, this section only presents a glimpse of our results in the non-oracle case. The technical details along with the additional numerical experiments can be found in Section C.

## 5 CONCLUSIONS

This is the first work that can identify the precise location of phase transition thresholds of permuted linear regressions. For the oracle case where the signal $\mathbf{B}^\natural$ is given as a prior, our analysis can predict the phase transition threshold $\mathsf{snr}_{\text{oracle}}$ to a good extent. For the non-oracle case where $\mathbf{B}^\natural$ is not given, we modified the leave-one-out technique to approximately compute the phase critical $\mathsf{snr}_{\text{non-oracle}}$ value for the phase transition, as the precise computation becomes significantly complicated as the high-order interaction between Gaussian random variables is involved. Moreover, we associated the singularity point in $\mathsf{snr}_{\text{non-oracle}}$ with a phase transition point w.r.t the allowed number of permuted rows. Moreover, we present numerous numerical experiments to confirm the accuracy of our theoretical predictions.

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

## A    NOTATIONS

We repeat the notations for the self-containing of the appendix. We denote $a \xrightarrow{\text{a.s.}} b$ when $a$ converges almost surely to $b$. $f(n) \simeq g(n)$ denotes $\lim_{n\to\infty} f(n)/g(n) = 1$, and $f(n) = \mathbb{O}_{\mathrm{P}}(g(n))$ if the sequence $f(n)/g(n)$ is bounded in probability, and $f(n) = o_{\mathrm{P}}(g(n))$ if $f(n)/g(n)$ converges to zero in probability. The inner product between two vectors (resp. matrices) are denoted as $\langle \cdot, \cdot \rangle$. For two distributions $d_1$ and $d_2$, we write $d_1 \cong d_2$ if they are equal up to normalization. Moreover, $\mathcal{P}_n$ corresponds to the set of all possible permutation matrices: $\mathcal{P}_n \triangleq \{\mathbf{\Pi} \in \{0,1\}^{n\times n}, \sum_i \mathbf{\Pi}_{ij} = 1, \sum_j \mathbf{\Pi}_{ij} = 1\}$. The *signal-to-noise-ratio* is $\mathsf{snr} = \frac{\|\mathbf{B}^\natural\|_{\mathrm{F}}^2}{m\cdot\sigma^2}$, where $\|\cdot\|_{\mathrm{F}}$ is the Frobenius norm and $\sigma^2$ is the variance of the sensing noise.

## B    LINK BETWEEN PERMUTATION RECOVERY WITH MESSAGE PASSING

### B.1    CONSTRUCTION OF THE GRAPHICAL MODEL

First, we construct the factor graph associated with the probability measure in (3). Adopting the same strategy as in Chapter 16 of Mezard & Montanari (2009), we conduct the following operations, e.g., 1) associating each variable $\mathbf{\Pi}_{ij}$ a variable node $v_{ij}$; 2) associating the variable node $v_{ij}$ a function node representing the term $e^{-\beta \mathbf{\Pi}_{ij} \mathbf{E}_{ij}}$; and 3) linking each constraint $\sum_i \mathbf{\Pi}_{ij} = 1$ to a function node and similarly for the constraint $\sum_j \mathbf{\Pi}_{ij} = 1$. A graphical representation is available in Figure 2.

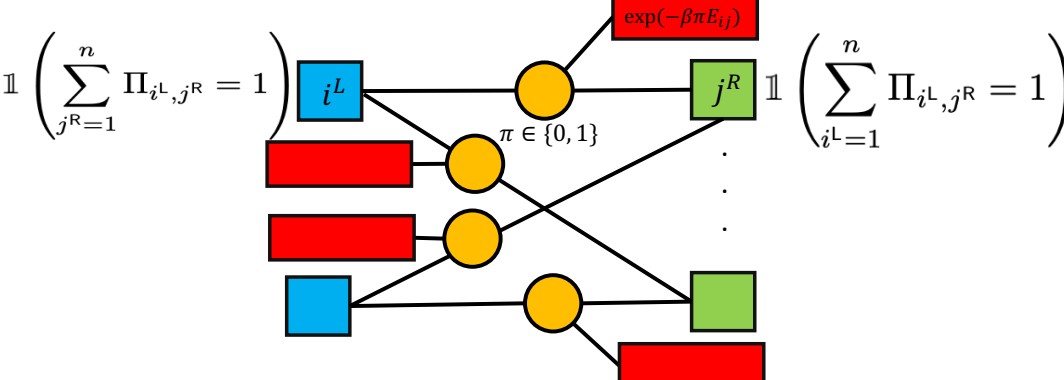

Figure 2: The constructed graphical model. Circle icons denote the variable nodes and square icons denote the function nodes: blue squares (green squares resp.) for the constraints on the rows (columns resp.) of $\mathbf{\Pi}$, and red squares for the function $e^{-\beta\pi\mathbf{E}_{ij}}$.

Now we briefly review the MP algorithm. Informally speaking, MP is a local algorithm to compute the marginal probabilities over the graphical model. In each iteration, the variable node $v$ transmits the message to its incident function node $f$ by multiplying all incoming messages except the message along the edge $(v, f)$. The function node $f$ transmits the message to its incident variable node $v$ by computing the weighted summary of all incoming messages except the message along the edge $(f, v)$. For a detailed introduction to MP, we refer readers to Kschischang et al. (2001), Chapter 16 in MacKay et al. (2003), and Chapter 14 in Mezard & Montanari (2009).

It is known that MP can obtain the exact marginals (Mezard & Montanari, 2009) for singly connected graphical models. For other types of graphs, however, whether MP can obtain the exact solution still remains an open problem (Cantwell & Newman, 2019; Kirkley et al., 2021). At the same time, numerical evidences have been witnessed to show that MP can yield meaningful results for graphs with loops; particular examples include applications in the coding theory (Chung, 2000; Richardson & Urbanke, 2001; 2008) and the LAP (which happens to be our case) (Mezard & Montanari, 2009; Chertkov et al., 2010; Caracciolo et al., 2017; Malatesta et al., 2019; Semerjian et al., 2020).

## B.2 THE MESSAGE PASSING (MP) ALGORITHM

Next, we perform permutation recovery via MP. The following derivation follows the standard procedure, which can be found in the previous works (Mezard & Montanari, 2009; Semerjian et al., 2020). We denote the message flow from the node $i^\mathsf{L}$ to the variable node $(i^\mathsf{L}, j^\mathsf{R})$ as $\widehat{m}_{i^\mathsf{L} \to (i^\mathsf{L}, j^\mathsf{R})}(\cdot)$ and that from the edge $(i^\mathsf{L}, j^\mathsf{R})$ to node $i^\mathsf{L}$ as $m_{(i^\mathsf{L}, j^\mathsf{R}) \to i^\mathsf{L}}(\cdot)$. Similarly, we define $\widehat{m}_{j^\mathsf{R} \to (i^\mathsf{L}, j^\mathsf{R})}(\cdot)$ and $m_{(i^\mathsf{L}, j^\mathsf{R}) \to j^\mathsf{R}}(\cdot)$ as the message flow transmitted between the functional node $j^\mathsf{R}$ and the variable node $(i^\mathsf{L}, j^\mathsf{R})$. Here the superscripts $\mathsf{L}$ and $\mathsf{R}$ are used to indicate the positions of the node (left and right, respectively). Roughly speaking, these transmitted messages can be viewed as (unnormalized) conditional probability $\mathbb{P}(\Pi_{i,j} = \{0, 1\}|(\cdot))$ with the joint PDF being defined in (3). The message transmission process is to iteratively compute these conditional probabilities.

First, we consider the message flows transmitted between the functional node $i^\mathsf{L}$ and the variable node $(i^\mathsf{L}, j^\mathsf{R})$, which are written as

$$m_{(i^\mathsf{L}, j^\mathsf{R}) \to i^\mathsf{L}}(\pi) \cong \widehat{m}_{j^\mathsf{R} \to (i^\mathsf{L}, j^\mathsf{R})}(\pi) e^{-\beta \pi \mathbf{E}_{i^\mathsf{L}, j^\mathsf{R}}},$$

$$\widehat{m}_{i^\mathsf{L} \to (i^\mathsf{L}, j^\mathsf{R})}(\pi) \cong \sum_{\pi_{i^\mathsf{L}, k^\mathsf{R}}} \prod_{k^\mathsf{R} \neq j^\mathsf{R}} \widehat{m}_{k^\mathsf{R} \to (i^\mathsf{L}, k^\mathsf{R})}(\pi_{i^\mathsf{L}, k^\mathsf{R}}) \times e^{-\beta \pi_{i^\mathsf{L}, k^\mathsf{R}} \mathbf{E}_{i^\mathsf{L}, k^\mathsf{R}}} \mathbb{1}(\pi + \sum_k \pi_{i^\mathsf{L}, k^\mathsf{R}} = 1), \quad (18)$$

where $\pi \in \{0, 1\}$ is a binary value. Similarly, we can write the message flows between the functional node $j^\mathsf{R}$ and the variable node $(i^\mathsf{L}, j^\mathsf{R})$, which are denoted as $m_{(i^\mathsf{L}, j^\mathsf{R}) \to j^\mathsf{R}}(\pi)$ and $\widehat{m}_{j^\mathsf{R} \to (i^\mathsf{L}, j^\mathsf{R})}(\pi)$, respectively. With the parametrization approach, we define

$$h_{i^\mathsf{L} \to (i^\mathsf{L}, j^\mathsf{R})} \triangleq \frac{1}{\beta} \log \frac{\widehat{m}_{i^\mathsf{L} \to (i^\mathsf{L}, j^\mathsf{R})}(1)}{\widehat{m}_{i^\mathsf{L} \to (i^\mathsf{L}, j^\mathsf{R})}(0)}, \quad h_{j^\mathsf{R} \to (i^\mathsf{L}, j^\mathsf{R})} \triangleq \frac{1}{\beta} \log \frac{\widehat{m}_{j^\mathsf{R} \to (i^\mathsf{L}, j^\mathsf{R})}(1)}{\widehat{m}_{j^\mathsf{R} \to (i^\mathsf{L}, j^\mathsf{R})}(0)}.$$

Following the routine derivations in MP, we get the edge selection criteria, i.e., we pick $\widehat{\pi}(i^\mathsf{L}) = j^\mathsf{R}$ if

$$h_{i^\mathsf{L} \to (i^\mathsf{L}, j^\mathsf{R})} + h_{j^\mathsf{R} \to (i^\mathsf{L}, j^\mathsf{R})} > \mathbf{E}_{i^\mathsf{L}, j^\mathsf{R}}; \quad (19)$$

otherwise, we have $\widehat{\pi}(i^\mathsf{L}) \neq j^\mathsf{R}$. Due to the fact that $\mu(\mathbf{\Pi})$ concentrates on $\widehat{\mathbf{\Pi}}$ when $\beta$ is sufficiently large, we can thus rewrite the MP update equation as

$$h_{i^\mathsf{L} \to (i^\mathsf{L}, j^\mathsf{R})} = \min_{k^\mathsf{R} \neq j^\mathsf{R}} \mathbf{E}_{i^\mathsf{L}, k^\mathsf{R}} - h_{k^\mathsf{R} \to (i^\mathsf{L}, k^\mathsf{R})}, \quad h_{j^\mathsf{R} \to (i^\mathsf{L}, j^\mathsf{R})} = \min_{k^\mathsf{L} \neq i^\mathsf{L}} \mathbf{E}_{k^\mathsf{L}, j^\mathsf{R}} - h_{k^\mathsf{L} \to (k^\mathsf{L}, j^\mathsf{R})}, \quad (20)$$

which is attained by letting $\beta \to \infty$.

## C ANALYSIS OF ORACLE CASE: PROOF OF LEMMA 1

*Proof.* Denote the singular values of $\mathbf{B}^\natural$ as $\{\lambda_i\}_{i=1}^{\mathrm{rank}(\mathbf{B}^\natural)}$. We exploit the rotation invariance property of Gaussian random variables; and have $\mathbf{\Xi}$ be identically distributed as

$$\mathbf{\Xi} = \sum_{i=1}^{\mathrm{rank}(\mathbf{B}^\natural)} \lambda_i^2 x_i (x_i - y_i) + \sigma \sum_{i=1}^{\mathrm{rank}(\mathbf{B}^\natural)} \lambda_i w_i (x_i - y_i).$$

Due to the independence across $\boldsymbol{w}$, $\boldsymbol{x}$, and $\boldsymbol{y}$, we have

$$\mathbb{E} e^{-\theta \mathbf{\Xi}} = \prod_{i=1}^{\mathrm{rank}(\mathbf{B}^\natural)} \mathbb{E}_{x,y,w} \exp \left[ -\theta \lambda_i^2 x (x - y) - \theta \sigma \lambda_i w (x - y) \right]$$

$$= \prod_{i=1}^{\mathrm{rank}(\mathbf{B}^\natural)} \mathbb{E}_{x,y} \exp \left( \frac{\theta \lambda_i^2 (x - y) (\theta \sigma^2 (x - y) - 2x)}{2} \right)$$

$$\stackrel{\text{\tiny\textcircled{1}}}{=} \prod_{i=1}^{\mathrm{rank}(\mathbf{B}^\natural)} \mathbb{E}_x \frac{\exp \left( \frac{\theta \lambda_i^2 x^2 (\theta (\lambda_i^2 + \sigma^2) - 2)}{2 - 2\theta^2 \lambda_i^2 \sigma^2} \right)}{\sqrt{1 - \theta^2 \lambda_i^2 \sigma^2}}$$

$$\overset{②}{=} \prod_{i=1}^{\mathrm{rank}(\mathbf{B}^\natural)} \left(1 + 2\theta\lambda_i^2 - \theta^2\lambda_i^2\left(\lambda_i^2 + 2\sigma^2\right)\right)^{-\frac{1}{2}},$$

where in ① we use the fact $\theta^2\sigma^2\lambda_i^2 < 1$ and in ② we use the fact $\theta^2\lambda_i^2\left(\lambda_i^2 + 2\sigma^2\right) \le 1 + 2\theta\lambda_i^2$. $\quad\square$

## D  ANALYSIS OF THE NON-ORACLE CASE

This section presents the technical details in analyzing the non-oracle case.

### D.1  ADDITIONAL NUMERICAL RESULTS

We consider the same settings as in Subsection 4.1. Here, we present additional numerical results to evaluate the prediction accuracy of our method.

#### D.1.1  VERIFICATION OF PHASE TRANSITION POINTS

For the predicted phase transition $\mathrm{snr}_{\text{non-oracle}}$, we notice an increasing gap between the predicted value and the simulated value, unlike in the oracle case. This might be caused by the strong correlation across the edge weights $\{\mathbf{E}_{ij}\}_{1 \le i,j \le n}$, or due to the error with the approximation relation $\mathbb{E}e^{-\theta\Xi} \approx \mathbb{E}\exp\left(\theta\mathbb{E}\Xi - \theta^2\mathrm{Var}\Xi/2\right)$.

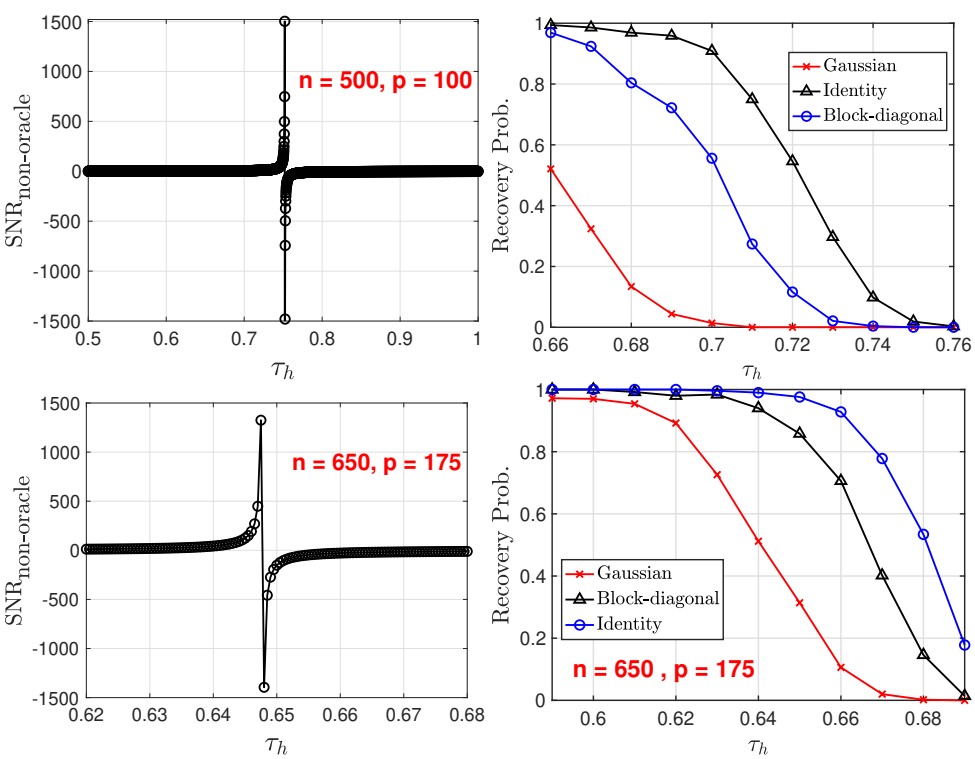

Figure 3: (**Left panel**) Predicted $\mathrm{snr}_{\text{non-oralce}}$. (**Right panel**) Plot of recovery rate under the noiseless setting, i.e., $\mathrm{snr} = \infty$. **Gaussian**: $\mathbf{B}_{ij}^\natural \overset{\text{i.i.d}}{\sim} \mathsf{N}(0,1)$; **Identity**: $\mathbf{B}^\natural = \mathbf{I}_{p\times p}$; **Block-diagonal**: $\mathbf{B}^\natural = \mathrm{diag}\{1,\cdots,1,0.5,\cdots,0.5\}$.

Additional experiments are available in Table 3, from which we conclude the solution (17) can predict the phase transition point w.r.t. $\tau_h$ to a good extent.

#### D.1.2  IMPACT OF $n$ ON THE PHASE TRANSITION POINT

We study the impact of $n$ on $\tau_h$. The numerical experiment is shown in the top row of Figure 3, from which we can see the predicted phase transition $\tau_h$ matches to a good extent to the numerical

Table 3: Comparison between the predicted value of the phase transition threshold $\tau_h$ and its simulated value when $n = 500$. **P** denotes the **predicted value** while **S** denotes the **simulated value**. **S** corresponds to the $\tau_h$ when the error rate drops below $0.05$. We adopt a similar algorithm as in Algorithm 1. The only difference is we replace (2) in Line 5 with (15).

| $p$ | 75 | 100 | 125 | 150 | 175 | 200 |
|---|---|---|---|---|---|---|
| **P** | 0.82 | 0.73 | 0.68 | 0.62 | 0.56 | 0.52 |
| **S** | 0.77 | 0.75 | 0.7 | 0.66 | 0.61 | 0.57 |

experiments. Then, we fix the $p$ and study the impact of $n$ on $\tau_h$. We observe that the phase transition $\tau_h$ increases together with the sample number $n$, which is also captured by our formula in (17).

### D.1.3 LIMITS OF $\tau_h$

We consider the limiting behavior of $\tau_h$ when $\tau_p$ approaches 0, or equivalently, $p = o_{\mathrm{P}}(n)$. We can simplify $\mathbb{E}\Xi$ and $\mathrm{Var}\Xi$ in Theorem 3 as

$$\mathbb{E}\Xi \simeq n(1 - \tau_h) \left\|\!\left\|\mathbf{B}^\natural\right\|\!\right\|_{\mathrm{F}}^2,$$

$$\mathrm{Var}\Xi \simeq 3n^2(1 - \tau_h)^2 \left\|\!\left\|\mathbf{B}^{\natural\top}\mathbf{B}^\natural\right\|\!\right\|_{\mathrm{F}}^2.$$

We notice that the singularity point in (17) disappears. In other words, we can have the correct permutation matrix $\mathbf{\Pi}^\natural$ even when $h \approx n$. This is (partly) verified by Figure 4, from which we observe that the phase transition point w.r.t. $\tau_h$ approaches to one, or equivalently, $h$ approaches $n$, as $\tau_p$ decreases to zero.

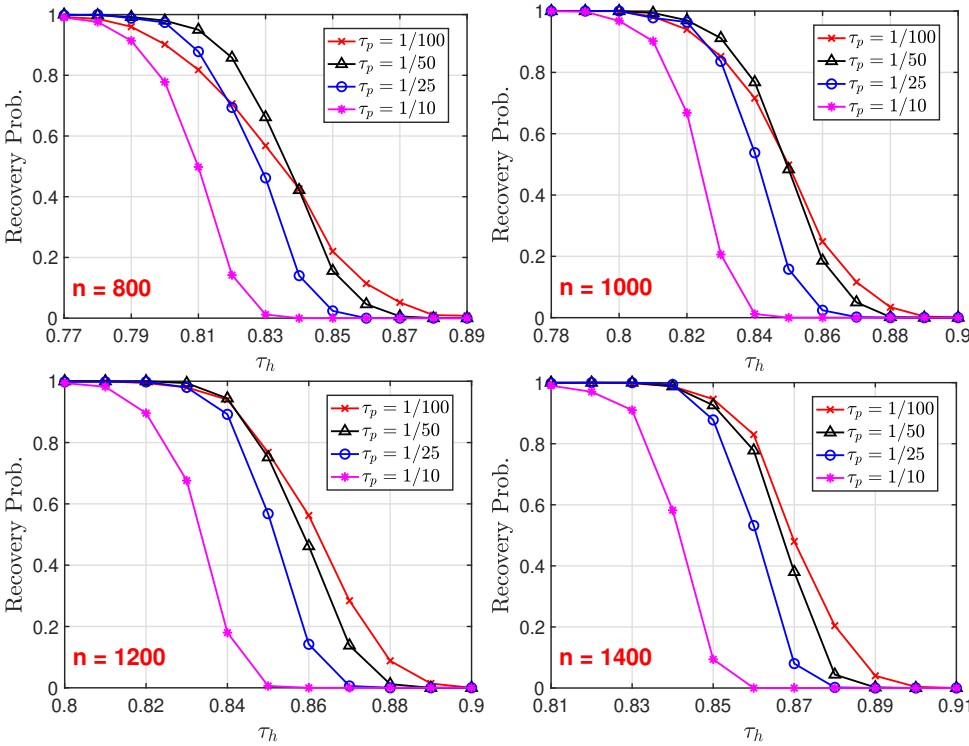

Figure 4: Plot of correct recovery rate w.r.t. $\tau_h$. We consider the noiseless scenario (i.e., $\mathsf{snr} = \infty$) and pick $n = \{800, 1000, 1200, 1400\}$.

## D.2 ANALYSIS OF NON-ORACLE CASE: PROOF OF THEOREM 3

This subsection presents the computational details of Theorem 3. To begin with, we decompose the random variable $\Xi$ as

$$\Xi = \Xi_1 + \sigma\left(\Xi_2 + \Xi_3\right) + \sigma^2 \Xi_4, \tag{21}$$

where $\Xi_i$ $(1 \leq i \leq 4)$ are respectively defined as

$$\Xi_1 \triangleq \mathbf{X}_{\pi^\natural(i)}^\top \mathbf{B}^\natural \mathbf{B}^{\natural\top} \mathbf{X}^\top \mathbf{\Pi}^{\natural\top} \mathbf{X}(\mathbf{X}_{\pi^\natural(i)} - \mathbf{X}_j),$$

$$\Xi_2 \triangleq \mathbf{X}_{\pi^\natural(i)}^\top \mathbf{B}^\natural \mathbf{W}^\top \mathbf{X}(\mathbf{X}_{\pi^\natural(i)} - \mathbf{X}_j),$$

$$\Xi_3 \triangleq \mathbf{W}_i^\top \mathbf{B}^{\natural\top} \mathbf{X}^\top \mathbf{\Pi}^{\natural\top} \mathbf{X}(\mathbf{X}_{\pi^\natural(i)} - \mathbf{X}_j),$$

$$\Xi_4 \triangleq \mathbf{W}_i^\top \mathbf{W}^\top \mathbf{X}(\mathbf{X}_{\pi^\natural(i)} - \mathbf{X}_j).$$

Unlike the oracle case, obtaining a closed-form expression of $\mathbb{E}e^{-\theta\Xi}$ would be too difficult. Hence, we adopt the Gaussian approximation method as presented in Section 3.3. The task then transforms to computing the expectation and variance of $\Xi$ with the computation details as follows.

### D.2.1 NOTATIONS

Note that our analysis can involve the terms containing $(\mathbf{X}_{\pi^\natural}(i) - \mathbf{X}_j)$ and $\mathbf{X}^\top \mathbf{\Pi}^\natural \mathbf{X}$ simultaneously. To decouple the dependence between $(\mathbf{X}_{\pi^\natural}(i) - \mathbf{X}_j)$ and $\mathbf{X}^\top \mathbf{\Pi}^\natural \mathbf{X}$, we first rewrite the matrix $\mathbf{X}^\top \mathbf{\Pi}^\natural \mathbf{X}$ as the sum $\sum_\ell \mathbf{X}_\ell \mathbf{X}_{\pi^\natural(\ell)}^\top$ and then collect all terms $\mathbf{X}_\ell \mathbf{X}_{\pi^\natural(\ell)}^\top$ independent of $\mathbf{X}_{\pi^\natural(i)}$ and $\mathbf{X}_j$ in the matrix $\mathbf{\Sigma}$, which is written as

$$\mathbf{\Sigma} \triangleq \sum_{\ell, \pi^\natural(\ell) \neq \pi^\natural(i), j} \mathbf{X}_\ell \mathbf{X}_{\pi^\natural(\ell)}^\top. \tag{22}$$

The rest terms are then put in the matrix $\mathbf{\Delta}$ such that $\mathbf{X}^\top \mathbf{\Pi}^\natural \mathbf{X} = \mathbf{\Sigma} + \mathbf{\Delta}$. Note that the expression of $\mathbf{\Delta}$ varies under different cases such that

- **Case $(s, s)$:** $i = \pi^\natural(i)$ **and** $j = \pi^\natural(j)$. We have

$$\mathbf{\Delta} = \mathbf{\Delta}^{(s,s)} = \mathbf{X}_i \mathbf{X}_i^\top + \mathbf{X}_j \mathbf{X}_j^\top. \tag{23}$$

- **Case $(s, d)$:** $i = \pi^\natural(i)$ **and** $j \neq \pi^\natural(j)$. We have

$$\mathbf{\Delta} = \mathbf{\Delta}^{(s,d)} = \mathbf{X}_i \mathbf{X}_i^\top + \mathbf{X}_j \mathbf{X}_{\pi^\natural(j)}^\top + \mathbf{X}_{\pi^{\natural-1}(j)} \mathbf{X}_j^\top. \tag{24}$$

- **Case $(d, s)$:** $i \neq \pi^\natural(i)$ **and** $j = \pi^\natural(j)$. We have

$$\mathbf{\Delta} = \mathbf{\Delta}^{(d,s)} = \mathbf{X}_i \mathbf{X}_{\pi^\natural(i)}^\top + \mathbf{X}_{\pi^\natural(i)} \mathbf{X}_{\pi^{\natural 2}(i)}^\top + \mathbf{X}_j \mathbf{X}_j^\top. \tag{25}$$

- **Case $(d, d)$:** $i \neq \pi^\natural(i)$ **and** $j \neq \pi^\natural(j)$. We have

$$\mathbf{\Delta} = \mathbf{\Delta}^{(d,d)} = \mathbf{X}_i \mathbf{X}_{\pi^\natural(i)}^\top + \mathbf{X}_{\pi^\natural(i)} \mathbf{X}_{\pi^{\natural 2}(i)}^\top + \mathbf{X}_j \mathbf{X}_{\pi^\natural(j)}^\top + \mathbf{X}_{\pi^{\natural-1}(j)} \mathbf{X}_j^\top. \tag{26}$$

In addition, we define the matrix $\mathbf{M}$ as $\mathbf{B}^\natural \mathbf{B}^{\natural\top}$, and define the index sets $\mathcal{S}, \mathcal{D},$ and $\mathcal{D}_{\text{pair}}$ as

$$\mathcal{S} \triangleq \left\{\ell \mid \ell \neq i \text{ or } j, \ell = \pi^\natural(\ell)\right\}, \tag{27}$$

$$\mathcal{D} \triangleq \left\{\ell \mid \ell, \pi^\natural(\ell) \neq i \text{ or } j, \ell \neq \pi^\natural(\ell)\right\}, \tag{28}$$

$$\mathcal{D}_{\text{pair}} \triangleq \left\{(\ell_1, \ell_2) : \ell_1 = \pi^\natural(\ell_2), \ell_2 = \pi^\natural(\ell1), \ell_1, \ell_2 \in \mathcal{D}\right\}, \tag{29}$$

respectively.

### D.2.2 MAIN COMPUTATION

In this case, we can write $\Xi$ as

$$\Xi = \underbrace{\mathbf{X}_{\pi^\natural(i)}^\top \mathbf{B}^\natural \mathbf{B}^{\natural\top} \mathbf{X}^\top \mathbf{\Pi}^{\natural\top} \mathbf{X} \left[ \mathbf{X}_{\pi^\natural(i)} - \mathbf{X}_j \right]}_{\triangleq \Xi_1} + \sigma \underbrace{\mathbf{X}_{\pi^\natural(i)}^\top \mathbf{B}^\natural \mathbf{W}^\top \mathbf{X} \left[ \mathbf{X}_{\pi^\natural(i)} - \mathbf{X}_j \right]}_{\triangleq \Xi_2}$$
$$+ \sigma \underbrace{\mathbf{W}_i^\top \mathbf{B}^{\natural\top} \mathbf{X}^\top \mathbf{\Pi}^{\natural\top} \mathbf{X} \left[ \mathbf{X}_{\pi^\natural(i)} - \mathbf{X}_j \right]}_{\triangleq \Xi_3} + \sigma^2 \underbrace{\mathbf{W}_i^\top \mathbf{W}^\top \mathbf{X} \left[ \mathbf{X}_{\pi^\natural(i)} - \mathbf{X}_j \right]}_{\triangleq \Xi_4}.$$

The following context separately computes its expectation $\mathbb{E}\Xi$ and its variance $\mathrm{Var}\Xi$.

**Expectation.** We can easily verify that both $\mathbb{E}\Xi_2$ and $\mathbb{E}\Xi_3$ are zero. Then our goal turns to calculating the expectation of $\mathbb{E}\Xi_1$ and $\mathbb{E}\Xi_4$. First, we have

$$\mathbb{E}\Xi_1 = \mathbb{E} \sum_{\ell = \pi^\natural(\ell)} \mathbf{X}_{\pi^\natural(i)}^\top \mathbf{M} \mathbf{X}_\ell \mathbf{X}_\ell^\top \mathbf{X}_{\pi^\natural(i)} - \mathbb{E} \sum_\ell \mathbf{X}_{\pi^\natural(i)}^\top \mathbf{M} \mathbf{X}_{\pi^\natural(\ell)} \mathbf{X}_\ell^\top \mathbf{X}_j.$$

With Lemma 16 and Lemma 17, we conclude

$$\mathbb{E}\Xi_1 = (n - h)\,\mathrm{Tr}(\mathbf{M}) + (p + 1)\mathbb{E}\mathbb{1}_{i = \pi^\natural(i)}\,\mathrm{Tr}(\mathbf{M}) - \left( p\mathbb{E}\mathbb{1}_{i = j} + \mathbb{E}\mathbb{1}_{j = \pi^{\natural 2}(i)} \right)\mathrm{Tr}(\mathbf{M})$$
$$= (n + p - h - hp/n)\,[1 + o(1)]\,\mathrm{Tr}(\mathbf{M}). \tag{30}$$

Meanwhile, we have

$$\mathbb{E}\Xi_4 = \mathbb{E} \left[ \mathbf{W}_i^\top \mathbf{W}_1 \;\cdots\; \mathbf{W}_i^\top \mathbf{W}_i \;\cdots\; \mathbf{W}_i^\top \mathbf{W}_n \right] \mathbf{X} \left( \mathbf{X}_{\pi^\natural(i)} - \mathbf{X}_j \right)$$
$$= m\mathbb{E}\mathbf{X}_i^\top \left( \mathbf{X}_{\pi^\natural(i)} - \mathbf{X}_j \right) = mp \left( \mathbb{E}\mathbb{1}_{i = \pi^\natural(i)} - \mathbb{E}\mathbb{1}_{i = j} \right) = \frac{mp(n - h)\sigma^2}{n}\,[1 + o(1)]. \tag{31}$$

Combining (30) and (31) and neglecting the $o(1)$ terms yields

$$\mathbb{E}\Xi \approx (n + p)(1 - h/n)\,\left\|\mathbf{B}^\natural\right\|_{\mathrm{F}}^2 + \frac{mp(n - h)\sigma^2}{n}.$$

**Variance.** Then we study the variance of $\Xi$. With the relation $\mathrm{Var}(\Xi) = \mathbb{E}\Xi^2 - (\mathbb{E}\Xi)^2$, our goal reduces to computing $\mathbb{E}\Xi^2$, which can be written as

$$\mathbb{E}\Xi^2 = \mathbb{E}\Xi_1^2 + \sigma^2\mathbb{E}\Xi_2^2 + \sigma^2\mathbb{E}\Xi_3^2 + \sigma^4\mathbb{E}\Xi_4^2 + 2\sigma^2\mathbb{E}\Xi_1\Xi_4 + 2\sigma^2\mathbb{E}\Xi_2\Xi_3.$$

The following context separately computes each terms

$$\mathbb{E}\Xi_1^2 \approx (n - h)^2 \left( 1 + \frac{2p}{n} + \frac{p^2}{n(n - h)} \right)[\mathrm{Tr}(\mathbf{M})]^2$$
$$+ n^2 \left[ \frac{2p}{n} + 3\left(1 - \frac{h}{n}\right)^2 + \frac{6(n - h)^2 p}{n^3} + \frac{(3n - h)p^2}{n^3} \right]\mathrm{Tr}(\mathbf{MM}),$$

$$\mathbb{E}\Xi_2^2 \approx 2np\,(1 + p/n)\,\mathrm{Tr}(\mathbf{M}),$$

$$\mathbb{E}\Xi_3^2 \approx 2n^2 \left( \frac{p}{n} + \left(1 - \frac{h}{n}\right)^2 + \frac{p^2}{n^2} + \frac{4p(n - h)^2}{n^3} \right)\mathrm{Tr}(\mathbf{M}),$$

$$\mathbb{E}\Xi_4^2 \approx \frac{(n - h)m^2 p^2}{n},$$

$$\mathbb{E}\Xi_1\Xi_4 \approx \frac{mp(n - h)(n + p - h)}{n}\,\mathrm{Tr}(\mathbf{M}),$$

$$\mathbb{E}\Xi_2\Xi_3 \approx \frac{p(n - h)(n + p - h)}{n}\,\mathrm{Tr}(\mathbf{M}).$$

The detailed computation is attached as follows.

**Lemma 2.** *We have*

$$
\mathbb{E}\Xi_1^2 = (n-h)^2 \left(1 + \frac{2p}{n} + \frac{p^2}{n(n-h)} + o(1)\right) [\mathrm{Tr}(\mathbf{M})]^2
$$

$$
+ n^2 \left[\frac{2p}{n} + 3\left(1 - \frac{h}{n}\right)^2 + \frac{6(n-h)^2 p}{n^3} + \frac{(3n-h)p^2}{n^3} + o(1)\right] \mathrm{Tr}(\mathbf{MM}),
$$

*where $\Xi_1$ is defined in* (21).

*Proof.* We begin the proof by decomposing $\Xi_1^2$ as

$$
\mathbb{E}\Xi_1^2 = \mathbb{E} \underbrace{\left(\mathbf{X}_{\pi^\natural(i)} - \mathbf{X}_j\right)^\top \mathbf{\Sigma M X}_{\pi^\natural(i)} \mathbf{X}_{\pi^\natural(i)}^\top \mathbf{M \Sigma}^\top \left(\mathbf{X}_{\pi^\natural(i)} - \mathbf{X}_j\right)}_{\Lambda_1}
$$

$$
+ 2\mathbb{E} \underbrace{\left(\mathbf{X}_{\pi^\natural(i)} - \mathbf{X}_j\right)^\top \mathbf{\Sigma M X}_{\pi^\natural(i)} \mathbf{X}_{\pi^\natural(i)}^\top \mathbf{M \Delta}^\top \left(\mathbf{X}_{\pi^\natural(i)} - \mathbf{X}_j\right)}_{\Lambda_2}
$$

$$
+ \mathbb{E} \underbrace{\left(\mathbf{X}_{\pi^\natural(i)} - \mathbf{X}_j\right)^\top \mathbf{\Delta M X}_{\pi^\natural(i)} \mathbf{X}_{\pi^\natural(i)}^\top \mathbf{M \Delta}^\top \left(\mathbf{X}_{\pi^\natural(i)} - \mathbf{X}_j\right)}_{\Lambda_3},
$$

and separately bound each term as in Lemma 3, Lemma 4, and Lemma 5.

□

**Lemma 3.** *We have*

$$
\mathbb{E} \left(\mathbf{X}_{\pi^\natural(i)} - \mathbf{X}_j\right)^\top \mathbf{\Sigma M X}_{\pi^\natural(i)} \mathbf{X}_{\pi^\natural(i)}^\top \mathbf{M \Sigma}^\top \left(\mathbf{X}_{\pi^\natural(i)} - \mathbf{X}_j\right)
$$

$$
= (n-h)^2 (1 + o(1)) [\mathrm{Tr}(\mathbf{M})]^2 + n^2 \left[\frac{2p}{n} + 3\left(1 - \frac{h}{n}\right)^2 + o(1)\right] \mathrm{Tr}(\mathbf{MM}).
$$

*Proof.* Due to the independence among different rows of the sensing matrix $\mathbf{X}$, we condition on $\mathbf{\Sigma}$ and take expectation w.r.t. $\mathbf{X}_{\pi^\natural(i)}$ and $\mathbf{X}_j$, which leads to

$$
\mathbb{E}\Lambda_1 = \mathbb{E} \underbrace{\mathbf{X}_{\pi^\natural(i)}^\top \mathbf{\Sigma M X}_{\pi^\natural(i)} \mathbf{X}_{\pi^\natural(i)}^\top \mathbf{M \Sigma}^\top \mathbf{X}_{\pi^\natural(i)}}_{\Lambda_{1,1}} + \mathbb{E} \underbrace{\mathbf{X}_j^\top \mathbf{\Sigma M X}_{\pi^\natural(i)} \mathbf{X}_{\pi^\natural(i)}^\top \mathbf{M \Sigma}^\top \mathbf{X}_j}_{\Lambda_{1,2}}.
$$

For $\mathbb{E}\Lambda_{1,1}$, we obtain

$$
\mathbb{E}\Lambda_{1,1} \overset{\text{①}}{=} \mathbb{E}\left[\mathrm{Tr}(\mathbf{\Sigma M}) \mathrm{Tr}(\mathbf{\Sigma M})\right] + \mathbb{E}\,\mathrm{Tr}\left(\mathbf{\Sigma M M}^\top \mathbf{\Sigma}^\top\right) + \mathbb{E}\,\mathrm{Tr}(\mathbf{\Sigma M \Sigma M})
$$

$$
\overset{\text{②}}{=} (n-h)^2 [1 + o(1)] [\mathrm{Tr}(\mathbf{M})]^2 + n^2 \left[\frac{p}{n} + 2\left(1 - \frac{h}{n}\right)^2 + o(1)\right] \mathrm{Tr}(\mathbf{MM}),
$$

where ① is due to (65), and ② is due to Lemma 13, Lemma 14, and Lemma 15. As for $\mathbb{E}\Lambda_{1,2}$, we have

$$
\mathbb{E}\Lambda_{1,2} = \mathbb{E}\,\mathrm{Tr}\left(\mathbf{\Sigma M M}^\top \mathbf{\Sigma}^\top\right) = n^2 \left[\frac{p}{n} + \left(1 - \frac{h}{n}\right)^2 + o(1)\right] \mathrm{Tr}\left(\mathbf{M}^\top \mathbf{M}\right),
$$

and hence complete the proof. □

**Lemma 4.** *We have*

$$
\mathbb{E} \left(\mathbf{X}_{\pi^\natural(i)} - \mathbf{X}_j\right)^\top \mathbf{\Sigma M X}_{\pi^\natural(i)} \mathbf{X}_{\pi^\natural(i)}^\top \mathbf{M \Delta}^\top \left(\mathbf{X}_{\pi^\natural(i)} - \mathbf{X}_j\right) \approx \frac{(n-h)^2 p}{n} \left[(\mathrm{Tr}(\mathbf{M}))^2 + 3\,\mathrm{Tr}(\mathbf{MM})\right].
$$

*Proof.* Similar as above, we first expand $\Lambda_2$ as

$$\mathbb{E}\Lambda_2 = (n-h)\mathbb{E}\underbrace{\mathbf{X}_{\pi^\natural(i)}^\top\mathbf{M}\mathbf{X}_{\pi^\natural(i)}\mathbf{X}_{\pi^\natural(i)}^\top\mathbf{M}\boldsymbol{\Delta}^\top\mathbf{X}_{\pi^\natural(i)}}_{\Lambda_{2,1}} + (n-h)\mathbb{E}\underbrace{\mathbf{X}_j^\top\mathbf{M}\mathbf{X}_{\pi^\natural(i)}\mathbf{X}_{\pi^\natural(i)}^\top\mathbf{M}\boldsymbol{\Delta}^\top\mathbf{X}_j}_{\Lambda_{2,2}}$$

$$- (n-h)\mathbb{E}\underbrace{\mathbf{X}_{\pi^\natural(i)}^\top\mathbf{M}\mathbf{X}_{\pi^\natural(i)}\mathbf{X}_{\pi^\natural(i)}^\top\mathbf{M}\boldsymbol{\Delta}^\top\mathbf{X}_j}_{\Lambda_{2,3}} - (n-h)\mathbb{E}\underbrace{\mathbf{X}_j^\top\mathbf{M}\mathbf{X}_{\pi^\natural(i)}\mathbf{X}_{\pi^\natural(i)}^\top\mathbf{M}\boldsymbol{\Delta}^\top\mathbf{X}_{\pi^\natural(i)}}_{\Lambda_{2,4}}.$$

**Case $(s,s)$: $i = \pi^\natural(i)$ and $j = \pi^\natural(j)$.** We first compute $\Lambda_{2,1}$ as

$$\mathbb{E}\Lambda_{2,1} = \underbrace{\mathbb{E}\mathbf{X}_i^\top\mathbf{M}\mathbf{X}_i\mathbf{X}_i^\top\mathbf{M}\mathbf{X}_i\mathbf{X}_i^\top\mathbf{X}_i}_{\mathbb{E}\|\mathbf{X}_i\|_2^2(\mathbf{X}_i^\top\mathbf{M}\mathbf{X}_i)^2} + \underbrace{\mathbb{E}\mathbf{X}_i^\top\mathbf{M}\mathbf{X}_i\mathbf{X}_i^\top\mathbf{M}\mathbf{X}_j\mathbf{X}_j^\top\mathbf{X}_i}_{\mathbb{E}(\mathbf{X}_i^\top\mathbf{M}\mathbf{X}_i)^2}$$

$$= (p+5)\left[(\mathrm{Tr}(\mathbf{M}))^2 + \mathrm{Tr}(\mathbf{M}\mathbf{M}) + \mathrm{Tr}\left(\mathbf{M}^\top\mathbf{M}\right)\right].$$

We consider $\Lambda_{2,2}$ as

$$\mathbb{E}\Lambda_{2,2} = \underbrace{\mathbb{E}\left(\mathbf{X}_i^\top\mathbf{M}\mathbf{X}_i\mathbf{X}_i^\top\mathbf{M}\mathbf{X}_i\right)}_{\mathbb{E}(\mathbf{X}_i^\top\mathbf{M}\mathbf{X}_i)^2} + \underbrace{\mathbb{E}\left(\mathbf{X}_j^\top\mathbf{M}\mathbf{M}\mathbf{X}_j\mathbf{X}_j^\top\mathbf{X}_j\right)}_{\mathbb{E}\|\mathbf{X}_j\|_2^2\mathbf{X}_j^\top\mathbf{M}\mathbf{M}\mathbf{X}_j}$$

$$= (\mathrm{Tr}(\mathbf{M}))^2 + \mathrm{Tr}(\mathbf{M}\mathbf{M}) + \mathrm{Tr}\left(\mathbf{M}^\top\mathbf{M}\right) + (p+2)\,\mathrm{Tr}\left(\mathbf{M}\mathbf{M}\right).$$

As for $\Lambda_{2,3}$ and $\Lambda_{2,4}$, we can verify that they are both zero, which gives

$$\mathbb{E}\Lambda_2 = (n-h)p\,[1+o(1)]\,[\mathrm{Tr}(\mathbf{M})]^2 + 3(n-h)p\,[1+o(1)]\,\mathrm{Tr}(\mathbf{M}\mathbf{M}). \tag{32}$$

**Case $(s,d)$: $i = \pi^\natural(i)$ and $j \neq \pi^\natural(j)$.** We can compute $\Lambda_{2,1}$ as

$$\mathbb{E}\Lambda_{2,1} = \mathbb{E}\mathbf{X}_i^\top\mathbf{M}\mathbf{X}_i\mathbf{X}_i^\top\mathbf{M}\mathbf{X}_i\mathbf{X}_i^\top\mathbf{X}_i + \underbrace{\mathbb{E}\mathbf{X}_{\pi^\natural(i)}^\top\mathbf{M}\mathbf{X}_{\pi^\natural(i)}\mathbf{X}_{\pi^\natural(i)}^\top\mathbf{M}\mathbf{X}_{\pi^\natural(j)}\mathbf{X}_j^\top\mathbf{X}_{\pi^\natural(i)}}_{0}$$

$$+ \underbrace{\mathbb{E}\mathbf{X}_{\pi^\natural(i)}^\top\mathbf{M}\mathbf{X}_{\pi^\natural(i)}\mathbf{X}_{\pi^\natural(i)}^\top\mathbf{M}\mathbf{X}_j\mathbf{X}_{\pi^{\natural-1}(j)}^\top\mathbf{X}_{\pi^\natural(i)}}_{0}$$

$$= (p+4)\left[(\mathrm{Tr}(\mathbf{M}))^2 + \mathrm{Tr}(\mathbf{M}\mathbf{M}) + \mathrm{Tr}\left(\mathbf{M}^\top\mathbf{M}\right)\right].$$

We consider $\Lambda_{2,2}$ as

$$\mathbb{E}\Lambda_{2,2} = \mathbb{E}\mathbf{X}_j^\top\mathbf{M}\mathbf{X}_i\mathbf{X}_i^\top\mathbf{M}\mathbf{X}_i\mathbf{X}_i^\top\mathbf{X}_j + \underbrace{\mathbb{E}\mathbf{X}_j^\top\mathbf{M}\mathbf{X}_{\pi^\natural(i)}\mathbf{X}_{\pi^\natural(i)}^\top\mathbf{M}\mathbf{X}_{\pi^\natural(j)}\mathbf{X}_j^\top\mathbf{X}_j}_{0}$$

$$+ \underbrace{\mathbb{E}\mathbf{X}_j^\top\mathbf{M}\mathbf{X}_{\pi^\natural(i)}\mathbf{X}_{\pi^\natural(i)}^\top\mathbf{M}\mathbf{X}_j\mathbf{X}_{\pi^{\natural-1}(j)}^\top\mathbf{X}_j}_{0}$$

$$= \mathbb{E}\left(\mathbf{X}_i^\top\mathbf{M}\mathbf{X}_i\right)^2 = (\mathrm{Tr}(\mathbf{M}))^2 + \mathrm{Tr}(\mathbf{M}\mathbf{M}) + \mathrm{Tr}\left(\mathbf{M}^\top\mathbf{M}\right).$$

Similarly, we can verify that both $\mathbb{E}\Lambda_{2,3}$ and $\mathbb{E}\Lambda_{2,4}$ are zero and hence have

$$\mathbb{E}\Lambda_2 = (n-h)p\,[1+o(1)]\,[\mathrm{Tr}(\mathbf{M})]^2 + 2(n-h)p\,[1+o(1)]\,\mathrm{Tr}(\mathbf{M}\mathbf{M}). \tag{33}$$

**Case $(d,s)$: $i \neq \pi^\natural(i)$ and $j = \pi^\natural(j)$.** We compute $\Lambda_{2,1}$ as

$$\mathbb{E}\Lambda_{2,1} = \underbrace{\mathbb{E}\mathbf{X}_{\pi^\natural(i)}^\top\mathbf{M}\mathbf{X}_{\pi^\natural(i)}\mathbf{X}_{\pi^\natural(i)}^\top\mathbf{M}\mathbf{X}_{\pi^\natural(i)}\mathbf{X}_i^\top\mathbf{X}_{\pi^\natural(i)}}_{0} + \underbrace{\mathbb{E}\mathbf{X}_{\pi^\natural(i)}^\top\mathbf{M}\mathbf{X}_{\pi^\natural(i)}\mathbf{X}_{\pi^\natural(i)}^\top\mathbf{M}\mathbf{X}_{\pi^{\natural 2}(i)}\mathbf{X}_{\pi^\natural(i)}^\top\mathbf{X}_{\pi^\natural(i)}}_{0}$$

$$+ \underbrace{\mathbb{E}\mathbf{X}_{\pi^\natural(i)}^\top\mathbf{M}\mathbf{X}_{\pi^\natural(i)}\mathbf{X}_{\pi^\natural(i)}^\top\mathbf{M}\mathbf{X}_j\mathbf{X}_j^\top\mathbf{X}_{\pi^\natural(i)}}_{\mathbb{E}\mathbf{X}_{\pi^\natural(i)}^\top\mathbf{M}\mathbf{X}_{\pi^\natural(i)}\mathbf{X}_{\pi^\natural(i)}^\top\mathbf{M}\mathbf{X}_{\pi^\natural(i)}} = (\mathrm{Tr}(\mathbf{M}))^2 + \mathrm{Tr}(\mathbf{M}\mathbf{M}) + \mathrm{Tr}\left(\mathbf{M}^\top\mathbf{M}\right).$$

We consider $\Lambda_{2,2}$ as

$$\mathbb{E}\Lambda_{2,2} = \underbrace{\mathbb{E}\mathbf{X}_j^\top \mathbf{M}\mathbf{X}_{\pi^\natural(i)}\mathbf{X}_{\pi^\natural(i)}^\top \mathbf{M}\mathbf{X}_{\pi^\natural(i)}\mathbf{X}_i^\top \mathbf{X}_j}_{0} + \underbrace{\mathbb{E}\mathbf{X}_j^\top \mathbf{M}\mathbf{X}_{\pi^\natural(i)}\mathbf{X}_{\pi^\natural(i)}^\top \mathbf{M}\mathbf{X}_{\pi^{\natural 2}(i)}\mathbf{X}_{\pi^\natural(i)}^\top \mathbf{X}_j}_{0}$$

$$+ \underbrace{\mathbb{E}\mathbf{X}_j^\top \mathbf{M}\mathbf{X}_{\pi^\natural(i)}\mathbf{X}_{\pi^\natural(i)}^\top \mathbf{M}\mathbf{X}_j\mathbf{X}_j^\top \mathbf{X}_j}_{\mathbb{E}\|\mathbf{X}_j\|_2^2 \mathbf{X}_j^\top \mathbf{M}\mathbf{M}\mathbf{X}_j} = (p+2)\operatorname{Tr}(\mathbf{M}\mathbf{M}).$$

As for $\mathbb{E}\Lambda_{2,3}$ and $\Lambda_{2,4}$, we can follow the same strategy and prove they are both zero, which yields

$$\mathbb{E}\Lambda_2 = (n-h)\left[\operatorname{Tr}(\mathbf{M})\right]^2 + (n-h)p\left[1 + o(1)\right]\operatorname{Tr}(\mathbf{M}\mathbf{M}). \tag{34}$$

**Case $(d,d)$: $i \neq \pi^\natural(i)$ and $j \neq \pi^\natural(j)$.** Contrary to the previous cases, we have $\mathbb{E}\Lambda_{2,1}$ and $\mathbb{E}\Lambda_{2,2}$ to be zero in this case rather than $\mathbb{E}\Lambda_{2,3}$ and $\mathbb{E}\Lambda_{2,4}$.

Hence our focus turns to the calculation of $\mathbb{E}\Lambda_{2,3}$ and that of $\mathbb{E}\Lambda_{2,4}$. For $\Lambda_{2,3}$, we have

$$\mathbb{E}\Lambda_{2,3} = \underbrace{\mathbb{E}\mathbf{X}_{\pi^\natural(i)}^\top \mathbf{M}\mathbf{X}_{\pi^\natural(i)}\mathbf{X}_{\pi^\natural(i)}^\top \mathbf{M}\mathbf{X}_{\pi^\natural(i)}\mathbf{X}_i^\top \mathbf{X}_j}_{p\mathbb{1}_{i=j}\mathbb{E}\mathbf{X}_{\pi^\natural(i)}^\top \mathbf{M}\mathbf{X}_{\pi^\natural(i)}\mathbf{X}_{\pi^\natural(i)}^\top \mathbf{M}\mathbf{X}_{\pi^\natural(i)}} + \underbrace{\mathbb{E}\mathbf{X}_{\pi^\natural(i)}^\top \mathbf{M}\mathbf{X}_{\pi^\natural(i)}\mathbf{X}_{\pi^\natural(i)}^\top \mathbf{M}\mathbf{X}_{\pi^{\natural 2}(i)}\mathbf{X}_{\pi^\natural(i)}^\top \mathbf{X}_j}_{\mathbb{1}_{j=\pi^{\natural 2}(i)}\mathbb{E}\mathbf{X}_{\pi^\natural(i)}^\top \mathbf{M}\mathbf{X}_{\pi^\natural(i)}\mathbf{X}_{\pi^\natural(i)}^\top \mathbf{M}\mathbf{X}_{\pi^\natural(i)}}$$

$$+ \underbrace{\mathbb{E}\mathbf{X}_{\pi^\natural(i)}^\top \mathbf{M}\mathbf{X}_{\pi^\natural(i)}\mathbf{X}_{\pi^\natural(i)}^\top \mathbf{M}\mathbf{X}_{\pi^\natural(j)}\mathbf{X}_j^\top \mathbf{X}_j}_{p\mathbb{1}_{i=j}\mathbb{E}\mathbf{X}_{\pi^\natural(i)}^\top \mathbf{M}\mathbf{X}_{\pi^\natural(i)}\mathbf{X}_{\pi^\natural(i)}^\top \mathbf{M}\mathbf{X}_{\pi^\natural(i)}} + \underbrace{\mathbb{E}\mathbf{X}_{\pi^\natural(i)}^\top \mathbf{M}\mathbf{X}_{\pi^\natural(i)}\mathbf{X}_{\pi^\natural(i)}^\top \mathbf{M}\mathbf{X}_j\mathbf{X}_{\pi^{\natural -1}(j)}^\top \mathbf{X}_j}_{\mathbb{1}_{j=\pi^{\natural 2}(i)}\mathbb{E}\mathbf{X}_{\pi^\natural(i)}^\top \mathbf{M}\mathbf{X}_{\pi^\natural(i)}\mathbf{X}_{\pi^\natural(i)}^\top \mathbf{M}\mathbf{X}_{\pi^\natural(i)}}$$

$$= 2\left(p\mathbb{1}_{i=j} + \mathbb{1}_{j=\pi^{\natural 2}(i)}\right)\left[\left(\operatorname{Tr}(\mathbf{M})\right)^2 + \operatorname{Tr}(\mathbf{M}\mathbf{M}) + \operatorname{Tr}\left(\mathbf{M}^\top \mathbf{M}\right)\right].$$

Then we turn to the calculation of $\mathbb{E}\Lambda_{2,4}$, which proceeds as

$$\mathbb{E}\Lambda_{2,4} = \underbrace{\mathbb{E}\mathbf{X}_j^\top \mathbf{M}\mathbf{X}_{\pi^\natural(i)}\mathbf{X}_{\pi^\natural(i)}^\top \mathbf{M}\mathbf{X}_{\pi^\natural(i)}\mathbf{X}_i^\top \mathbf{X}_{\pi^\natural(i)}}_{\mathbb{1}_{(i=j)}\mathbb{E}\mathbf{X}_{\pi^\natural(i)}^\top \mathbf{M}\mathbf{X}_{\pi^\natural(i)}\mathbf{X}_{\pi^\natural(i)}^\top \mathbf{M}\mathbf{X}_{\pi^\natural(i)}} + \underbrace{\mathbb{E}\mathbf{X}_j^\top \mathbf{M}\mathbf{X}_{\pi^\natural(i)}\mathbf{X}_{\pi^\natural(i)}^\top \mathbf{M}\mathbf{X}_{\pi^{\natural 2}(i)}\mathbf{X}_{\pi^\natural(i)}^\top \mathbf{X}_{\pi^\natural(i)}}_{\mathbb{1}_{j=\pi^{\natural 2}(i)}\mathbb{E}\left\|\mathbf{X}_{\pi^\natural(i)}\right\|_2^2 \mathbf{X}_{\pi^\natural(i)}^\top \mathbf{M}\mathbf{M}\mathbf{X}_{\pi^\natural(i)}}$$

$$+ \underbrace{\mathbb{E}\mathbf{X}_j^\top \mathbf{M}\mathbf{X}_{\pi^\natural(i)}\mathbf{X}_{\pi^\natural(i)}^\top \mathbf{M}\mathbf{X}_{\pi^\natural(j)}\mathbf{X}_j^\top \mathbf{X}_{\pi^\natural(i)}}_{\mathbb{1}_{(i=j)}\mathbb{E}\mathbf{X}_{\pi^\natural(i)}^\top \mathbf{M}\mathbf{X}_{\pi^\natural(i)}\mathbf{X}_{\pi^\natural(i)}^\top \mathbf{M}\mathbf{X}_{\pi^\natural(i)}} + \underbrace{\mathbb{E}\mathbf{X}_j^\top \mathbf{M}\mathbf{X}_{\pi^\natural(i)}\mathbf{X}_{\pi^\natural(i)}^\top \mathbf{M}\mathbf{X}_j\mathbf{X}_{\pi^{\natural -1}(j)}^\top \mathbf{X}_{\pi^\natural(i)}}_{\mathbb{1}_{j=\pi^{\natural 2}(i)}\mathbb{E}\left\|\mathbf{X}_{\pi^\natural(i)}\right\|_2^2 \mathbf{X}_{\pi^\natural(i)}^\top \mathbf{M}\mathbf{M}\mathbf{X}_{\pi^\natural(i)}}$$

$$= 2\mathbb{1}_{j=\pi^{\natural 2}(i)}(p+2)\operatorname{Tr}(\mathbf{M}\mathbf{M}) + 2\mathbb{1}(i=j)\left[\left(\operatorname{Tr}(\mathbf{M})\right)^2 + \operatorname{Tr}(\mathbf{M}\mathbf{M}) + \operatorname{Tr}\left(\mathbf{M}^\top \mathbf{M}\right)\right].$$

Then we conclude

$$\mathbb{E}\Lambda_2 = -2(n-h)\left[(p+1)\mathbb{1}_{i=j} + \mathbb{1}_{j=\pi^{\natural 2}(i)}\right]\left[\left(\operatorname{Tr}(\mathbf{M})\right)^2 + \operatorname{Tr}(\mathbf{M}\mathbf{M}) + \operatorname{Tr}\left(\mathbf{M}^\top \mathbf{M}\right)\right]$$

$$- 2(n-h)(p+2)\mathbb{1}_{j=\pi^{\natural 2}(i)}\operatorname{Tr}(\mathbf{M}\mathbf{M}). \tag{35}$$

The proof is thus completed by combining (32), (33), (34), and (35).

$\square$

**Lemma 5.** *We have*

$$\mathbb{E}\left(\mathbf{X}_{\pi^\natural(i)} - \mathbf{X}_j\right)^\top \mathbf{\Delta}\mathbf{M}\mathbf{X}_{\pi^\natural(i)}\mathbf{X}_{\pi^\natural(i)}^\top \mathbf{M}\mathbf{\Delta}^\top \left(\mathbf{X}_{\pi^\natural(i)} - \mathbf{X}_j\right)$$

$$= \left(1 - \frac{h}{n} + o(1)\right)p^2 \left[\operatorname{Tr}(\mathbf{M})\right]^2 + \left(3 - \frac{h}{n} + o(1)\right)p^2 \operatorname{Tr}(\mathbf{M}\mathbf{M}).$$

*Proof.* We begin the proof by expanding $\Lambda_3$ as

$$\mathbb{E}\Lambda_3 = \mathbb{E}\underbrace{\mathbf{X}_{\pi^\natural(i)}^\top \mathbf{\Delta}\mathbf{M}\mathbf{X}_{\pi^\natural(i)}\mathbf{X}_{\pi^\natural(i)}^\top \mathbf{M}\mathbf{\Delta}^\top \mathbf{X}_{\pi^\natural(i)}}_{\Lambda_{3,1}} + \mathbb{E}\underbrace{\mathbf{X}_j^\top \mathbf{\Delta}\mathbf{M}\mathbf{X}_{\pi^\natural(i)}\mathbf{X}_{\pi^\natural(i)}^\top \mathbf{M}\mathbf{\Delta}^\top \mathbf{X}_j}_{\Lambda_{3,2}}$$

$$- \mathbb{E}\,\underbrace{\mathbf{X}_{\pi^{\natural}(i)}^{\top}\boldsymbol{\Delta}\mathbf{M}\mathbf{X}_{\pi^{\natural}(i)}\mathbf{X}_{\pi^{\natural}(i)}^{\top}\mathbf{M}\boldsymbol{\Delta}^{\top}\mathbf{X}_{j}}_{\Lambda_{3,3}} - \mathbb{E}\,\underbrace{\mathbf{X}_{j}^{\top}\boldsymbol{\Delta}\mathbf{M}\mathbf{X}_{\pi^{\natural}(i)}\mathbf{X}_{\pi^{\natural}(i)}^{\top}\mathbf{M}\boldsymbol{\Delta}^{\top}\mathbf{X}_{\pi^{\natural}(i)}}_{\Lambda_{3,4}}.$$

**Case $(s,s)$: $i = \pi^{\natural}(i)$ and $j = \pi^{\natural}(j)$.** First we compute $\Lambda_{3,1}$ as

$$\mathbb{E}\Lambda_{3,1} = \underbrace{\mathbb{E}\mathbf{X}_{i}^{\top}\mathbf{X}_{i}\mathbf{X}_{i}^{\top}\mathbf{M}\mathbf{X}_{i}\mathbf{X}_{i}^{\top}\mathbf{M}\mathbf{X}_{i}\mathbf{X}_{i}^{\top}\mathbf{X}_{i}}_{\mathbb{E}\|\mathbf{X}_{i}\|_{2}^{4}(\mathbf{X}_{i}^{\top}\mathbf{M}\mathbf{X}_{i})^{2}} + \underbrace{\mathbb{E}\mathbf{X}_{i}^{\top}\mathbf{X}_{i}\mathbf{X}_{i}^{\top}\mathbf{M}\mathbf{X}_{i}\mathbf{X}_{i}^{\top}\mathbf{M}\mathbf{X}_{j}\mathbf{X}_{j}^{\top}\mathbf{X}_{i}}_{\mathbb{E}\|\mathbf{X}_{i}\|_{2}^{2}(\mathbf{X}_{i}^{\top}\mathbf{M}\mathbf{X}_{i})^{2}}$$

$$+ \underbrace{\mathbb{E}\mathbf{X}_{i}^{\top}\mathbf{X}_{j}\mathbf{X}_{j}^{\top}\mathbf{M}\mathbf{X}_{i}\mathbf{X}_{i}^{\top}\mathbf{M}\mathbf{X}_{i}\mathbf{X}_{i}^{\top}\mathbf{X}_{i}}_{\mathbb{E}\|\mathbf{X}_{i}\|_{2}^{2}(\mathbf{X}_{i}^{\top}\mathbf{M}\mathbf{X}_{i})^{2}} + \underbrace{\mathbb{E}\mathbf{X}_{i}^{\top}\mathbf{X}_{j}\mathbf{X}_{j}^{\top}\mathbf{M}\mathbf{X}_{i}\mathbf{X}_{i}^{\top}\mathbf{M}\mathbf{X}_{j}\mathbf{X}_{j}^{\top}\mathbf{X}_{i}}_{\mathbb{E}(\mathbf{X}_{j}^{\top}\mathbf{X}_{i})^{2}\mathbf{X}_{j}^{\top}\mathbf{M}\mathbf{X}_{i}\mathbf{X}_{i}^{\top}\mathbf{M}\mathbf{X}_{j}}$$

$$= (p+4)(p+8)\left[(\mathrm{Tr}(\mathbf{M}))^{2} + 2\,\mathrm{Tr}(\mathbf{M}\mathbf{M})\right] + 2\,(\mathrm{Tr}(\mathbf{M}))^{2} + (p+6)\,\mathrm{Tr}(\mathbf{M}\mathbf{M}).$$

Then, we consider $\Lambda_{3,2}$ as

$$\mathbb{E}\Lambda_{3,2} = \underbrace{\mathbb{E}\mathbf{X}_{j}^{\top}\mathbf{X}_{i}\mathbf{X}_{i}^{\top}\mathbf{M}\mathbf{X}_{i}\mathbf{X}_{i}^{\top}\mathbf{M}\mathbf{X}_{i}\mathbf{X}_{i}^{\top}\mathbf{X}_{j}}_{\mathbb{E}\|\mathbf{X}_{i}\|_{2}^{2}(\mathbf{X}_{i}^{\top}\mathbf{M}\mathbf{X}_{i})^{2}} + \underbrace{\mathbb{E}\mathbf{X}_{j}^{\top}\mathbf{X}_{i}\mathbf{X}_{i}^{\top}\mathbf{M}\mathbf{X}_{i}\mathbf{X}_{i}^{\top}\mathbf{M}\mathbf{X}_{j}\mathbf{X}_{j}^{\top}\mathbf{X}_{j}}_{(p+2)\mathbb{E}(\mathbf{X}_{i}^{\top}\mathbf{M}\mathbf{X}_{i})^{2}}$$

$$+ \underbrace{\mathbb{E}\mathbf{X}_{j}^{\top}\mathbf{X}_{j}\mathbf{X}_{j}^{\top}\mathbf{M}\mathbf{X}_{i}\mathbf{X}_{i}^{\top}\mathbf{M}\mathbf{X}_{i}\mathbf{X}_{i}^{\top}\mathbf{X}_{j}}_{(p+2)\mathbb{E}(\mathbf{X}_{i}^{\top}\mathbf{M}\mathbf{X}_{i})^{2}} + \underbrace{\mathbb{E}\mathbf{X}_{j}^{\top}\mathbf{X}_{j}\mathbf{X}_{j}^{\top}\mathbf{M}\mathbf{X}_{i}\mathbf{X}_{i}^{\top}\mathbf{M}\mathbf{X}_{j}\mathbf{X}_{j}^{\top}\mathbf{X}_{j}}_{\mathbb{E}\|\mathbf{X}_{j}\|_{2}^{4}\mathbf{X}_{j}^{\top}\mathbf{M}\mathbf{M}\mathbf{X}_{j}}$$

$$= (3p+8)\left[(\mathrm{Tr}(\mathbf{M}))^{2} + \mathrm{Tr}(\mathbf{M}\mathbf{M}) + \mathrm{Tr}\left(\mathbf{M}^{\top}\mathbf{M}\right)\right] + (p+2)(p+4)\,\mathrm{Tr}(\mathbf{M}\mathbf{M}).$$

In addition, we can verify that $\mathbb{E}\Lambda_{3,3}$ and $\mathbb{E}\Lambda_{3,4}$ are both zero. Hence we conclude

$$\mathbb{E}\Lambda_{3} = \left(p^{2} + 15p + 42\right)\left[\mathrm{Tr}(\mathbf{M})\right]^{2} + \left(3p^{2} + 37p + 94\right)\mathrm{Tr}(\mathbf{M}\mathbf{M}). \tag{36}$$

**Case $(s,d)$: $i = \pi^{\natural}(i)$ and $j \neq \pi^{\natural}(j)$.** We can compute $\Lambda_{3,1}$ as

$$\mathbb{E}\Lambda_{3,1} = \underbrace{\mathbb{E}\mathbf{X}_{i}^{\top}\mathbf{X}_{i}\mathbf{X}_{i}^{\top}\mathbf{M}\mathbf{X}_{i}\mathbf{X}_{i}^{\top}\mathbf{M}\mathbf{X}_{i}\mathbf{X}_{i}^{\top}\mathbf{X}_{i}}_{\mathbb{E}\|\mathbf{X}_{i}\|_{2}^{4}(\mathbf{X}_{i}^{\top}\mathbf{M}\mathbf{X}_{i})^{2}} + \underbrace{\mathbb{E}\mathbf{X}_{i}^{\top}\mathbf{X}_{i}\mathbf{X}_{i}^{\top}\mathbf{M}\mathbf{X}_{i}\mathbf{X}_{i}^{\top}\mathbf{M}\mathbf{X}_{\pi^{\natural}(j)}\mathbf{X}_{j}^{\top}\mathbf{X}_{i}}_{0}$$

$$+ \underbrace{\mathbb{E}\mathbf{X}_{i}^{\top}\mathbf{X}_{i}\mathbf{X}_{i}^{\top}\mathbf{M}\mathbf{X}_{i}\mathbf{X}_{i}^{\top}\mathbf{M}\mathbf{X}_{j}\mathbf{X}_{\pi^{\natural-1}(j)}^{\top}\mathbf{X}_{i}}_{0} + \underbrace{\mathbb{E}\mathbf{X}_{i}^{\top}\mathbf{X}_{j}\mathbf{X}_{\pi^{\natural}(j)}^{\top}\mathbf{M}\mathbf{X}_{i}\mathbf{X}_{i}^{\top}\mathbf{M}\mathbf{X}_{i}\mathbf{X}_{i}^{\top}\mathbf{X}_{i}}_{0}$$

$$+ \underbrace{\mathbb{E}\mathbf{X}_{i}^{\top}\mathbf{X}_{j}\mathbf{X}_{\pi^{\natural}(j)}^{\top}\mathbf{M}\mathbf{X}_{i}\mathbf{X}_{i}^{\top}\mathbf{M}\mathbf{X}_{\pi^{\natural}(j)}\mathbf{X}_{j}^{\top}\mathbf{X}_{i}}_{\mathbb{E}\|\mathbf{X}_{i}\|_{2}^{2}\mathbf{X}_{i}^{\top}\mathbf{M}\mathbf{M}\mathbf{X}_{i}} + \underbrace{\mathbb{E}\mathbf{X}_{i}^{\top}\mathbf{X}_{j}\mathbf{X}_{\pi^{\natural}(j)}^{\top}\mathbf{M}\mathbf{X}_{i}\mathbf{X}_{i}^{\top}\mathbf{M}\mathbf{X}_{j}\mathbf{X}_{\pi^{\natural-1}(j)}^{\top}\mathbf{X}_{i}}_{\mathbb{1}_{j=\pi^{\natural2}(j)}\mathbb{E}(\mathbf{X}_{i}^{\top}\mathbf{M}\mathbf{X}_{i})^{2}}$$

$$+ \underbrace{\mathbb{E}\mathbf{X}_{i}^{\top}\mathbf{X}_{\pi^{\natural-1}(j)}\mathbf{X}_{j}^{\top}\mathbf{M}\mathbf{X}_{i}\mathbf{X}_{i}^{\top}\mathbf{M}\mathbf{X}_{i}\mathbf{X}_{i}^{\top}\mathbf{X}_{i}}_{0} + \underbrace{\mathbb{E}\mathbf{X}_{i}^{\top}\mathbf{X}_{\pi^{\natural-1}(j)}\mathbf{X}_{j}^{\top}\mathbf{M}\mathbf{X}_{i}\mathbf{X}_{i}^{\top}\mathbf{M}\mathbf{X}_{\pi^{\natural}(j)}\mathbf{X}_{j}^{\top}\mathbf{X}_{i}}_{\mathbb{1}_{j=\pi^{\natural2}(j)}\mathbb{E}(\mathbf{X}_{i}^{\top}\mathbf{M}\mathbf{X}_{i})^{2}}$$

$$+ \underbrace{\mathbb{E}\mathbf{X}_{i}^{\top}\mathbf{X}_{\pi^{\natural-1}(j)}\mathbf{X}_{j}^{\top}\mathbf{M}\mathbf{X}_{i}\mathbf{X}_{i}^{\top}\mathbf{M}\mathbf{X}_{j}\mathbf{X}_{\pi^{\natural-1}(j)}^{\top}\mathbf{X}_{i}}_{\mathbb{E}\|\mathbf{X}_{i}\|_{2}^{2}\mathbf{X}_{i}^{\top}\mathbf{M}\mathbf{M}\mathbf{X}_{i}}$$

$$= \left(p^{2} + 10p + 24 + 2\,\mathbb{1}_{j=\pi^{\natural2}(j)}\right)\left[(\mathrm{Tr}(\mathbf{M}))^{2} + 2\,\mathrm{Tr}(\mathbf{M}\mathbf{M})\right] + 2(p+2)\,\mathrm{Tr}(\mathbf{M}\mathbf{M}).$$

We consider $\Lambda_{3,2}$ as

$$\mathbb{E}\Lambda_{3,2} = \underbrace{\mathbb{E}\mathbf{X}_{j}^{\top}\mathbf{X}_{i}\mathbf{X}_{i}^{\top}\mathbf{M}\mathbf{X}_{i}\mathbf{X}_{i}^{\top}\mathbf{M}\mathbf{X}_{i}\mathbf{X}_{i}^{\top}\mathbf{X}_{j}}_{\mathbb{E}\|\mathbf{X}_{i}\|_{2}^{2}(\mathbf{X}_{i}^{\top}\mathbf{M}\mathbf{X}_{i})^{2}} + \underbrace{\mathbb{E}\mathbf{X}_{j}^{\top}\mathbf{X}_{i}\mathbf{X}_{i}^{\top}\mathbf{M}\mathbf{X}_{i}\mathbf{X}_{i}^{\top}\mathbf{M}\mathbf{X}_{\pi^{\natural}(j)}\mathbf{X}_{j}^{\top}\mathbf{X}_{j}}_{0}$$

$$+ \underbrace{\mathbb{E}\mathbf{X}_{j}^{\top}\mathbf{X}_{i}\mathbf{X}_{i}^{\top}\mathbf{M}\mathbf{X}_{i}\mathbf{X}_{i}^{\top}\mathbf{M}\mathbf{X}_{j}\mathbf{X}_{\pi^{\natural-1}(j)}^{\top}\mathbf{X}_{j}}_{0}$$

$$+ \underbrace{\mathbb{E}\mathbf{X}_{j}^{\top}\mathbf{X}_{j}\mathbf{X}_{\pi^{\natural}(j)}^{\top}\mathbf{M}\mathbf{X}_{i}\mathbf{X}_{i}^{\top}\mathbf{M}\mathbf{X}_{i}\mathbf{X}_{i}^{\top}\mathbf{X}_{j}}_{0} + \underbrace{\mathbb{E}\mathbf{X}_{j}^{\top}\mathbf{X}_{j}\mathbf{X}_{\pi^{\natural}(j)}^{\top}\mathbf{M}\mathbf{X}_{i}\mathbf{X}_{i}^{\top}\mathbf{M}\mathbf{X}_{\pi^{\natural}(j)}\mathbf{X}_{j}^{\top}\mathbf{X}_{j}}_{\mathbb{E}\|\mathbf{X}_{i}\|_{2}^{4}\,\mathrm{Tr}(\mathbf{M}\mathbf{M})}$$

$$+ \underbrace{\mathbb{E}\mathbf{X}_j^\top \mathbf{X}_j \mathbf{X}_{\pi^\natural(j)}^\top \mathbf{M}\mathbf{X}_i \mathbf{X}_i^\top \mathbf{M}\mathbf{X}_j \mathbf{X}_{\pi^{\natural-1}(j)}^\top \mathbf{X}_j}_{\mathbb{1}_{j=\pi^{\natural 2}(j)}\mathbb{E}\|\mathbf{X}_j\|_2^2 \mathbf{X}_j^\top \mathbf{M}\mathbf{M}\mathbf{X}_j} + \underbrace{\mathbb{E}\mathbf{X}_j^\top \mathbf{X}_{\pi^{\natural-1}(j)} \mathbf{X}_j^\top \mathbf{M}\mathbf{X}_i \mathbf{X}_i^\top \mathbf{M}\mathbf{X}_i \mathbf{X}_i^\top \mathbf{X}_j}_{0}$$

$$+ \underbrace{\mathbb{E}\mathbf{X}_j^\top \mathbf{X}_{\pi^{\natural-1}(j)} \mathbf{X}_j^\top \mathbf{M}\mathbf{X}_i \mathbf{X}_i^\top \mathbf{M}\mathbf{X}_{\pi^\natural(j)} \mathbf{X}_j^\top \mathbf{X}_j}_{\mathbb{1}_{j=\pi^{\natural 2}(j)}\mathbb{E}\|\mathbf{X}_j\|_2^2 \mathbf{X}_j^\top \mathbf{M}\mathbf{M}\mathbf{X}_j} + \underbrace{\mathbb{E}\mathbf{X}_j^\top \mathbf{X}_{\pi^{\natural-1}(j)} \mathbf{X}_j^\top \mathbf{M}\mathbf{X}_i \mathbf{X}_i^\top \mathbf{M}\mathbf{X}_j \mathbf{X}_{\pi^{\natural-1}(j)}^\top \mathbf{X}_j}_{\mathbb{E}\|\mathbf{X}_j\|_2^2 \mathbf{X}_j^\top \mathbf{M}\mathbf{M}\mathbf{X}_j}$$

$$= (p+4)\left[(\mathrm{Tr}(\mathbf{M}))^2 + 2\,\mathrm{Tr}(\mathbf{M}\mathbf{M})\right] + (p+2)\left(p+1+2\mathbb{1}_{j=\pi^{\natural 2}(j)}\right)\mathrm{Tr}(\mathbf{M}\mathbf{M}).$$

As for $\Lambda_{3,3}$ and $\Lambda_{3,4}$, we can prove that they are both zero in this case. Hence we conclude

$$\mathbb{E}\Lambda_3 = \left(p^2 + 11p + 28 + 2\mathbb{1}_{j=\pi^{\natural 2}(j)}\right)(\mathrm{Tr}(\mathbf{M}))^2$$
$$+ \left[3p^2 + 27p + 62 + 2\,(p+4)\,\mathbb{1}_{j=\pi^{\natural 2}(j)}\right]\mathrm{Tr}(\mathbf{M}\mathbf{M}). \tag{37}$$

**Case $(d,s)$: $i \neq \pi^\natural(i)$ and $j = \pi^\natural(j)$.** We consider the term $\Lambda_{3,1}$ as

$$\mathbb{E}\Lambda_{3,1} = \underbrace{\mathbb{E}\mathbf{X}_{\pi^\natural(i)}^\top \mathbf{X}_i \mathbf{X}_{\pi^\natural(i)}^\top \mathbf{M}\mathbf{X}_{\pi^\natural(i)} \mathbf{X}_{\pi^\natural(i)}^\top \mathbf{M}\mathbf{X}_{\pi^\natural(i)} \mathbf{X}_i^\top \mathbf{X}_{\pi^\natural(i)}}_{\mathbb{E}\left\|\mathbf{X}_{\pi^\natural(i)}\right\|_{\mathrm{F}}^2 \left(\mathbf{X}_{\pi^\natural(i)}^\top \mathbf{M}\mathbf{X}_{\pi^\natural(i)}\right)^2}$$

$$+ \underbrace{\mathbb{E}\mathbf{X}_{\pi^\natural(i)}^\top \mathbf{X}_i \mathbf{X}_{\pi^\natural(i)}^\top \mathbf{M}\mathbf{X}_{\pi^\natural(i)} \mathbf{X}_{\pi^\natural(i)}^\top \mathbf{M}\mathbf{X}_{\pi^{\natural 2}(i)} \mathbf{X}_{\pi^\natural(i)}^\top \mathbf{X}_{\pi^\natural(i)}}_{\mathbb{1}_{i=\pi^{\natural 2}(i)}\mathbb{E}\left\|\mathbf{X}_{\pi^\natural(i)}\right\|_{\mathrm{F}}^2 \left(\mathbf{X}_{\pi^\natural(i)}^\top \mathbf{M}\mathbf{X}_{\pi^\natural(i)}\right)^2}$$

$$+ \underbrace{\mathbb{E}\mathbf{X}_{\pi^\natural(i)}^\top \mathbf{X}_i \mathbf{X}_{\pi^\natural(i)}^\top \mathbf{M}\mathbf{X}_{\pi^\natural(i)} \mathbf{X}_{\pi^\natural(i)}^\top \mathbf{M}\mathbf{X}_j \mathbf{X}_j^\top \mathbf{X}_{\pi^\natural(i)}}_{0}$$

$$+ \underbrace{\mathbb{E}\mathbf{X}_{\pi^\natural(i)}^\top \mathbf{X}_{\pi^\natural(i)} \mathbf{X}_{\pi^{\natural 2}(i)}^\top \mathbf{M}\mathbf{X}_{\pi^\natural(i)} \mathbf{X}_{\pi^\natural(i)}^\top \mathbf{M}\mathbf{X}_{\pi^\natural(i)} \mathbf{X}_i^\top \mathbf{X}_{\pi^\natural(i)}}_{\mathbb{1}_{i=\pi^{\natural 2}(i)}\mathbb{E}\left\|\mathbf{X}_{\pi^\natural(i)}\right\|_{\mathrm{F}}^2 \left(\mathbf{X}_{\pi^\natural(i)}^\top \mathbf{M}\mathbf{X}_{\pi^\natural(i)}\right)^2}$$

$$+ \underbrace{\mathbb{E}\mathbf{X}_{\pi^\natural(i)}^\top \mathbf{X}_{\pi^\natural(i)} \mathbf{X}_{\pi^{\natural 2}(i)}^\top \mathbf{M}\mathbf{X}_{\pi^\natural(i)} \mathbf{X}_{\pi^\natural(i)}^\top \mathbf{M}\mathbf{X}_{\pi^{\natural 2}(i)} \mathbf{X}_{\pi^\natural(i)}^\top \mathbf{X}_{\pi^\natural(i)}}_{\mathbb{E}\|\mathbf{X}_i\|_2^4 \mathbf{X}_i^\top \mathbf{M}\mathbf{M}\mathbf{X}_i}$$

$$+ \underbrace{\mathbb{E}\mathbf{X}_{\pi^\natural(i)}^\top \mathbf{X}_{\pi^\natural(i)} \mathbf{X}_{\pi^{\natural 2}(i)}^\top \mathbf{M}\mathbf{X}_{\pi^\natural(i)} \mathbf{X}_{\pi^\natural(i)}^\top \mathbf{M}\mathbf{X}_j \mathbf{X}_j^\top \mathbf{X}_{\pi^\natural(i)}}_{0}$$

$$+ \underbrace{\mathbb{E}\mathbf{X}_{\pi^\natural(i)}^\top \mathbf{X}_j \mathbf{X}_j^\top \mathbf{M}\mathbf{X}_{\pi^\natural(i)} \mathbf{X}_{\pi^\natural(i)}^\top \mathbf{M}\mathbf{X}_{\pi^\natural(i)} \mathbf{X}_i^\top \mathbf{X}_{\pi^\natural(i)}}_{0}$$

$$+ \underbrace{\mathbb{E}\mathbf{X}_{\pi^\natural(i)}^\top \mathbf{X}_j \mathbf{X}_j^\top \mathbf{M}\mathbf{X}_{\pi^\natural(i)} \mathbf{X}_{\pi^\natural(i)}^\top \mathbf{M}\mathbf{X}_{\pi^{\natural 2}(i)} \mathbf{X}_{\pi^\natural(i)}^\top \mathbf{X}_{\pi^\natural(i)}}_{0}$$

$$+ \underbrace{\mathbb{E}\mathbf{X}_{\pi^\natural(i)}^\top \mathbf{X}_j \mathbf{X}_j^\top \mathbf{M}\mathbf{X}_{\pi^\natural(i)} \mathbf{X}_{\pi^\natural(i)}^\top \mathbf{M}\mathbf{X}_j \mathbf{X}_j^\top \mathbf{X}_{\pi^\natural(i)}}_{\mathbb{E}\left(\mathbf{X}_i^\top \mathbf{X}_j\right)^2 \mathbf{X}_j^\top \mathbf{M}\mathbf{X}_i \mathbf{X}_i^\top \mathbf{M}\mathbf{X}_j}$$

$$= (p+6)(\mathrm{Tr}(\mathbf{M}))^2 + \left(p^2 + 9p + 22\right)\mathrm{Tr}(\mathbf{M}\mathbf{M}) + 2\mathbb{1}_{i=\pi^{\natural 2}(i)}\,(p+4)\left[(\mathrm{Tr}(\mathbf{M}))^2 + 2\,\mathrm{Tr}(\mathbf{M}\mathbf{M})\right].$$

We consider the term $\Lambda_{3,2}$ as

$$\mathbb{E}\Lambda_{3,2} = \underbrace{\mathbb{E}\mathbf{X}_j^\top \mathbf{X}_i \mathbf{X}_{\pi^\natural(i)}^\top \mathbf{M}\mathbf{X}_{\pi^\natural(i)} \mathbf{X}_{\pi^\natural(i)}^\top \mathbf{M}\mathbf{X}_{\pi^\natural(i)} \mathbf{X}_i^\top \mathbf{X}_j}_{p \times \mathbb{E}\left(\mathbf{X}_{\pi^\natural(i)}^\top \mathbf{M}\mathbf{X}_{\pi^\natural(i)}\right)^2}$$

$$+ \underbrace{\mathbb{E}\mathbf{X}_j^\top \mathbf{X}_i \mathbf{X}_{\pi^\natural(i)}^\top \mathbf{M}\mathbf{X}_{\pi^\natural(i)} \mathbf{X}_{\pi^\natural(i)}^\top \mathbf{M}\mathbf{X}_{\pi^{\natural 2}(i)} \mathbf{X}_{\pi^\natural(i)}^\top \mathbf{X}_j}_{\mathbb{1}_{i=\pi^{\natural 2}(i)}\mathbb{E}\left(\mathbf{X}_i^\top \mathbf{M}\mathbf{X}_i\right)^2}$$

$$+ \underbrace{\mathbb{E}\mathbf{X}_j^\top \mathbf{X}_i \mathbf{X}_{\pi^\natural(i)}^\top \mathbf{M}\mathbf{X}_{\pi^\natural(i)} \mathbf{X}_{\pi^\natural(i)}^\top \mathbf{M}\mathbf{X}_j \mathbf{X}_j^\top \mathbf{X}_j}_{0}$$

$$+ \underbrace{\mathbb{E}\mathbf{X}_j^\top \mathbf{X}_{\pi^\natural(i)} \mathbf{X}_{\pi^{\natural 2}(i)}^\top \mathbf{M}\mathbf{X}_{\pi^\natural(i)} \mathbf{X}_{\pi^\natural(i)}^\top \mathbf{M}\mathbf{X}_{\pi^\natural(i)} \mathbf{X}_i^\top \mathbf{X}_j}_{\mathbb{1}_{i=\pi^{\natural 2}(i)} \mathbb{E}(\mathbf{X}_i^\top \mathbf{M}\mathbf{X}_i)^2}$$

$$+ \underbrace{\mathbb{E}\mathbf{X}_j^\top \mathbf{X}_{\pi^\natural(i)} \mathbf{X}_{\pi^{\natural 2}(i)}^\top \mathbf{M}\mathbf{X}_{\pi^\natural(i)} \mathbf{X}_{\pi^\natural(i)}^\top \mathbf{M}\mathbf{X}_{\pi^{\natural 2}(i)} \mathbf{X}_{\pi^\natural(i)}^\top \mathbf{X}_j}_{\mathbb{E}\|\mathbf{X}_i\|_2^2 \mathbf{X}_i^\top \mathbf{M}\mathbf{M}\mathbf{X}_i}$$

$$+ \underbrace{\mathbb{E}\mathbf{X}_j^\top \mathbf{X}_{\pi^\natural(i)} \mathbf{X}_{\pi^{\natural 2}(i)}^\top \mathbf{M}\mathbf{X}_{\pi^\natural(i)} \mathbf{X}_{\pi^\natural(i)}^\top \mathbf{M}\mathbf{X}_j \mathbf{X}_j^\top \mathbf{X}_j}_{0}$$

$$+ \underbrace{\mathbb{E}\mathbf{X}_j^\top \mathbf{X}_j \mathbf{X}_j^\top \mathbf{M}\mathbf{X}_{\pi^\natural(i)} \mathbf{X}_{\pi^\natural(i)}^\top \mathbf{M}\mathbf{X}_{\pi^\natural(i)} \mathbf{X}_i^\top \mathbf{X}_j}_{0}$$

$$+ \underbrace{\mathbb{E}\mathbf{X}_j^\top \mathbf{X}_j \mathbf{X}_j^\top \mathbf{M}\mathbf{X}_{\pi^\natural(i)} \mathbf{X}_{\pi^\natural(i)}^\top \mathbf{M}\mathbf{X}_{\pi^{\natural 2}(i)} \mathbf{X}_{\pi^\natural(i)}^\top \mathbf{X}_j}_{0}$$

$$+ \underbrace{\mathbb{E}\mathbf{X}_j^\top \mathbf{X}_j \mathbf{X}_j^\top \mathbf{M}\mathbf{X}_{\pi^\natural(i)} \mathbf{X}_{\pi^\natural(i)}^\top \mathbf{M}\mathbf{X}_j \mathbf{X}_j^\top \mathbf{X}_j}_{\mathbb{E}\|\mathbf{X}_j\|_F^4 \times \mathbf{X}_j^\top \mathbf{M}\mathbf{M}\mathbf{X}_j}$$

$$= p\left(\mathrm{Tr}(\mathbf{M})\right)^2 + \left(p^2 + 9p + 10\right)\mathrm{Tr}(\mathbf{M}\mathbf{M}) + 2\mathbb{1}_{i=\pi^{\natural 2}(i)}\left[\left(\mathrm{Tr}(\mathbf{M})\right)^2 + 2\,\mathrm{Tr}(\mathbf{M}\mathbf{M})\right].$$

As for $\Lambda_{3,3}$ and $\Lambda_{3,4}$, easily we can verify they are both zero and hence

$$\mathbb{E}\Lambda_3 = 2\left(p+3\right)\left(\mathrm{Tr}(\mathbf{M})\right)^2 + 2\left(p^2 + 9p + 16\right)\mathrm{Tr}(\mathbf{M}\mathbf{M})$$
$$+ 2\mathbb{1}_{i=\pi^{\natural 2}(i)}\left(p+5\right)\left[\left(\mathrm{Tr}(\mathbf{M})\right)^2 + 2\,\mathrm{Tr}(\mathbf{M}\mathbf{M})\right]. \tag{38}$$

**Case** $(d,d)$**:** $i \neq \pi^\natural(i)$ **and** $j \neq \pi^\natural(j)$**.** First, We compute $\mathbb{E}\Lambda_{3,1}$ as

$$\mathbb{E}\Lambda_{3,1} = \underbrace{\mathbb{E}\mathbf{X}_{\pi^\natural(i)}^\top \mathbf{X}_i \mathbf{X}_{\pi^\natural(i)}^\top \mathbf{M}\mathbf{X}_{\pi^\natural(i)} \mathbf{X}_{\pi^\natural(i)}^\top \mathbf{M}\mathbf{X}_{\pi^\natural(i)} \mathbf{X}_i^\top \mathbf{X}_{\pi^\natural(i)}}_{\mathbb{E}\left\|\mathbf{X}_{\pi^\natural(i)}\right\|_F^2 \left(\mathbf{X}_{\pi^\natural(i)}^\top \mathbf{M}\mathbf{X}_{\pi^\natural(i)}\right)^2}$$

$$+ \underbrace{\mathbb{E}\mathbf{X}_{\pi^\natural(i)}^\top \mathbf{X}_i \mathbf{X}_{\pi^\natural(i)}^\top \mathbf{M}\mathbf{X}_{\pi^\natural(i)} \mathbf{X}_{\pi^\natural(i)}^\top \mathbf{M}\mathbf{X}_{\pi^{\natural 2}(i)} \mathbf{X}_{\pi^\natural(i)}^\top \mathbf{X}_{\pi^\natural(i)}}_{\mathbb{1}_{i=\pi^{\natural 2}(i)}\mathbb{E}\left\|\mathbf{X}_{\pi^\natural(i)}\right\|_F^2 \left(\mathbf{X}_{\pi^\natural(i)}^\top \mathbf{M}\mathbf{X}_{\pi^\natural(i)}\right)^2}$$

$$+ \underbrace{\mathbb{E}\mathbf{X}_{\pi^\natural(i)}^\top \mathbf{X}_i \mathbf{X}_{\pi^\natural(i)}^\top \mathbf{M}\mathbf{X}_{\pi^\natural(i)} \mathbf{X}_{\pi^\natural(i)}^\top \mathbf{M}\mathbf{X}_{\pi^\natural(j)} \mathbf{X}_j^\top \mathbf{X}_{\pi^\natural(i)}}_{\mathbb{1}_{i=j}\mathbb{E}\left\|\mathbf{X}_{\pi^\natural(i)}\right\|_F^2 \left(\mathbf{X}_{\pi^\natural(i)}^\top \mathbf{M}\mathbf{X}_{\pi^\natural(i)}\right)^2}$$

$$+ \underbrace{\mathbb{E}\mathbf{X}_{\pi^\natural(i)}^\top \mathbf{X}_i \mathbf{X}_{\pi^\natural(i)}^\top \mathbf{M}\mathbf{X}_{\pi^\natural(i)} \mathbf{X}_{\pi^\natural(i)}^\top \mathbf{M}\mathbf{X}_j \mathbf{X}_{\pi^{\natural-1}(j)}^\top \mathbf{X}_{\pi^\natural(i)}}_{\mathbb{1}_{j=i}\mathbb{1}_{j=\pi^{\natural 2}(i)}\mathbb{E}\left\|\mathbf{X}_{\pi^\natural(i)}\right\|_F^2 \left(\mathbf{X}_{\pi^\natural(i)}^\top \mathbf{M}\mathbf{X}_{\pi^\natural(i)}\right)^2}$$

$$+ \underbrace{\mathbb{E}\mathbf{X}_{\pi^\natural(i)}^\top \mathbf{X}_{\pi^\natural(i)} \mathbf{X}_{\pi^{\natural 2}(i)}^\top \mathbf{M}\mathbf{X}_{\pi^\natural(i)} \mathbf{X}_{\pi^\natural(i)}^\top \mathbf{M}\mathbf{X}_{\pi^\natural(i)} \mathbf{X}_i^\top \mathbf{X}_{\pi^\natural(i)}}_{\mathbb{1}_{i=\pi^{\natural 2}(i)}\mathbb{E}\left\|\mathbf{X}_{\pi^\natural(i)}\right\|_F^2 \left(\mathbf{X}_{\pi^\natural(i)}^\top \mathbf{M}\mathbf{X}_{\pi^\natural(i)}\right)^2}$$

$$+ \underbrace{\mathbb{E}\mathbf{X}_{\pi^\natural(i)}^\top \mathbf{X}_{\pi^\natural(i)} \mathbf{X}_{\pi^{\natural 2}(i)}^\top \mathbf{M}\mathbf{X}_{\pi^\natural(i)} \mathbf{X}_{\pi^\natural(i)}^\top \mathbf{M}\mathbf{X}_{\pi^{\natural 2}(i)} \mathbf{X}_{\pi^\natural(i)}^\top \mathbf{X}_{\pi^\natural(i)}}_{\mathbb{E}\left\|\mathbf{X}_{\pi^\natural(i)}\right\|_F^4 \left(\mathbf{X}_{\pi^\natural(i)}^\top \mathbf{M}\mathbf{M}\mathbf{X}_{\pi^\natural(i)}\right)}$$

$$+ \underbrace{\mathbb{E}\mathbf{X}_{\pi^\natural(i)}^\top \mathbf{X}_{\pi^\natural(i)} \mathbf{X}_{\pi^{\natural 2}(i)}^\top \mathbf{M}\mathbf{X}_{\pi^\natural(i)} \mathbf{X}_{\pi^\natural(i)}^\top \mathbf{M}\mathbf{X}_{\pi^\natural(j)} \mathbf{X}_j^\top \mathbf{X}_{\pi^\natural(i)}}_{\mathbb{1}_{j=i}\mathbb{1}_{j=\pi^{\natural 2}(i)}\mathbb{E}\left\|\mathbf{X}_{\pi^\natural(i)}\right\|_F^2 \left(\mathbf{X}_{\pi^\natural(i)}^\top \mathbf{M}\mathbf{X}_{\pi^\natural(i)}\right)^2}$$

$$+ \underbrace{\mathbb{E}\mathbf{X}_{\pi^\natural(i)}^\top \mathbf{X}_{\pi^\natural(i)} \mathbf{X}_{\pi^{\natural 2}(i)}^\top \mathbf{M}\mathbf{X}_{\pi^\natural(i)} \mathbf{X}_{\pi^\natural(i)}^\top \mathbf{M}\mathbf{X}_j \mathbf{X}_{\pi^{\natural-1}(j)}^\top \mathbf{X}_{\pi^\natural(i)}}_{\mathbb{1}_{j=\pi^{\natural 2}(i)}\mathbb{E}\left\|\mathbf{X}_{\pi^\natural(i)}\right\|_F^4 \left(\mathbf{X}_{\pi^\natural(i)}^\top \mathbf{M}\mathbf{M}\mathbf{X}_{\pi^\natural(i)}\right)}$$

$$+ \underbrace{\mathbb{E}\mathbf{X}_{\pi^\natural(i)}^\top \mathbf{X}_j \mathbf{X}_{\pi^\natural(j)}^\top \mathbf{M}\mathbf{X}_{\pi^\natural(i)} \mathbf{X}_{\pi^\natural(i)}^\top \mathbf{M}\mathbf{X}_{\pi^\natural(i)} \mathbf{X}_i^\top \mathbf{X}_{\pi^\natural(i)}}_{\mathbb{1}_{i=j}\mathbb{E}\left\|\mathbf{X}_{\pi^\natural(i)}\right\|_F^2 \left(\mathbf{X}_{\pi^\natural(i)}^\top \mathbf{M}\mathbf{X}_{\pi^\natural(i)}\right)^2}$$

$$+ \underbrace{\mathbb{E}\mathbf{X}_{\pi^\natural(i)}^\top \mathbf{X}_j \mathbf{X}_{\pi^\natural(j)}^\top \mathbf{M}\mathbf{X}_{\pi^\natural(i)} \mathbf{X}_{\pi^\natural(i)}^\top \mathbf{M}\mathbf{X}_{\pi^{\natural 2}(i)} \mathbf{X}_{\pi^\natural(i)}^\top \mathbf{X}_{\pi^\natural(i)}}_{\mathbb{1}_{j=i}\mathbb{1}_{j=\pi^{\natural 2}(i)}\mathbb{E}\left\|\mathbf{X}_{\pi^\natural(i)}\right\|_F^2 \left(\mathbf{X}_{\pi^\natural(i)}^\top \mathbf{M}\mathbf{X}_{\pi^\natural(i)}\right)^2}$$

$$+ \underbrace{\mathbb{E}\mathbf{X}_{\pi^\natural(i)}^\top \mathbf{X}_j \mathbf{X}_{\pi^\natural(j)}^\top \mathbf{M}\mathbf{X}_{\pi^\natural(i)} \mathbf{X}_{\pi^\natural(i)}^\top \mathbf{M}\mathbf{X}_{\pi^\natural(j)} \mathbf{X}_j^\top \mathbf{X}_{\pi^\natural(i)}}_{\mathbb{1}_{i=j}\mathbb{E}\left\|\mathbf{X}_{\pi^\natural(i)}\right\|_F^2 \left(\mathbf{X}_{\pi^\natural(i)}^\top \mathbf{M}\mathbf{X}_{\pi^\natural(i)}\right)^2 + \mathbb{1}_{i\neq j}\mathbb{E}\left\|\mathbf{X}_{\pi^\natural(i)}\right\|_F^2 \mathbf{X}_{\pi^\natural(i)}^\top \mathbf{M}\mathbf{M}\mathbf{X}_{\pi^\natural(i)}}$$

$$+ \underbrace{\mathbb{E}\mathbf{X}_{\pi^\natural(i)}^\top \mathbf{X}_j \mathbf{X}_{\pi^\natural(j)}^\top \mathbf{M}\mathbf{X}_{\pi^\natural(i)} \mathbf{X}_{\pi^\natural(i)}^\top \mathbf{M}\mathbf{X}_j \mathbf{X}_{\pi^{\natural-1}(j)}^\top \mathbf{X}_{\pi^\natural(i)}}_{\mathbb{1}_{j=\pi^{\natural 2}(j)}\left[\mathbb{1}_{i=j}\mathbb{E}\left\|\mathbf{X}_{\pi^\natural(i)}\right\|_F^2 \left(\mathbf{X}_{\pi^\natural(i)}^\top \mathbf{M}\mathbf{X}_{\pi^\natural(i)}\right)^2 + \mathbb{1}_{i\neq j}\mathbb{E}\left(\mathbf{X}_{\pi^\natural(i)}^\top \mathbf{M}\mathbf{X}_{\pi^\natural(i)}\right)^2\right]}$$

$$+ \underbrace{\mathbb{E}\mathbf{X}_{\pi^\natural(i)}^\top \mathbf{X}_{\pi^{\natural-1}(j)} \mathbf{X}_j^\top \mathbf{M}\mathbf{X}_{\pi^\natural(i)} \mathbf{X}_{\pi^\natural(i)}^\top \mathbf{M}\mathbf{X}_{\pi^\natural(i)} \mathbf{X}_i^\top \mathbf{X}_{\pi^\natural(i)}}_{\mathbb{1}_{i=j}\mathbb{1}_{j=\pi^{\natural 2}(i)}\mathbb{E}\left\|\mathbf{X}_{\pi^\natural(i)}\right\|_F^2 \left(\mathbf{X}_{\pi^\natural(i)}^\top \mathbf{M}\mathbf{X}_{\pi^\natural(i)}\right)^2}$$

$$+ \underbrace{\mathbb{E}\mathbf{X}_{\pi^\natural(i)}^\top \mathbf{X}_{\pi^{\natural-1}(j)} \mathbf{X}_j^\top \mathbf{M}\mathbf{X}_{\pi^\natural(i)} \mathbf{X}_{\pi^\natural(i)}^\top \mathbf{M}\mathbf{X}_{\pi^{\natural 2}(i)} \mathbf{X}_{\pi^\natural(i)}^\top \mathbf{X}_{\pi^\natural(i)}}_{\mathbb{1}_{j=\pi^{\natural 2}(i)}\mathbb{E}\left\|\mathbf{X}_{\pi^\natural(i)}\right\|_F^4 \left(\mathbf{X}_{\pi^\natural(i)}^\top \mathbf{M}\mathbf{M}\mathbf{X}_{\pi^\natural(i)}\right)}$$

$$+ \underbrace{\mathbb{E}\mathbf{X}_{\pi^\natural(i)}^\top \mathbf{X}_{\pi^{\natural-1}(j)} \mathbf{X}_j^\top \mathbf{M}\mathbf{X}_{\pi^\natural(i)} \mathbf{X}_{\pi^\natural(i)}^\top \mathbf{M}\mathbf{X}_{\pi^\natural(j)} \mathbf{X}_j^\top \mathbf{X}_{\pi^\natural(i)}}_{\mathbb{1}_{j=\pi^{\natural 2}(j)}\left[\mathbb{1}_{i=j}\mathbb{E}\left\|\mathbf{X}_{\pi^\natural(i)}\right\|_F^2 \left(\mathbf{X}_{\pi^\natural(i)}^\top \mathbf{M}\mathbf{X}_{\pi^\natural(i)}\right)^2 + \mathbb{1}_{i\neq j}\mathbb{E}\left(\mathbf{X}_{\pi^\natural(i)}^\top \mathbf{M}\mathbf{X}_{\pi^\natural(i)}\right)^2\right]}$$

$$+ \underbrace{\mathbb{E}\mathbf{X}_{\pi^\natural(i)}^\top \mathbf{X}_{\pi^{\natural-1}(j)} \mathbf{X}_j^\top \mathbf{M}\mathbf{X}_{\pi^\natural(i)} \mathbf{X}_{\pi^\natural(i)}^\top \mathbf{M}\mathbf{X}_j \mathbf{X}_{\pi^{\natural-1}(j)}^\top \mathbf{X}_{\pi^\natural(i)}}_{\mathbb{1}_{j=\pi^{\natural 2}(i)}\mathbb{E}\left\|\mathbf{X}_{\pi^\natural(i)}\right\|_F^4 \left(\mathbf{X}_{\pi^\natural(i)}^\top \mathbf{M}\mathbf{M}\mathbf{X}_{\pi^\natural(i)}\right) + \mathbb{1}_{j\neq\pi^{\natural 2}(i)}\mathbb{E}\left\|\mathbf{X}_{\pi^\natural(i)}\right\|_F^2 \left(\mathbf{X}_{\pi^\natural(i)}^\top \mathbf{M}\mathbf{M}\mathbf{X}_{\pi^\natural(i)}\right)}$$

$$= \left(1 + 2\mathbb{1}_{i=\pi^{\natural 2}(i)} + 3\mathbb{1}_{i=j} + 6\mathbb{1}_{i=j}\mathbb{1}_{j=\pi^{\natural 2}(i)}\right)(p+4)\left[(\operatorname{Tr}(\mathbf{M}))^2 + 2\operatorname{Tr}(\mathbf{M}\mathbf{M})\right]$$

$$+ \left(1 + 3\mathbb{1}_{j=\pi^{\natural 2}(i)}\right)(p+2)(p+4)\operatorname{Tr}(\mathbf{M}\mathbf{M})$$

$$+ 2\mathbb{1}_{j=\pi^{\natural 2}(j)}\mathbb{1}_{i\neq j}\left[(\operatorname{Tr}(\mathbf{M}))^2 + 2\operatorname{Tr}(\mathbf{M}\mathbf{M})\right]$$

$$+ \mathbb{1}_{j\neq\pi^{\natural 2}(i)}(p+2)\operatorname{Tr}(\mathbf{M}\mathbf{M}) + \mathbb{1}_{i\neq j}(p+2)\operatorname{Tr}(\mathbf{M}\mathbf{M}). \tag{39}$$

Then we calculate $\mathbb{E}\Lambda_{3,2}$ as

$$\mathbb{E}\Lambda_{3,2} = \underbrace{\mathbb{E}\mathbf{X}_j^\top \mathbf{X}_i \mathbf{X}_{\pi^\natural(i)}^\top \mathbf{M}\mathbf{X}_{\pi^\natural(i)} \mathbf{X}_{\pi^\natural(i)}^\top \mathbf{M}\mathbf{X}_{\pi^\natural(i)} \mathbf{X}_i^\top \mathbf{X}_j}_{\mathbb{1}_{i=j}\mathbb{E}\|\mathbf{X}_i\|_F^4 \left(\mathbf{X}_{\pi^\natural(i)}^\top \mathbf{M}\mathbf{X}_{\pi^\natural(i)}\right)^2 + p\mathbb{1}_{i\neq j}\mathbb{E}\left(\mathbf{X}_{\pi^\natural(i)}^\top \mathbf{M}\mathbf{X}_{\pi^\natural(i)}\right)^2}$$

$$+ \underbrace{\mathbb{E}\mathbf{X}_j^\top \mathbf{X}_i \mathbf{X}_{\pi^\natural(i)}^\top \mathbf{M}\mathbf{X}_{\pi^\natural(i)} \mathbf{X}_{\pi^\natural(i)}^\top \mathbf{M}\mathbf{X}_{\pi^{\natural 2}(i)} \mathbf{X}_{\pi^\natural(i)}^\top \mathbf{X}_j}_{\mathbb{1}_{i=\pi^{\natural 2}(i)}\left[\mathbb{1}_{i=j}\mathbb{E}\mathbf{X}_i^\top \mathbf{X}_i \mathbf{X}_{\pi^\natural(i)}^\top \mathbf{M}\mathbf{X}_{\pi^\natural(i)} \mathbf{X}_{\pi^\natural(i)}^\top \mathbf{M}\mathbf{X}_i \mathbf{X}_i^\top \mathbf{X}_{\pi^\natural(i)} + \mathbb{1}_{i\neq j}\mathbb{E}\left(\mathbf{X}_{\pi^\natural(i)}^\top \mathbf{M}\mathbf{X}_{\pi^\natural(i)}\right)^2\right]}$$

$$+ \underbrace{\mathbb{E}\mathbf{X}_j^\top \mathbf{X}_i \mathbf{X}_{\pi^\natural(i)}^\top \mathbf{M}\mathbf{X}_{\pi^\natural(i)} \mathbf{X}_{\pi^\natural(i)}^\top \mathbf{M}\mathbf{X}_{\pi^\natural(j)} \mathbf{X}_j^\top \mathbf{X}_j}_{\mathbb{1}_{i=j}\mathbb{E}\|\mathbf{X}_i\|_F^4 \left(\mathbf{X}_{\pi^\natural(i)}^\top \mathbf{M}\mathbf{X}_{\pi^\natural(i)}\right)^2}$$

$$+ \underbrace{\mathbb{E}\mathbf{X}_j^\top \mathbf{X}_i \mathbf{X}_{\pi^\natural(i)}^\top \mathbf{M}\mathbf{X}_{\pi^\natural(i)} \mathbf{X}_{\pi^\natural(i)}^\top \mathbf{M}\mathbf{X}_j \mathbf{X}_{\pi^{\natural-1}(j)}^\top \mathbf{X}_j}_{\mathbb{1}_{i=j}\mathbb{1}_{i=\pi^{\natural 2}(i)}\mathbb{E}\mathbf{X}_i^\top \mathbf{X}_i \mathbf{X}_{\pi^\natural(i)}^\top \mathbf{M}\mathbf{X}_{\pi^\natural(i)} \mathbf{X}_{\pi^\natural(i)}^\top \mathbf{M}\mathbf{X}_i \mathbf{X}_i^\top \mathbf{X}_{\pi^\natural(i)}}$$

$$+ \underbrace{\mathbb{E}\mathbf{X}_j^\top \mathbf{X}_{\pi^\natural(i)} \mathbf{X}_{\pi^{\natural 2}(i)}^\top \mathbf{M}\mathbf{X}_{\pi^\natural(i)} \mathbf{X}_{\pi^\natural(i)}^\top \mathbf{M}\mathbf{X}_{\pi^\natural(i)} \mathbf{X}_i^\top \mathbf{X}_j}_{\mathbb{1}_{i=\pi^{\natural 2}(i)}\left[\mathbb{1}_{i=j}\mathbb{E}\mathbf{X}_i^\top \mathbf{X}_i \mathbf{X}_{\pi^\natural(i)}^\top \mathbf{M}\mathbf{X}_{\pi^\natural(i)} \mathbf{X}_{\pi^\natural(i)}^\top \mathbf{M}\mathbf{X}_i \mathbf{X}_i^\top \mathbf{X}_{\pi^\natural(i)} + \mathbb{1}_{i\neq j}\mathbb{E}\left(\mathbf{X}_{\pi^\natural(i)}^\top \mathbf{M}\mathbf{X}_{\pi^\natural(i)}\right)^2\right]}$$

$$+ \underbrace{\mathbb{E}\mathbf{X}_j^\top \mathbf{X}_{\pi^\natural(i)} \mathbf{X}_{\pi^{\natural 2}(i)}^\top \mathbf{M} \mathbf{X}_{\pi^\natural(i)} \mathbf{X}_{\pi^\natural(i)}^\top \mathbf{M} \mathbf{X}_{\pi^{\natural 2}(i)} \mathbf{X}_{\pi^\natural(i)}^\top \mathbf{X}_j}_{\mathbb{1}_{j \neq \pi^{\natural 2}(i)} \mathbb{E}\left\|\mathbf{X}_{\pi^\natural(i)}\right\|_{\mathrm{F}}^2 \mathbf{X}_{\pi^\natural(i)}^\top \mathbf{M} \mathbf{M} \mathbf{X}_{\pi^\natural(i)} + \mathbb{1}_{j = \pi^{\natural 2}(i)} \mathbb{E}\left(\mathbf{X}_j^\top \mathbf{X}_{\pi^\natural(i)}\right)^2 \left(\mathbf{X}_{\pi^\natural(i)}^\top \mathbf{M} \mathbf{X}_j\right)^2}$$

$$+ \underbrace{\mathbb{E}\mathbf{X}_j^\top \mathbf{X}_{\pi^\natural(i)} \mathbf{X}_{\pi^{\natural 2}(i)}^\top \mathbf{M} \mathbf{X}_{\pi^\natural(i)} \mathbf{X}_{\pi^\natural(i)}^\top \mathbf{M} \mathbf{X}_{\pi^\natural(j)} \mathbf{X}_j^\top \mathbf{X}_j}_{\mathbb{1}_{i=j} \mathbb{1}_{i = \pi^{\natural 2}(i)} \mathbb{E}\mathbf{X}_i^\top \mathbf{X}_i \mathbf{X}_{\pi^\natural(i)}^\top \mathbf{M} \mathbf{X}_{\pi^\natural(i)} \mathbf{X}_{\pi^\natural(i)}^\top \mathbf{M} \mathbf{X}_i \mathbf{X}_i^\top \mathbf{X}_{\pi^\natural(i)}}$$

$$+ \underbrace{\mathbb{E}\mathbf{X}_j^\top \mathbf{X}_{\pi^\natural(i)} \mathbf{X}_{\pi^{\natural 2}(i)}^\top \mathbf{M} \mathbf{X}_{\pi^\natural(i)} \mathbf{X}_{\pi^\natural(i)}^\top \mathbf{M} \mathbf{X}_j \mathbf{X}_{\pi^{\natural-1}(j)}^\top \mathbf{X}_j}_{\mathbb{1}_{j = \pi^{\natural 2}(i)} \mathbb{E}\left(\mathbf{X}_j^\top \mathbf{X}_{\pi^\natural(i)}\right)^2 \left(\mathbf{X}_{\pi^\natural(i)}^\top \mathbf{M} \mathbf{X}_j\right)^2}$$

$$+ \underbrace{\mathbb{E}\mathbf{X}_j^\top \mathbf{X}_j \mathbf{X}_{\pi^\natural(j)}^\top \mathbf{M} \mathbf{X}_{\pi^\natural(i)} \mathbf{X}_{\pi^\natural(i)}^\top \mathbf{M} \mathbf{X}_{\pi^\natural(i)} \mathbf{X}_i^\top \mathbf{X}_j}_{\mathbb{1}_{i=j} \mathbb{E}\|\mathbf{X}_i\|_{\mathrm{F}}^4 \left(\mathbf{X}_{\pi^\natural(i)}^\top \mathbf{M} \mathbf{X}_{\pi^\natural(i)}\right)^2}$$

$$+ \underbrace{\mathbb{E}\mathbf{X}_j^\top \mathbf{X}_j \mathbf{X}_{\pi^\natural(j)}^\top \mathbf{M} \mathbf{X}_{\pi^\natural(i)} \mathbf{X}_{\pi^\natural(i)}^\top \mathbf{M} \mathbf{X}_{\pi^{\natural 2}(i)} \mathbf{X}_{\pi^\natural(i)}^\top \mathbf{X}_j}_{\mathbb{1}_{i=j} \mathbb{1}_{i = \pi^{\natural 2}(i)} \mathbb{E}\mathbf{X}_i^\top \mathbf{X}_i \mathbf{X}_{\pi^\natural(i)}^\top \mathbf{M} \mathbf{X}_{\pi^\natural(i)} \mathbf{X}_{\pi^\natural(i)}^\top \mathbf{M} \mathbf{X}_i \mathbf{X}_i^\top \mathbf{X}_{\pi^\natural(i)}}$$

$$+ \underbrace{\mathbb{E}\mathbf{X}_j^\top \mathbf{X}_j \mathbf{X}_{\pi^\natural(j)}^\top \mathbf{M} \mathbf{X}_{\pi^\natural(i)} \mathbf{X}_{\pi^\natural(i)}^\top \mathbf{M} \mathbf{X}_{\pi^\natural(j)} \mathbf{X}_j^\top \mathbf{X}_j}_{\mathbb{1}_{i=j} \mathbb{E}\|\mathbf{X}_i\|_{\mathrm{F}}^4 \left(\mathbf{X}_{\pi^\natural(i)}^\top \mathbf{M} \mathbf{X}_{\pi^\natural(i)}\right)^2 + \mathbb{1}_{i \neq j} \mathbb{E}\|\mathbf{X}_j\|_{\mathrm{F}}^4 \mathbf{X}_{\pi^\natural(i)}^\top \mathbf{M} \mathbf{M} \mathbf{X}_{\pi^\natural(i)}}$$

$$+ \underbrace{\mathbb{E}\mathbf{X}_j^\top \mathbf{X}_j \mathbf{X}_{\pi^\natural(j)}^\top \mathbf{M} \mathbf{X}_{\pi^\natural(i)} \mathbf{X}_{\pi^\natural(i)}^\top \mathbf{M} \mathbf{X}_j \mathbf{X}_{\pi^{\natural-1}(j)}^\top \mathbf{X}_j}_{\mathbb{1}_{j = \pi^{\natural 2}(j)} \left[\mathbb{1}_{i=j} \mathbb{E}\mathbf{X}_i^\top \mathbf{X}_i \mathbf{X}_{\pi^\natural(i)}^\top \mathbf{M} \mathbf{X}_{\pi^\natural(i)} \mathbf{X}_{\pi^\natural(i)}^\top \mathbf{M} \mathbf{X}_i \mathbf{X}_i^\top \mathbf{X}_{\pi^\natural(i)} + \mathbb{1}_{i \neq j} \mathbb{E}\|\mathbf{X}_j\|_{\mathrm{F}}^2 \mathbf{X}_j^\top \mathbf{M} \mathbf{M} \mathbf{X}_j\right]}$$

$$+ \underbrace{\mathbb{E}\mathbf{X}_j^\top \mathbf{X}_{\pi^{\natural-1}(j)} \mathbf{X}_j^\top \mathbf{M} \mathbf{X}_{\pi^\natural(i)} \mathbf{X}_{\pi^\natural(i)}^\top \mathbf{M} \mathbf{X}_{\pi^\natural(i)} \mathbf{X}_i^\top \mathbf{X}_j}_{\mathbb{1}_{i=j} \mathbb{1}_{i = \pi^{\natural 2}(i)} \mathbb{E}\mathbf{X}_i^\top \mathbf{X}_i \mathbf{X}_{\pi^\natural(i)}^\top \mathbf{M} \mathbf{X}_{\pi^\natural(i)} \mathbf{X}_{\pi^\natural(i)}^\top \mathbf{M} \mathbf{X}_i \mathbf{X}_i^\top \mathbf{X}_{\pi^\natural(i)}}$$

$$+ \underbrace{\mathbb{E}\mathbf{X}_j^\top \mathbf{X}_{\pi^{\natural-1}(j)} \mathbf{X}_j^\top \mathbf{M} \mathbf{X}_{\pi^\natural(i)} \mathbf{X}_{\pi^\natural(i)}^\top \mathbf{M} \mathbf{X}_{\pi^{\natural 2}(i)} \mathbf{X}_{\pi^\natural(i)}^\top \mathbf{X}_j}_{\mathbb{1}_{j = \pi^{\natural 2}(i)} \mathbb{E}\left(\mathbf{X}_j^\top \mathbf{X}_{\pi^\natural(i)}\right)^2 \left(\mathbf{X}_{\pi^\natural(i)}^\top \mathbf{M} \mathbf{X}_j\right)^2}$$

$$+ \underbrace{\mathbb{E}\mathbf{X}_j^\top \mathbf{X}_{\pi^{\natural-1}(j)} \mathbf{X}_j^\top \mathbf{M} \mathbf{X}_{\pi^\natural(i)} \mathbf{X}_{\pi^\natural(i)}^\top \mathbf{M} \mathbf{X}_{\pi^\natural(j)} \mathbf{X}_j^\top \mathbf{X}_j}_{\mathbb{1}_{j = \pi^{\natural 2}(j)} \left[\mathbb{1}_{i=j} \mathbb{E}\mathbf{X}_i^\top \mathbf{X}_i \mathbf{X}_{\pi^\natural(i)}^\top \mathbf{M} \mathbf{X}_{\pi^\natural(i)} \mathbf{X}_{\pi^\natural(i)}^\top \mathbf{M} \mathbf{X}_i \mathbf{X}_i^\top \mathbf{X}_{\pi^\natural(i)} + \mathbb{1}_{i \neq j} \mathbb{E}\|\mathbf{X}_j\|_{\mathrm{F}}^2 \mathbf{X}_j^\top \mathbf{M} \mathbf{M} \mathbf{X}_j\right]}$$

$$+ \underbrace{\mathbb{E}\mathbf{X}_j^\top \left(\mathbf{X}_{\pi^{\natural-1}(j)} \mathbf{X}_j^\top\right) \mathbf{M} \mathbf{X}_{\pi^\natural(i)} \mathbf{X}_{\pi^\natural(i)}^\top \mathbf{M} \left(\mathbf{X}_j \mathbf{X}_{\pi^{\natural-1}(j)}^\top\right) \mathbf{X}_j}_{\mathbb{1}_{j = \pi^{\natural 2}(i)} \mathbb{E}\left(\mathbf{X}_j^\top \mathbf{X}_{\pi^\natural(i)}\right)^2 \left(\mathbf{X}_{\pi^\natural(i)}^\top \mathbf{M} \mathbf{X}_j\right)^2 + \mathbb{1}_{j \neq \pi^{\natural 2}(i)} \mathbb{E}\|\mathbf{X}_j\|_{\mathrm{F}}^2 \mathbf{X}_j^\top \mathbf{M} \mathbf{M} \mathbf{X}_j}$$

$$= 4 \mathbb{1}_{i=j} p(p+2) \left[[\mathrm{Tr}(\mathbf{M})]^2 + 2 \mathrm{Tr}(\mathbf{M}\mathbf{M})\right] + \mathbb{1}_{i \neq j} p(p+2) \mathrm{Tr}(\mathbf{M}\mathbf{M})$$

$$+ 8 \mathbb{1}_{i=j} \mathbb{1}_{i = \pi^{\natural 2}(i)} (p+2) \left[(\mathrm{Tr}(\mathbf{M}))^2 + 2 \mathrm{Tr}(\mathbf{M}\mathbf{M})\right]$$

$$+ 4 \mathbb{1}_{j = \pi^{\natural 2}(i)} \left[2 [\mathrm{Tr}(\mathbf{M})]^2 + (p+6) \mathrm{Tr}(\mathbf{M}\mathbf{M})\right]$$

$$+ 2 \mathbb{1}_{j \neq \pi^{\natural 2}(i)} (p+2) \mathrm{Tr}(\mathbf{M}\mathbf{M}) + 2 \mathbb{1}_{i \neq j} \mathbb{1}_{j = \pi^{\natural 2}(j)} (p+2) \mathrm{Tr}(\mathbf{M}\mathbf{M})$$

$$+ \mathbb{1}_{i \neq j} \left(p + 2 \mathbb{1}_{i = \pi^{\natural 2}(i)}\right) \left[(\mathrm{Tr}(\mathbf{M}))^2 + 2 \mathrm{Tr}(\mathbf{M}\mathbf{M})\right]. \tag{40}$$

The term $\Lambda_{3,3}$ is computed as

$$\mathbb{E}\Lambda_{3,3} = \underbrace{\mathbb{E}\mathbf{X}_{\pi^\natural(i)}^\top \mathbf{X}_i \mathbf{X}_{\pi^\natural(i)}^\top \mathbf{M} \mathbf{X}_{\pi^\natural(i)} \mathbf{X}_{\pi^\natural(i)}^\top \mathbf{M} \mathbf{X}_{\pi^\natural(i)} \mathbf{X}_i^\top \mathbf{X}_j}_{0}$$

$$+ \underbrace{\mathbb{E}\mathbf{X}_{\pi^\natural(i)}^\top \mathbf{X}_i \mathbf{X}_{\pi^\natural(i)}^\top \mathbf{M} \mathbf{X}_{\pi^\natural(i)} \mathbf{X}_{\pi^\natural(i)}^\top \mathbf{M} \mathbf{X}_{\pi^{\natural 2}(i)} \mathbf{X}_{\pi^\natural(i)}^\top \mathbf{X}_j}_{0}$$

$$+ \underbrace{\mathbb{E}\mathbf{X}_{\pi^\natural(i)}^\top \mathbf{X}_i \mathbf{X}_{\pi^\natural(i)}^\top \mathbf{M} \mathbf{X}_{\pi^\natural(i)} \mathbf{X}_{\pi^\natural(i)}^\top \mathbf{M} \mathbf{X}_{\pi^\natural(j)} \mathbf{X}_j^\top \mathbf{X}_j}_{\mathbb{1}_{i = \pi^\natural(j)} p \mathbb{E}\left[\mathbf{X}_{\pi^\natural(i)}^\top \mathbf{M} \mathbf{X}_{\pi^\natural(i)}\right]^2}$$

$$+ \underbrace{\mathbb{E}\mathbf{X}_{\pi^\natural(i)}^\top \mathbf{X}_i \mathbf{X}_{\pi^\natural(i)}^\top \mathbf{M}\mathbf{X}_{\pi^\natural(i)} \mathbf{X}_{\pi^\natural(i)}^\top \mathbf{M}\mathbf{X}_j \mathbf{X}_{\pi^{\natural-1}(j)}^\top \mathbf{X}_j}_{0}$$

$$+ \underbrace{\mathbb{E}\mathbf{X}_{\pi^\natural(i)}^\top \mathbf{X}_{\pi^\natural(i)} \mathbf{X}_{\pi^{\natural2}(i)}^\top \mathbf{M}\mathbf{X}_{\pi^\natural(i)} \mathbf{X}_{\pi^\natural(i)}^\top \mathbf{M}\mathbf{X}_{\pi^\natural(i)} \mathbf{X}_i^\top \mathbf{X}_j}_{0}$$

$$+ \underbrace{\mathbb{E}\mathbf{X}_{\pi^\natural(i)}^\top \mathbf{X}_{\pi^\natural(i)} \mathbf{X}_{\pi^{\natural2}(i)}^\top \mathbf{M}\mathbf{X}_{\pi^\natural(i)} \mathbf{X}_{\pi^\natural(i)}^\top \mathbf{M}\mathbf{X}_{\pi^{\natural2}(i)} \mathbf{X}_{\pi^\natural(i)}^\top \mathbf{X}_j}_{0}$$

$$+ \underbrace{\mathbb{E}\mathbf{X}_{\pi^\natural(i)}^\top \mathbf{X}_{\pi^\natural(i)} \mathbf{X}_{\pi^{\natural2}(i)}^\top \mathbf{M}\mathbf{X}_{\pi^\natural(i)} \mathbf{X}_{\pi^\natural(i)}^\top \mathbf{M}\mathbf{X}_{\pi^\natural(j)} \mathbf{X}_j^\top \mathbf{X}_j}_{0}$$

$$+ \underbrace{\mathbb{E}\mathbf{X}_{\pi^\natural(i)}^\top \mathbf{X}_{\pi^\natural(i)} \mathbf{X}_{\pi^{\natural2}(i)}^\top \mathbf{M}\mathbf{X}_{\pi^\natural(i)} \mathbf{X}_{\pi^\natural(i)}^\top \mathbf{M}\mathbf{X}_j \mathbf{X}_{\pi^{\natural-1}(j)}^\top \mathbf{X}_j}_{\mathbb{1}_{j=\pi^{\natural3}(i)}\mathbb{E}\left\|\mathbf{X}_{\pi^\natural(i)}\right\|_F^2 \mathbf{X}_{\pi^\natural(i)}^\top \mathbf{M}\mathbf{M}\mathbf{X}_{\pi^\natural(i)}}$$

$$+ \underbrace{\mathbb{E}\mathbf{X}_{\pi^\natural(i)}^\top \mathbf{X}_j \mathbf{X}_{\pi^\natural(j)}^\top \mathbf{M}\mathbf{X}_{\pi^\natural(i)} \mathbf{X}_{\pi^\natural(i)}^\top \mathbf{M}\mathbf{X}_{\pi^\natural(i)} \mathbf{X}_i^\top \mathbf{X}_j}_{\mathbb{1}_{i=\pi^\natural(j)}\mathbb{E}\left[\mathbf{X}_{\pi^\natural(i)}^\top \mathbf{M}\mathbf{X}_{\pi^\natural(i)}\right]^2}$$

$$+ \underbrace{\mathbb{E}\mathbf{X}_{\pi^\natural(i)}^\top \mathbf{X}_j \mathbf{X}_{\pi^\natural(j)}^\top \mathbf{M}\mathbf{X}_{\pi^\natural(i)} \mathbf{X}_{\pi^\natural(i)}^\top \mathbf{M}\mathbf{X}_{\pi^{\natural2}(i)} \mathbf{X}_{\pi^\natural(i)}^\top \mathbf{X}_j}_{0}$$

$$+ \underbrace{\mathbb{E}\mathbf{X}_{\pi^\natural(i)}^\top \mathbf{X}_j \mathbf{X}_{\pi^\natural(j)}^\top \mathbf{M}\mathbf{X}_{\pi^\natural(i)} \mathbf{X}_{\pi^\natural(i)}^\top \mathbf{M}\mathbf{X}_{\pi^\natural(j)} \mathbf{X}_j^\top \mathbf{X}_j}_{0}$$

$$+ \underbrace{\mathbb{E}\mathbf{X}_{\pi^\natural(i)}^\top \mathbf{X}_j \mathbf{X}_{\pi^\natural(j)}^\top \mathbf{M}\mathbf{X}_{\pi^\natural(i)} \mathbf{X}_{\pi^\natural(i)}^\top \mathbf{M}\mathbf{X}_j \mathbf{X}_{\pi^{\natural-1}(j)}^\top \mathbf{X}_j}_{0}$$

$$+ \underbrace{\mathbb{E}\mathbf{X}_{\pi^\natural(i)}^\top \mathbf{X}_{\pi^{\natural-1}(j)} \mathbf{X}_j^\top \mathbf{M}\mathbf{X}_{\pi^\natural(i)} \mathbf{X}_{\pi^\natural(i)}^\top \mathbf{M}\mathbf{X}_{\pi^\natural(i)} \mathbf{X}_i^\top \mathbf{X}_j}_{0}$$

$$+ \underbrace{\mathbb{E}\mathbf{X}_{\pi^\natural(i)}^\top \mathbf{X}_{\pi^{\natural-1}(j)} \mathbf{X}_j^\top \mathbf{M}\mathbf{X}_{\pi^\natural(i)} \mathbf{X}_{\pi^\natural(i)}^\top \mathbf{M}\mathbf{X}_{\pi^{\natural2}(i)} \mathbf{X}_{\pi^\natural(i)}^\top \mathbf{X}_j}_{\mathbb{1}_{j=\pi^{\natural3}(i)}\mathbb{E}\left[\mathbf{X}_{\pi^\natural(i)}^\top \mathbf{M}\mathbf{X}_{\pi^\natural(i)}\right]^2}$$

$$+ \underbrace{\mathbb{E}\mathbf{X}_{\pi^\natural(i)}^\top \mathbf{X}_{\pi^{\natural-1}(j)} \mathbf{X}_j^\top \mathbf{M}\mathbf{X}_{\pi^\natural(i)} \mathbf{X}_{\pi^\natural(i)}^\top \mathbf{M}\mathbf{X}_{\pi^\natural(j)} \mathbf{X}_j^\top \mathbf{X}_j}_{0}$$

$$+ \underbrace{\mathbb{E}\mathbf{X}_{\pi^\natural(i)}^\top \mathbf{X}_{\pi^{\natural-1}(j)} \mathbf{X}_j^\top \mathbf{M}\mathbf{X}_{\pi^\natural(i)} \mathbf{X}_{\pi^\natural(i)}^\top \mathbf{M}\mathbf{X}_j \mathbf{X}_{\pi^{\natural-1}(j)}^\top \mathbf{X}_j}_{0}$$

$$= \left[(p+1)\mathbb{1}_{i=\pi^\natural(j)} + \mathbb{1}_{j=\pi^{\natural3}(i)}\right][\mathrm{Tr}(\mathbf{M})]^2$$
$$+ \left[2(p+1)\mathbb{1}_{i=\pi^\natural(j)} + (p+4)\mathbb{1}_{j=\pi^{\natural3}(i)}\right]\mathrm{Tr}(\mathbf{M}\mathbf{M}). \tag{41}$$

Then, we consider the term $\mathbb{E}\Lambda_{3,4}$, which can be written as

$$\mathbb{E}\Lambda_{3,4} = \underbrace{\mathbb{E}\mathbf{X}_j^\top \mathbf{X}_i \mathbf{X}_{\pi^\natural(i)}^\top \mathbf{M}\mathbf{X}_{\pi^\natural(i)} \mathbf{X}_{\pi^\natural(i)}^\top \mathbf{M}\mathbf{X}_{\pi^\natural(i)} \mathbf{X}_i^\top \mathbf{X}_{\pi^\natural(i)}}_{0}$$

$$+ \underbrace{\mathbb{E}\mathbf{X}_j^\top \mathbf{X}_i \mathbf{X}_{\pi^\natural(i)}^\top \mathbf{M}\mathbf{X}_{\pi^\natural(i)} \mathbf{X}_{\pi^\natural(i)}^\top \mathbf{M}\mathbf{X}_{\pi^{\natural2}(i)} \mathbf{X}_{\pi^\natural(i)}^\top \mathbf{X}_{\pi^\natural(i)}}_{0}$$

$$+ \underbrace{\mathbb{E}\mathbf{X}_j^\top \mathbf{X}_i \mathbf{X}_{\pi^\natural(i)}^\top \mathbf{M}\mathbf{X}_{\pi^\natural(i)} \mathbf{X}_{\pi^\natural(i)}^\top \mathbf{M}\mathbf{X}_{\pi^\natural(j)} \mathbf{X}_j^\top \mathbf{X}_{\pi^\natural(i)}}_{\mathbb{1}_{i=\pi^\natural(j)}\mathbb{E}\left[\mathbf{X}_{\pi^\natural(i)}^\top \mathbf{M}\mathbf{X}_{\pi^\natural(i)}\right]^2}$$

$$+ \underbrace{\mathbb{E}\mathbf{X}_j^\top \mathbf{X}_i \mathbf{X}_{\pi^\natural(i)}^\top \mathbf{M}\mathbf{X}_{\pi^\natural(i)} \mathbf{X}_{\pi^\natural(i)}^\top \mathbf{M}\mathbf{X}_j \mathbf{X}_{\pi^{\natural-1}(j)}^\top \mathbf{X}_{\pi^\natural(i)}}_{0}$$

$$+ \underbrace{\mathbb{E}\mathbf{X}_j^\top \mathbf{X}_{\pi^\natural(i)} \mathbf{X}_{\pi^{\natural2}(i)}^\top \mathbf{M}\mathbf{X}_{\pi^\natural(i)} \mathbf{X}_{\pi^\natural(i)}^\top \mathbf{M}\mathbf{X}_{\pi^\natural(i)} \mathbf{X}_i^\top \mathbf{X}_{\pi^\natural(i)}}_{0}$$

$$+ \underbrace{\mathbb{E}\mathbf{X}_j^\top \mathbf{X}_{\pi^\natural(i)} \mathbf{X}_{\pi^{\natural 2}(i)}^\top \mathbf{M} \mathbf{X}_{\pi^\natural(i)} \mathbf{X}_{\pi^\natural(i)}^\top \mathbf{M} \mathbf{X}_{\pi^{\natural 2}(i)} \mathbf{X}_{\pi^\natural(i)}^\top \mathbf{X}_{\pi^\natural(i)}}_{0}$$

$$+ \underbrace{\mathbb{E}\mathbf{X}_j^\top \mathbf{X}_{\pi^\natural(i)} \mathbf{X}_{\pi^{\natural 2}(i)}^\top \mathbf{M} \mathbf{X}_{\pi^\natural(i)} \mathbf{X}_{\pi^\natural(i)}^\top \mathbf{M} \mathbf{X}_{\pi^\natural(j)} \mathbf{X}_j^\top \mathbf{X}_{\pi^\natural(i)}}_{0}$$

$$+ \underbrace{\mathbb{E}\mathbf{X}_j^\top \mathbf{X}_{\pi^\natural(i)} \mathbf{X}_{\pi^{\natural 2}(i)}^\top \mathbf{M} \mathbf{X}_{\pi^\natural(i)} \mathbf{X}_{\pi^\natural(i)}^\top \mathbf{M} \mathbf{X}_j \mathbf{X}_{\pi^{\natural-1}(j)}^\top \mathbf{X}_{\pi^\natural(i)}}_{\mathbb{1}_{j=\pi^{\natural 3}(i)} \mathbb{E}\left[\mathbf{X}_{\pi^\natural(i)}^\top \mathbf{M} \mathbf{X}_{\pi^\natural(i)}\right]^2}$$

$$+ \underbrace{\mathbb{E}\mathbf{X}_j^\top \mathbf{X}_j \mathbf{X}_{\pi^\natural(j)}^\top \mathbf{M} \mathbf{X}_{\pi^\natural(i)} \mathbf{X}_{\pi^\natural(i)}^\top \mathbf{M} \mathbf{X}_{\pi^\natural(i)} \mathbf{X}_i^\top \mathbf{X}_{\pi^\natural(i)}}_{\mathbb{1}_{i=\pi^\natural(j)} p \mathbb{E}\left[\mathbf{X}_{\pi^\natural(i)}^\top \mathbf{M} \mathbf{X}_{\pi^\natural(i)}\right]^2}$$

$$+ \underbrace{\mathbb{E}\mathbf{X}_j^\top \mathbf{X}_j \mathbf{X}_{\pi^\natural(j)}^\top \mathbf{M} \mathbf{X}_{\pi^\natural(i)} \mathbf{X}_{\pi^\natural(i)}^\top \mathbf{M} \mathbf{X}_{\pi^{\natural 2}(i)} \mathbf{X}_{\pi^\natural(i)}^\top \mathbf{X}_{\pi^\natural(i)}}_{0}$$

$$+ \underbrace{\mathbb{E}\mathbf{X}_j^\top \mathbf{X}_j \mathbf{X}_{\pi^\natural(j)}^\top \mathbf{M} \mathbf{X}_{\pi^\natural(i)} \mathbf{X}_{\pi^\natural(i)}^\top \mathbf{M} \mathbf{X}_{\pi^\natural(j)} \mathbf{X}_j^\top \mathbf{X}_{\pi^\natural(i)}}_{0}$$

$$+ \underbrace{\mathbb{E}\mathbf{X}_j^\top \mathbf{X}_j \mathbf{X}_{\pi^\natural(j)}^\top \mathbf{M} \mathbf{X}_{\pi^\natural(i)} \mathbf{X}_{\pi^\natural(i)}^\top \mathbf{M} \mathbf{X}_j \mathbf{X}_{\pi^{\natural-1}(j)}^\top \mathbf{X}_{\pi^\natural(i)}}_{0}$$

$$+ \underbrace{\mathbb{E}\mathbf{X}_j^\top \mathbf{X}_{\pi^{\natural-1}(j)} \mathbf{X}_j^\top \mathbf{M} \mathbf{X}_{\pi^\natural(i)} \mathbf{X}_{\pi^\natural(i)}^\top \mathbf{M} \mathbf{X}_{\pi^\natural(i)} \mathbf{X}_i^\top \mathbf{X}_{\pi^\natural(i)}}_{0}$$

$$+ \underbrace{\mathbb{E}\mathbf{X}_j^\top \mathbf{X}_{\pi^{\natural-1}(j)} \mathbf{X}_j^\top \mathbf{M} \mathbf{X}_{\pi^\natural(i)} \mathbf{X}_{\pi^\natural(i)}^\top \mathbf{M} \mathbf{X}_{\pi^{\natural 2}(i)} \mathbf{X}_{\pi^\natural(i)}^\top \mathbf{X}_{\pi^\natural(i)}}_{\mathbb{1}_{j=\pi^{\natural 3}(i)} \mathbb{E}\left\|\mathbf{X}_{\pi^\natural(i)}\right\|_F^2 \mathbf{X}_{\pi^\natural(i)}^\top \mathbf{M} \mathbf{M} \mathbf{X}_{\pi^\natural(i)}}$$

$$+ \underbrace{\mathbb{E}\mathbf{X}_j^\top \mathbf{X}_{\pi^{\natural-1}(j)} \mathbf{X}_j^\top \mathbf{M} \mathbf{X}_{\pi^\natural(i)} \mathbf{X}_{\pi^\natural(i)}^\top \mathbf{M} \mathbf{X}_{\pi^\natural(j)} \mathbf{X}_j^\top \mathbf{X}_{\pi^\natural(i)}}_{0}$$

$$+ \underbrace{\mathbb{E}\mathbf{X}_j^\top \mathbf{X}_{\pi^{\natural-1}(j)} \mathbf{X}_j^\top \mathbf{M} \mathbf{X}_{\pi^\natural(i)} \mathbf{X}_{\pi^\natural(i)}^\top \mathbf{M} \mathbf{X}_j \mathbf{X}_{\pi^{\natural-1}(j)}^\top \mathbf{X}_{\pi^\natural(i)}}_{0}$$

$$= \left[(p+1)\mathbb{1}_{i=\pi^\natural(j)} + \mathbb{1}_{j=\pi^{\natural 3}(i)}\right] \left[\mathrm{Tr}(\mathbf{M})\right]^2$$
$$+ \left[2(p+1)\mathbb{1}_{i=\pi^\natural(j)} + (p+4)\mathbb{1}_{j=\pi^{\natural 3}(i)}\right] \mathrm{Tr}(\mathbf{M}\mathbf{M}). \tag{42}$$

Combing (39), (40), (41), and (42) together then yields

$$\mathbb{E}\Lambda_3 = \frac{4hp(p+2)}{n^2} \left[1 + o(1)\right] \left[\mathrm{Tr}(\mathbf{M})\right]^2 + 2p^2 \left[1 + o(1)\right] \mathrm{Tr}(\mathbf{M}\mathbf{M}). \tag{43}$$

The proof is thus completed by summarizing the computations thereof. $\qquad\square$

.

**Lemma 6.** *We have*

$$\mathbb{E}\Xi_2^2 = 2\left[(p+2)(p+3) + (n-2)(p+1)\right] \left\|\!\left\|\mathbf{B}^\natural\right\|\!\right\|_F^2 = 2np\left(1 + p/n + o(1)\right) \left\|\!\left\|\mathbf{B}^\natural\right\|\!\right\|_F^2,$$

*where $\Xi_2$ is defined in* (21).

*Proof.* We have

$$\mathbb{E}\Xi_2^2 = \mathbb{E}\left[\mathbf{X}_{\pi^\natural(i)}^\top \mathbf{B}^\natural \mathbf{W}^\top \mathbf{X} \left(\mathbf{X}_{\pi^\natural(i)} - \mathbf{X}_j\right) \left(\mathbf{X}_{\pi^\natural(i)} - \mathbf{X}_j\right)^\top \mathbf{X}^\top \mathbf{W} \mathbf{B}^{\natural\top} \mathbf{X}_{\pi^\natural(i)}\right]$$

$$= \mathbb{E}\left[\mathbf{X}_{\pi^\natural(i)}^\top \mathbf{B}^\natural \mathrm{Tr}\left[\mathbf{X}\left(\mathbf{X}_{\pi^\natural(i)} - \mathbf{X}_j\right) \left(\mathbf{X}_{\pi^\natural(i)} - \mathbf{X}_j\right)^\top \mathbf{X}^\top\right] \mathbf{B}^{\natural\top} \mathbf{X}_{\pi^\natural(i)}\right]$$

$$= \mathbb{E}\left[\left\|\mathbf{X}\left(\mathbf{X}_{\pi^\natural(i)} - \mathbf{X}_j\right)\right\|_2^2 \mathbf{X}_{\pi^\natural(i)}^\top \mathbf{B}^\natural \mathbf{B}^{\natural\top} \mathbf{X}_{\pi^\natural(i)}\right]$$

$$= \mathbb{E}\left[\left\|\mathbf{X}\left(\mathbf{X}_{\pi^\natural(i)} - \mathbf{X}_j\right)\right\|_2^2 \times \left\|\mathbf{B}^{\natural\top} \mathbf{X}_{\pi^\natural(i)}\right\|_2^2\right].$$

For the conciseness of notation, we assume $\pi^\natural(i) = 1$ and $j = 2$ w.l.o.g. Decomposing the term $\|\mathbf{X}\left(\mathbf{X}_1 - \mathbf{X}_2\right)\|_2^2$ as

$$\|\mathbf{X}\left(\mathbf{X}_1 - \mathbf{X}_2\right)\|_F^2 = \underbrace{\left[\mathbf{X}_1^\top\left(\mathbf{X}_1 - \mathbf{X}_2\right)\right]^2}_{\mathcal{T}_1} + \underbrace{\left[\mathbf{X}_2^\top\left(\mathbf{X}_1 - \mathbf{X}_2\right)\right]^2}_{\mathcal{T}_2} + \underbrace{\sum_{i=3}^{n}\left[\mathbf{X}_i^\top\left(\mathbf{X}_1 - \mathbf{X}_2\right)\right]^2}_{\mathcal{T}_3},$$

we then separately bound the above three terms. For the first term $\mathbb{E}\mathcal{T}_1\left\|\|\mathbf{B}^{\natural\top}\mathbf{X}_1\right\|\|_F^2$, we have

$$\begin{aligned}
\mathbb{E}\mathcal{T}_1\left\|\|\mathbf{B}^{\natural\top}\mathbf{X}_1\right\|\|_F^2 &= \mathbb{E}\left[\left(\|\mathbf{X}_1\|_2^4 + \left(\mathbf{X}_1^\top\mathbf{X}_2\right)^2\right)\left\|\|\mathbf{B}^{\natural\top}\mathbf{X}_1\right\|\|_F^2\right] \\
&= \underbrace{\mathbb{E}\|\mathbf{X}_1\|_2^4\left\|\|\mathbf{B}^{\natural\top}\mathbf{X}_1\right\|\|_F^2}_{(p+2)(p+4)\||\mathbf{B}^\natural\||_F^2} + \underbrace{\mathbb{E}\left(\mathbf{X}_1^\top\mathbf{X}_2\right)^2\left\|\|\mathbf{B}^{\natural\top}\mathbf{X}_1\right\|\|_F^2}_{(p+2)\||\mathbf{B}^\natural\||_F^2} \overset{\text{①}}{=} (p+2)(p+5)\||\mathbf{B}^\natural\||_F^2,
\end{aligned}$$

$$(44)$$

where ① is due to (66) and (67).

Similarly, for term $\mathbb{E}\mathcal{T}_2\left\|\|\mathbf{B}^{\natural\top}\mathbf{X}_1\right\|\|_F^2$, we invoke (66) and (67), which gives

$$\mathbb{E}\mathcal{T}_2\left\|\|\mathbf{B}^{\natural\top}\mathbf{X}_1\right\|\|_F^2 = \underbrace{\mathbb{E}\left[\left(\mathbf{X}_1^\top\mathbf{X}_2\right)^2\left\|\|\mathbf{B}^{\natural\top}\mathbf{X}_1\right\|\|_F^2\right]}_{(p+2)\||\mathbf{B}^\natural\||_F^2} + \underbrace{\mathbb{E}\|\mathbf{X}_2\|_2^4 \times \mathbb{E}\left\|\|\mathbf{B}^{\natural\top}\mathbf{X}_1\right\|\|_F^2}_{p(p+2)\||\mathbf{B}^\natural\||_F^2} \tag{45}$$

$$\overset{\text{②}}{=} (p+1)(p+2)\||\mathbf{B}^\natural\||_F^2, \tag{46}$$

where ② is due to (66).

For the last term $\mathbb{E}\mathcal{T}_3\left\|\|\mathbf{B}^{\natural\top}\mathbf{X}_1\right\|\|_F^2$, we exploit the independence among the rows of matrix $\mathbf{X}$ and have

$$\begin{aligned}
\mathbb{E}T_3\left\|\|\mathbf{B}^{\natural\top}\mathbf{X}_1\right\|\|_F^2 &= \sum_{i\geq 3}\mathbb{E}\left[\left(\mathbf{X}_i^\top\left(\mathbf{X}_1 - \mathbf{X}_2\right)\right)^2\left\|\|\mathbf{B}^{\natural\top}\mathbf{X}_1\right\|\|_F^2\right] \\
&= \sum_{i\geq 3}\mathbb{E}\left[\|\mathbf{X}_1 - \mathbf{X}_2\|_2^2 \cdot \left\|\|\mathbf{B}^{\natural\top}\mathbf{X}_1\right\|\|_F^2\right] \\
&= \sum_{i\geq 3}\mathbb{E}\left[\left(\|\mathbf{X}_1\|_2^2 + \|\mathbf{X}_2\|_2^2\right) \cdot \left\|\|\mathbf{B}^{\natural\top}\mathbf{X}_1\right\|\|_F^2\right] \\
&= 2\sum_{i\geq 3}(p+1)\||\mathbf{B}^\natural\||_F^2 = 2(n-2)(p+1)\||\mathbf{B}^\natural\||_F^2.
\end{aligned} \tag{47}$$

The proof is then completed by combining (44), (46), and (47).

$\square$

**Lemma 7.** *We have*

$$\mathbb{E}\Xi_3^2 = 2n^2\left[\frac{p}{n} + \left(1 - \frac{h}{n}\right)^2 + \frac{p^2}{n^2} + \frac{4p(n-h)^2}{n^3} + o(1)\right]\mathrm{Tr}(\mathbf{M}),$$

*where $\Xi_3$ is defined in (21).*

*Proof.* To begin with, we decompose the term $\mathbb{E}\Xi_3^2$ as

$$\mathbb{E}\Xi_3^2 = \underbrace{\mathbb{E}\left[\left(\mathbf{X}_{\pi^\natural(i)} - \mathbf{X}_j\right)^\top \mathbf{\Sigma M \Sigma}^\top\left(\mathbf{X}_{\pi^\natural(i)} - \mathbf{X}_j\right)\right]}_{\triangleq\Lambda_1} + 2\underbrace{\mathbb{E}\left[\left(\mathbf{X}_{\pi^\natural(i)} - \mathbf{X}_j\right)^\top \mathbf{\Sigma M \Delta}^\top\left(\mathbf{X}_{\pi^\natural(i)} - \mathbf{X}_j\right)\right]}_{\triangleq\Lambda_2}$$

$$+ \underbrace{\mathbb{E}\left[\left(\mathbf{X}_{\pi^\natural(i)} - \mathbf{X}_j\right)^\top \mathbf{\Delta M \Delta}^\top\left(\mathbf{X}_{\pi^\natural(i)} - \mathbf{X}_j\right)\right]}_{\triangleq\Lambda_3}. \tag{48}$$

**Step I.** First we consider $\mathbb{E}\Lambda_1$, which can be written as

$$\mathbb{E}\Lambda_1 = 2\mathbb{E}\operatorname{Tr}\left(\boldsymbol{\Sigma}\mathbf{M}\boldsymbol{\Sigma}^\top\right) \stackrel{\textcircled{1}}{=} 2n^2\left[\frac{p}{n} + \left(1 - \frac{h}{n}\right)^2 + o(1)\right]\operatorname{Tr}(\mathbf{M}), \qquad (49)$$

where $\textcircled{1}$ is due to Lemma 13.

**Step II.** Then we turn to $\mathbb{E}\Lambda_2$, which can be written as

$$\mathbb{E}\Lambda_2 = (n-h)\mathbb{E}\underbrace{\left[\mathbf{X}_{\pi^\natural(i)}^\top \mathbf{M}\boldsymbol{\Delta}^\top \mathbf{X}_{\pi^\natural(i)}\right]}_{\Lambda_{2,1}} + (n-h)\mathbb{E}\underbrace{\left[\mathbf{X}_j^\top \mathbf{M}\boldsymbol{\Delta}^\top \mathbf{X}_j\right]}_{\Lambda_{2,2}}$$

$$- (n-h)\mathbb{E}\underbrace{\left[\mathbf{X}_{\pi^\natural(i)}^\top \mathbf{M}\boldsymbol{\Delta}^\top \mathbf{X}_j\right]}_{\Lambda_{2,3}} - (n-h)\mathbb{E}\underbrace{\left[\mathbf{X}_j^\top \mathbf{M}\boldsymbol{\Delta}^\top \mathbf{X}_{\pi^\natural(i)}\right]}_{\Lambda_{2,4}}.$$

**Case $(s,s)$: $i = \pi^\natural(i)$ and $j = \pi^\natural(j)$.** We have

$$\mathbb{E}\Lambda_{2,1} = \mathbb{E}\mathbf{X}_i^\top \mathbf{M}\left(\mathbf{X}_i\mathbf{X}_i^\top + \mathbf{X}_j\mathbf{X}_j^\top\right)\mathbf{X}_i = (p+3)\operatorname{Tr}(\mathbf{M}),$$

$$\mathbb{E}\Lambda_{2,2} = \mathbb{E}\mathbf{X}_j^\top \mathbf{M}\left(\mathbf{X}_i\mathbf{X}_i^\top + \mathbf{X}_j\mathbf{X}_j^\top\right)\mathbf{X}_j = (p+3)\operatorname{Tr}(\mathbf{M}).$$

In addition, we can verify that $\mathbb{E}\Lambda_{2,2}$ and $\Lambda_{2,3}$ are both zero, which suggests that

$$\mathbb{E}\Lambda_2 = 2(n-h)(p+3)\operatorname{Tr}(\mathbf{M}). \qquad (50)$$

**Case $(s,d)$: $i = \pi^\natural(i)$ and $j \neq \pi^\natural(j)$.** We have

$$\mathbb{E}\Lambda_{2,1} = \mathbb{E}\mathbf{X}_i^\top \mathbf{M}\left(\mathbf{X}_i\mathbf{X}_i^\top + \mathbf{X}_j\mathbf{X}_{\pi^\natural(j)}^\top + \mathbf{X}_{\pi^{\natural-1}(j)}\mathbf{X}_j^\top\right)\mathbf{X}_i = (p+2)\operatorname{Tr}(\mathbf{M});$$

$$\mathbb{E}\Lambda_{2,2} = \mathbb{E}\mathbf{X}_j^\top \mathbf{M}\left(\mathbf{X}_i\mathbf{X}_i^\top + \mathbf{X}_j\mathbf{X}_{\pi^\natural(j)}^\top + \mathbf{X}_{\pi^{\natural-1}(j)}\mathbf{X}_j^\top\right)\mathbf{X}_j = \operatorname{Tr}(\mathbf{M}).$$

Moreover, we have both $\mathbb{E}\Lambda_{2,3}$ and $\mathbb{E}\Lambda_{2,4}$ be zero, which suggests that

$$\mathbb{E}\Lambda_2 = (n-h)(p+3)\operatorname{Tr}(\mathbf{M}). \qquad (51)$$

**Case $(d,s)$: $i \neq \pi^\natural(i)$ and $j = \pi^\natural(j)$.** We have

$$\mathbb{E}\Lambda_{2,1} = \mathbb{E}\left(\mathbf{X}_{\pi^\natural(i)}^\top \mathbf{M}\mathbf{X}_j\mathbf{X}_j^\top \mathbf{X}_{\pi^\natural(i)}\right) = \operatorname{Tr}(\mathbf{M}),$$

$$\mathbb{E}\Lambda_{2,2} = \mathbb{E}\mathbf{X}_j^\top \mathbf{M}\mathbf{X}_j\mathbf{X}_j^\top \mathbf{X}_j = (p+2)\operatorname{Tr}(\mathbf{M}).$$

Similar as above, we can verify both $\mathbb{E}\Lambda_{2,3}$ and $\mathbb{E}\Lambda_{2,4}$ are zero, which suggests that

$$\mathbb{E}\Lambda_2 = (n-h)(p+3)\operatorname{Tr}(\mathbf{M}). \qquad (52)$$

**Case $(d,d)$: $i \neq \pi^\natural(i)$ and $j \neq \pi^\natural(j)$.** Different from the above three cases, we have $\mathbb{E}\Lambda_{2,1}$ and $\mathbb{E}\Lambda_{2,2}$ be zero and focus on the calculation of $\mathbb{E}\Lambda_{2,3}$ and $\mathbb{E}\Lambda_{2,4}$, which proceeds as

$$\mathbb{E}\Lambda_{2,3} = \mathbb{E}\underbrace{\left[\mathbf{X}_{\pi^\natural(i)}^\top \mathbf{M}\mathbf{X}_{\pi^\natural(i)}\mathbf{X}_i^\top \mathbf{X}_j\right]}_{p\mathbb{1}_{i=j}\|\mathbf{B}^\natural\|_{\mathrm{F}}^2} + \mathbb{E}\underbrace{\left[\mathbf{X}_{\pi^\natural(i)}^\top \mathbf{M}\mathbf{X}_{\pi^{\natural2}(i)}\mathbf{X}_{\pi^\natural(i)}^\top \mathbf{X}_j\right]}_{\mathbb{1}_{j=\pi^{\natural2}(i)}\|\mathbf{B}^\natural\|_{\mathrm{F}}^2}$$

$$+ \mathbb{E}\underbrace{\left[\mathbf{X}_{\pi^\natural(i)}^\top \mathbf{M}\mathbf{X}_{\pi^\natural(j)}\mathbf{X}_j^\top \mathbf{X}_j\right]}_{p\mathbb{1}_{i=j}\|\mathbf{B}^\natural\|_{\mathrm{F}}^2} + \mathbb{E}\underbrace{\left[\mathbf{X}_{\pi^\natural(i)}^\top \mathbf{M}\mathbf{X}_j\mathbf{X}_{\pi^{\natural-1}(j)}^\top \mathbf{X}_j\right]}_{\mathbb{1}_{\pi^\natural(i)=\pi^{\natural-1}(j)}\|\mathbf{B}^\natural\|_{\mathrm{F}}^2}$$

$$= 2\left[p\mathbb{1}_{i=j} + \mathbb{1}_{j=\pi^{\natural2}(i)}\right]\operatorname{Tr}(\mathbf{M});$$

$$\mathbb{E}\Lambda_{2,4} = \underbrace{\mathbb{E}\left[\mathbf{X}_j^\top \mathbf{M}\mathbf{X}_{\pi^\natural(i)}\mathbf{X}_i^\top \mathbf{X}_{\pi^\natural(i)}\right]}_{\mathbb{1}_{i=j}\|\mathbf{B}^\natural\|_F^2} + \underbrace{\mathbb{E}\left[\mathbf{X}_j^\top \mathbf{M}\mathbf{X}_{\pi^{\natural 2}(i)}\mathbf{X}_{\pi^\natural(i)}^\top \mathbf{X}_{\pi^\natural(i)}\right]}_{p\mathbb{1}_{j=\pi^{\natural 2}(i)}\|\mathbf{B}^\natural\|_F^2}$$

$$+ \underbrace{\mathbb{E}\left[\mathbf{X}_j^\top \mathbf{M}\mathbf{X}_{\pi^\natural(j)}\mathbf{X}_j^\top \mathbf{X}_{\pi^\natural(i)}\right]}_{\mathbb{1}_{i=j}\|\mathbf{B}^\natural\|_F^2} + \underbrace{\mathbb{E}\left[\mathbf{X}_j^\top \mathbf{M}\mathbf{X}_j\mathbf{X}_{\pi^{\natural -1}(j)}^\top \mathbf{X}_{\pi^\natural(i)}\right]}_{p\mathbb{1}_{j=\pi^{\natural 2}(i)}\|\mathbf{B}^\natural\|_F^2}$$

$$= 2\left[p\mathbb{1}_{j=\pi^{\natural 2}(i)} + \mathbb{1}_{i=j}\right]\mathrm{Tr}(\mathbf{M}),$$

which suggests that

$$\mathbb{E}\Lambda_2 = -2(n-h)(p+1)\left(\mathbb{1}_{j=\pi^{\natural 2}(i)} + \mathbb{1}_{i=j}\right)\mathrm{Tr}(\mathbf{M}). \tag{53}$$

Combing (50), (51), (52), and (53), we conclude

$$\mathbb{E}\Lambda_2 = \frac{2p(n-h)^2}{n}\mathrm{Tr}(\mathbf{M})\left[1 + o(1)\right]. \tag{54}$$

**Step III.** Then we turn to the calculation of $\mathbb{E}\Lambda_3$. First we perform the following decomposition

$$\Lambda_3 = \underbrace{\mathbf{X}_{\pi^\natural(i)}^\top \mathbf{\Delta}\mathbf{M}\mathbf{\Delta}^\top \mathbf{X}_{\pi^\natural(i)}}_{\Lambda_{3,1}} + \underbrace{\mathbf{X}_j^\top \mathbf{\Delta}\mathbf{M}\mathbf{\Delta}^\top \mathbf{X}_j}_{\Lambda_{3,2}} - \underbrace{\mathbf{X}_{\pi^\natural(i)}^\top \mathbf{\Delta}\mathbf{M}\mathbf{\Delta}^\top \mathbf{X}_j}_{\Lambda_{3,3}} - \underbrace{\mathbf{X}_j^\top \mathbf{\Delta}\mathbf{M}\mathbf{\Delta}^\top \mathbf{X}_{\pi^\natural(i)}}_{\Lambda_{3,4}}.$$

**Case $(s,s)$: $i = \pi^\natural(i)$ and $j = \pi^\natural(j)$.** We have

$$\mathbb{E}\Lambda_{3,1} = \underbrace{\mathbb{E}\left(\mathbf{X}_i^\top \mathbf{X}_i\mathbf{X}_i^\top \mathbf{M}\mathbf{X}_i\mathbf{X}_i^\top \mathbf{X}_i\right)}_{\mathbb{E}\|\mathbf{X}_i\|_2^4\mathbf{X}_i^\top \mathbf{M}\mathbf{X}_i} + \underbrace{\mathbb{E}\left(\mathbf{X}_i^\top \mathbf{X}_i\mathbf{X}_i^\top \mathbf{M}\mathbf{X}_j\mathbf{X}_j^\top \mathbf{X}_i\right)}_{(p+2)\|\mathbf{B}^\natural\|_F^2}$$

$$+ \underbrace{\mathbb{E}\left(\mathbf{X}_i^\top \mathbf{X}_j\mathbf{X}_j^\top \mathbf{M}\mathbf{X}_i\mathbf{X}_i^\top \mathbf{X}_i\right)}_{(p+2)\|\mathbf{B}^\natural\|_F^2} + \underbrace{\mathbb{E}\left(\mathbf{X}_i^\top \mathbf{X}_j\mathbf{X}_j^\top \mathbf{M}\mathbf{X}_j\mathbf{X}_j^\top \mathbf{X}_i\right)}_{(p+2)\|\mathbf{B}^\natural\|_F^2} = (p+2)(p+7)\mathrm{Tr}(\mathbf{M});$$

$$\mathbb{E}\Lambda_{3,2} = \mathbb{E}\mathbf{X}_j^\top \left(\mathbf{X}_i\mathbf{X}_i^\top + \mathbf{X}_j\mathbf{X}_j^\top\right)\mathbf{M}\left(\mathbf{X}_i\mathbf{X}_i^\top + \mathbf{X}_j\mathbf{X}_j^\top\right)\mathbf{X}_j = (p+2)(p+7)\mathrm{Tr}(\mathbf{M}).$$

As for $\mathbb{E}\Lambda_{3,3}$ and $\mathbb{E}\Lambda_{3,4}$, easily we can verify that they are both zero and hence have

$$\mathbb{E}\Lambda_3 = 2(p+2)(p+7)\mathrm{Tr}(\mathbf{M}) = 2p^2\mathrm{Tr}(\mathbf{M})\left[1 + o(1)\right]. \tag{55}$$

**Case $(s,d)$: $i = \pi^\natural(i)$ and $j \neq \pi^\natural(j)$.** We can write $\Lambda_{3,1}$ as

$$\mathbb{E}\Lambda_{3,1} = \underbrace{\mathbb{E}\left(\mathbf{X}_i^\top \mathbf{X}_i\mathbf{X}_i^\top \mathbf{M}\mathbf{X}_i\mathbf{X}_i^\top \mathbf{X}_i\right)}_{\mathbb{E}\|\mathbf{X}_i\|_2^4\mathbf{X}_i^\top \mathbf{M}\mathbf{X}_i} + \underbrace{\mathbb{E}\left(\mathbf{X}_i^\top \mathbf{X}_i\mathbf{X}_i^\top \mathbf{M}\mathbf{X}_{\pi^\natural(j)}\mathbf{X}_j^\top \mathbf{X}_i\right)}_{0}$$

$$+ \underbrace{\mathbb{E}\left(\mathbf{X}_i^\top \mathbf{X}_i\mathbf{X}_i^\top \mathbf{M}\mathbf{X}_j\mathbf{X}_{\pi^{\natural -1}(j)}^\top \mathbf{X}_i\right)}_{0}$$

$$+ \underbrace{\mathbb{E}\left(\mathbf{X}_i^\top \mathbf{X}_j\mathbf{X}_{\pi^\natural(j)}^\top \mathbf{M}\mathbf{X}_i\mathbf{X}_i^\top \mathbf{X}_i\right)}_{0} + \underbrace{\mathbb{E}\left(\mathbf{X}_i^\top \mathbf{X}_j\mathbf{X}_{\pi^\natural(j)}^\top \mathbf{M}\mathbf{X}_{\pi^\natural(j)}\mathbf{X}_j^\top \mathbf{X}_i\right)}_{p\|\mathbf{B}^\natural\|_F^2}$$

$$+ \underbrace{\mathbb{E}\left(\mathbf{X}_i^\top \mathbf{X}_j\mathbf{X}_{\pi^\natural(j)}^\top \mathbf{M}\mathbf{X}_j\mathbf{X}_{\pi^{\natural -1}(j)}^\top \mathbf{X}_i\right)}_{\mathbb{1}_{j=\pi^{\natural 2}(j)}\mathrm{Tr}(\mathbf{M})}$$

$$+ \underbrace{\mathbb{E}\left(\mathbf{X}_i^\top \mathbf{X}_{\pi^{\natural -1}(j)}\mathbf{X}_j^\top \mathbf{M}\mathbf{X}_i\mathbf{X}_i^\top \mathbf{X}_i\right)}_{0} + \underbrace{\mathbb{E}\left(\mathbf{X}_i^\top \mathbf{X}_{\pi^{\natural -1}(j)}\mathbf{X}_j^\top \mathbf{M}\mathbf{X}_{\pi^\natural(j)}\mathbf{X}_j^\top \mathbf{X}_i\right)}_{\mathbb{1}_{j=\pi^{\natural 2}(j)}\mathrm{Tr}(\mathbf{M})}$$

$$+ \underbrace{\mathbb{E}\left(\mathbf{X}_i^\top \mathbf{X}_{\pi^{\natural -1}(j)}\mathbf{X}_j^\top \mathbf{M}\mathbf{X}_j\mathbf{X}_{\pi^{\natural -1}(j)}^\top \mathbf{X}_i\right)}_{p\|\mathbf{B}^\natural\|_F^2}$$

$$= \left(p^2 + 8p + 8 + 2\mathbb{1}_{j=\pi^{\natural 2}(j)}\right)\mathrm{Tr}(\mathbf{M}).$$

Mean $\Lambda_{3,2}$ can be written as

$$\mathbb{E}\Lambda_{3,2} = \underbrace{\mathbb{E}\left(\mathbf{X}_j^\top \mathbf{X}_i \mathbf{X}_i^\top \mathbf{M} \mathbf{X}_i \mathbf{X}_i^\top \mathbf{X}_j\right)}_{(p+2)\|\mathbf{B}^\natural\|_F^2} + \underbrace{\mathbb{E}\left(\mathbf{X}_j^\top \mathbf{X}_i \mathbf{X}_i^\top \mathbf{M} \mathbf{X}_{\pi^\natural(j)} \mathbf{X}_j^\top \mathbf{X}_j\right)}_{0}$$

$$+ \underbrace{\mathbb{E}\left(\mathbf{X}_j^\top \mathbf{X}_i \mathbf{X}_i^\top \mathbf{M} \mathbf{X}_j \mathbf{X}_{\pi^{\natural-1}(j)}^\top \mathbf{X}_j\right)}_{0}$$

$$+ \underbrace{\mathbb{E}\left(\mathbf{X}_j^\top \mathbf{X}_j \mathbf{X}_{\pi^\natural(j)}^\top \mathbf{M} \mathbf{X}_i \mathbf{X}_i^\top \mathbf{X}_j\right)}_{0} + \underbrace{\mathbb{E}\left(\mathbf{X}_j^\top \mathbf{X}_j \mathbf{X}_{\pi^\natural(j)}^\top \mathbf{M} \mathbf{X}_{\pi^\natural(j)} \mathbf{X}_j^\top \mathbf{X}_j\right)}_{\mathbb{E}\|\mathbf{X}_j\|_F^4 \operatorname{Tr}(\mathbf{M})}$$

$$+ \underbrace{\mathbb{E}\left(\mathbf{X}_j^\top \mathbf{X}_j \mathbf{X}_{\pi^\natural(j)}^\top \mathbf{M} \mathbf{X}_j \mathbf{X}_{\pi^{\natural-1}(j)}^\top \mathbf{X}_j\right)}_{\mathbb{1}_{\pi^{\natural-1}(j)=\pi^\natural(j)}(p+2)\|\mathbf{B}^\natural\|_F^2}$$

$$+ \underbrace{\mathbb{E}\left(\mathbf{X}_j^\top \mathbf{X}_{\pi^{\natural-1}(j)} \mathbf{X}_j^\top \mathbf{M} \mathbf{X}_i \mathbf{X}_i^\top \mathbf{X}_j\right)}_{0} + \underbrace{\mathbb{E}\left(\mathbf{X}_j^\top \mathbf{X}_{\pi^{\natural-1}(j)} \mathbf{X}_j^\top \mathbf{M} \mathbf{X}_{\pi^\natural(j)} \mathbf{X}_j^\top \mathbf{X}_j\right)}_{\mathbb{1}_{\pi^{\natural-1}(j)=\pi^\natural(j)}(p+2)\|\mathbf{B}^\natural\|_F^2}$$

$$+ \underbrace{\mathbb{E}\left(\mathbf{X}_j^\top \mathbf{X}_{\pi^{\natural-1}(j)} \mathbf{X}_j^\top \mathbf{M} \mathbf{X}_j \mathbf{X}_{\pi^{\natural-1}(j)}^\top \mathbf{X}_j\right)}_{(p+2)\|\mathbf{B}^\natural\|_F^2}$$

$$= (p+2)\left(p+2+2\mathbb{1}_{j=\pi^{\natural 2}(j)}\right)\operatorname{Tr}(\mathbf{M}).$$

And for $\mathbb{E}\Lambda_{3,3}$ and $\mathbb{E}\Lambda_{3,4}$, easily we can verify that they are both zero. Then we conclude

$$\mathbb{E}\Lambda_3 = 2\left(p^2+6p+6+(p+3)\mathbb{1}_{j=\pi^{\natural 2}(j)}\right)\operatorname{Tr}(\mathbf{M}) = 2p^2 \operatorname{Tr}(\mathbf{M})\left[1+o(1)\right]. \tag{56}$$

**Case $(d,s)$: $i \neq \pi^\natural(i)$ and $j = \pi^\natural(j)$.** In this case, we can write $\Lambda_{3,1}$ as

$$\mathbb{E}\Lambda_{3,1} = \underbrace{\mathbb{E}\left(\mathbf{X}_{\pi^\natural(i)}^\top \mathbf{X}_i \mathbf{X}_{\pi^\natural(i)}^\top \mathbf{M} \mathbf{X}_{\pi^\natural(i)} \mathbf{X}_i^\top \mathbf{X}_{\pi^\natural(i)}\right)}_{\mathbb{E}\|\mathbf{X}_i\|_F^2 \mathbf{X}_i^\top \mathbf{M} \mathbf{X}_i}$$

$$+ \underbrace{\mathbb{E}\left(\mathbf{X}_{\pi^\natural(i)}^\top \mathbf{X}_i \mathbf{X}_{\pi^\natural(i)}^\top \mathbf{M} \mathbf{X}_{\pi^{\natural 2}(i)} \mathbf{X}_{\pi^\natural(i)}^\top \mathbf{X}_{\pi^\natural(i)}\right)}_{\mathbb{1}_{i=\pi^{\natural 2}(i)}\mathbb{E}\|\mathbf{X}_i\|_F^2 \mathbf{X}_i^\top \mathbf{M} \mathbf{X}_i}$$

$$+ \underbrace{\mathbb{E}\left(\mathbf{X}_{\pi^\natural(i)}^\top \mathbf{X}_i \mathbf{X}_{\pi^\natural(i)}^\top \mathbf{M} \mathbf{X}_j \mathbf{X}_j^\top \mathbf{X}_{\pi^\natural(i)}\right)}_{0}$$

$$+ \underbrace{\mathbb{E}\left(\mathbf{X}_{\pi^\natural(i)}^\top \mathbf{X}_{\pi^\natural(i)} \mathbf{X}_{\pi^{\natural 2}(i)}^\top \mathbf{M} \mathbf{X}_{\pi^\natural(i)} \mathbf{X}_i^\top \mathbf{X}_{\pi^\natural(i)}\right)}_{\mathbb{1}_{i=\pi^{\natural 2}(i)}\mathbb{E}\|\mathbf{X}_i\|_F^2 \mathbf{X}_i^\top \mathbf{M} \mathbf{X}_i}$$

$$+ \underbrace{\mathbb{E}\left(\mathbf{X}_{\pi^\natural(i)}^\top \mathbf{X}_{\pi^\natural(i)} \mathbf{X}_{\pi^{\natural 2}(i)}^\top \mathbf{M} \mathbf{X}_{\pi^{\natural 2}(i)} \mathbf{X}_{\pi^\natural(i)}^\top \mathbf{X}_{\pi^\natural(i)}\right)}_{\mathbb{E}\|\mathbf{X}_i\|_F^4 \operatorname{Tr}(\mathbf{M})}$$

$$+ \underbrace{\mathbb{E}\left(\mathbf{X}_{\pi^\natural(i)}^\top \mathbf{X}_{\pi^\natural(i)} \mathbf{X}_{\pi^{\natural 2}(i)}^\top \mathbf{M} \mathbf{X}_j \mathbf{X}_j^\top \mathbf{X}_{\pi^\natural(i)}\right)}_{0}$$

$$+ \underbrace{\mathbb{E}\left(\mathbf{X}_{\pi^\natural(i)}^\top \mathbf{X}_j \mathbf{X}_j^\top \mathbf{M} \mathbf{X}_{\pi^\natural(i)} \mathbf{X}_i^\top \mathbf{X}_{\pi^\natural(i)}\right)}_{0}$$

$$+ \underbrace{\mathbb{E}\left(\mathbf{X}_{\pi^\natural(i)}^\top \mathbf{X}_j \mathbf{X}_j^\top \mathbf{M} \mathbf{X}_{\pi^{\natural 2}(i)} \mathbf{X}_{\pi^\natural(i)}^\top \mathbf{X}_{\pi^\natural(i)}\right)}_{0}$$

$$+ \underbrace{\mathbb{E}\left(\mathbf{X}_{\pi^\natural(i)}^\top \mathbf{X}_j \mathbf{X}_j^\top \mathbf{M} \mathbf{X}_j \mathbf{X}_j^\top \mathbf{X}_{\pi^\natural(i)}\right)}_{\mathbb{E}\|\mathbf{X}_i\|_F^2 \mathbf{X}_i^\top \mathbf{M} \mathbf{X}_i}$$

$$= (p+2)\left(p + 2 + 2\mathbb{1}_{i=\pi^{\natural 2}(i)}\right)\operatorname{Tr}(\mathbf{M}).$$

We consider $\Lambda_{3,2}$ as

$$\mathbb{E}\Lambda_{3,2} = \underbrace{\mathbb{E}\mathbf{X}_j^\top \mathbf{X}_i \mathbf{X}_{\pi^\natural(i)}^\top \mathbf{M}\mathbf{X}_{\pi^\natural(i)}\mathbf{X}_i^\top \mathbf{X}_j}_{p\operatorname{Tr}(\mathbf{M})} + \underbrace{\mathbb{E}\mathbf{X}_j^\top \mathbf{X}_i \mathbf{X}_{\pi^\natural(i)}^\top \mathbf{M}\mathbf{X}_{\pi^{\natural 2}(i)}\mathbf{X}_{\pi^\natural(i)}^\top \mathbf{X}_j}_{\mathbb{1}_{i=\pi^{\natural 2}(i)}\operatorname{Tr}(\mathbf{M})}$$

$$+ \underbrace{\mathbb{E}\mathbf{X}_j^\top \mathbf{X}_i \mathbf{X}_{\pi^\natural(i)}^\top \mathbf{M}\mathbf{X}_j \mathbf{X}_j^\top \mathbf{X}_j}_{0}$$

$$+ \underbrace{\mathbb{E}\mathbf{X}_j^\top \mathbf{X}_{\pi^\natural(i)}\mathbf{X}_{\pi^{\natural 2}(i)}^\top \mathbf{M}\mathbf{X}_{\pi^\natural(i)}\mathbf{X}_i^\top \mathbf{X}_j}_{\mathbb{1}_{i=\pi^{\natural 2}(i)}\operatorname{Tr}(\mathbf{M})} + \underbrace{\mathbb{E}\mathbf{X}_j^\top \mathbf{X}_{\pi^\natural(i)}\mathbf{X}_{\pi^{\natural 2}(i)}^\top \mathbf{M}\mathbf{X}_{\pi^{\natural 2}(i)}\mathbf{X}_{\pi^\natural(i)}^\top \mathbf{X}_j}_{p\operatorname{Tr}(\mathbf{M})}$$

$$+ \underbrace{\mathbb{E}\mathbf{X}_j^\top \mathbf{X}_{\pi^\natural(i)}\mathbf{X}_{\pi^{\natural 2}(i)}^\top \mathbf{M}\mathbf{X}_j \mathbf{X}_j^\top \mathbf{X}_j}_{0}$$

$$+ \underbrace{\mathbb{E}\mathbf{X}_j^\top \mathbf{X}_j \mathbf{X}_j^\top \mathbf{M}\mathbf{X}_{\pi^\natural(i)}\mathbf{X}_i^\top \mathbf{X}_j}_{0} + \underbrace{\mathbb{E}\mathbf{X}_j^\top \mathbf{X}_j \mathbf{X}_j^\top \mathbf{M}\mathbf{X}_{\pi^{\natural 2}(i)}\mathbf{X}_{\pi^\natural(i)}^\top \mathbf{X}_j}_{0} + \underbrace{\mathbb{E}\mathbf{X}_j^\top \mathbf{X}_j \mathbf{X}_j^\top \mathbf{M}\mathbf{X}_j \mathbf{X}_j^\top \mathbf{X}_j}_{\mathbb{E}\|\mathbf{X}_i\|_{\mathrm{F}}^4 \mathbf{X}_i^\top \mathbf{M}\mathbf{X}_i}$$

$$= \left(p^2 + 8p + 8 + 2\mathbb{1}_{i=\pi^{\natural 2}(i)}\right)\operatorname{Tr}(\mathbf{M}).$$

Similarly, as above, we can verify that $\mathbb{E}\Lambda_{3,3} = 0$ and $\mathbb{E}\Lambda_{3,4} = 0$. Hence, we can conclude

$$\mathbb{E}\Lambda_3 = 2\left(p^2 + 6p + 6 + (p+3)\mathbb{1}_{i=\pi^{\natural 2}(i)}\right)\operatorname{Tr}(\mathbf{M}) = 2p^2 \operatorname{Tr}(\mathbf{M})\left[1 + o(1)\right]. \tag{57}$$

**Case $(d,d)$: $i \neq \pi^\natural(i)$ and $j \neq \pi^\natural(j)$.** We write $\Lambda_{3,1}$ as

$$\mathbb{E}\Lambda_{3,1} = \underbrace{\mathbb{E}\,\mathbf{X}_{\pi^\natural(i)}^\top \mathbf{X}_i \mathbf{X}_{\pi^\natural(i)}^\top \mathbf{M}\mathbf{X}_{\pi^\natural(i)}\mathbf{X}_i^\top \mathbf{X}_{\pi^\natural(i)}}_{\mathbb{E}\|\mathbf{X}_i\|_{\mathrm{F}}^2 \mathbf{X}_i^\top \mathbf{M}\mathbf{X}_i} + \underbrace{\mathbb{E}\,\mathbf{X}_{\pi^\natural(i)}^\top \mathbf{X}_i \mathbf{X}_{\pi^\natural(i)}^\top \mathbf{M}\mathbf{X}_{\pi^{\natural 2}(i)}\mathbf{X}_{\pi^\natural(i)}^\top \mathbf{X}_{\pi^\natural(i)}}_{\mathbb{1}_{i=\pi^{\natural 2}(i)}\mathbb{E}\|\mathbf{X}_i\|_{\mathrm{F}}^2 \mathbf{X}_i^\top \mathbf{M}\mathbf{X}_i}$$

$$+ \underbrace{\mathbb{E}\,\mathbf{X}_{\pi^\natural(i)}^\top \mathbf{X}_i \mathbf{X}_{\pi^\natural(i)}^\top \mathbf{M}\mathbf{X}_{\pi^\natural(j)}\mathbf{X}_j^\top \mathbf{X}_{\pi^\natural(i)}}_{\mathbb{1}_{i=j}\mathbb{E}\|\mathbf{X}_i\|_{\mathrm{F}}^2 \mathbf{X}_i^\top \mathbf{M}\mathbf{X}_i} + \underbrace{\mathbb{E}\,\mathbf{X}_{\pi^\natural(i)}^\top \mathbf{X}_i \mathbf{X}_{\pi^\natural(i)}^\top \mathbf{M}\mathbf{X}_j \mathbf{X}_{\pi^{\natural -1}(j)}^\top \mathbf{X}_{\pi^\natural(i)}}_{\mathbb{1}_{i=j}\mathbb{1}_{i=\pi^{\natural 2}(i)}\mathbb{E}\|\mathbf{X}_i\|_{\mathrm{F}}^2 \mathbf{X}_i^\top \mathbf{M}\mathbf{X}_i}$$

$$+ \underbrace{\mathbb{E}\,\mathbf{X}_{\pi^\natural(i)}^\top \mathbf{X}_{\pi^\natural(i)}\mathbf{X}_{\pi^{\natural 2}(i)}^\top \mathbf{M}\mathbf{X}_{\pi^\natural(i)}\mathbf{X}_i^\top \mathbf{X}_{\pi^\natural(i)}}_{\mathbb{1}_{i=\pi^{\natural 2}(i)}\mathbb{E}\|\mathbf{X}_i\|_{\mathrm{F}}^2 \mathbf{X}_i^\top \mathbf{M}\mathbf{X}_i} + \underbrace{\mathbb{E}\,\mathbf{X}_{\pi^\natural(i)}^\top \mathbf{X}_{\pi^\natural(i)}\mathbf{X}_{\pi^{\natural 2}(i)}^\top \mathbf{M}\mathbf{X}_{\pi^{\natural 2}(i)}\mathbf{X}_{\pi^\natural(i)}^\top \mathbf{X}_{\pi^\natural(i)}}_{\mathbb{E}\|\mathbf{X}_i\|_{\mathrm{F}}^4 \operatorname{Tr}(\mathbf{M})}$$

$$+ \underbrace{\mathbb{E}\,\mathbf{X}_{\pi^\natural(i)}^\top \mathbf{X}_{\pi^\natural(i)}\mathbf{X}_{\pi^{\natural 2}(i)}^\top \mathbf{M}\mathbf{X}_{\pi^\natural(j)}\mathbf{X}_j^\top \mathbf{X}_{\pi^\natural(i)}}_{\mathbb{1}_{i=j}\mathbb{1}_{i=\pi^{\natural 2}(i)}\mathbb{E}\|\mathbf{X}_i\|_{\mathrm{F}}^2 \mathbf{X}_i^\top \mathbf{M}\mathbf{X}_i} + \underbrace{\mathbb{E}\,\mathbf{X}_{\pi^\natural(i)}^\top \mathbf{X}_{\pi^\natural(i)}\mathbf{X}_{\pi^{\natural 2}(i)}^\top \mathbf{M}\mathbf{X}_j \mathbf{X}_{\pi^{\natural -1}(j)}^\top \mathbf{X}_{\pi^\natural(i)}}_{\mathbb{1}_{j=\pi^{\natural 2}(i)}\mathbb{E}\|\mathbf{X}_i\|_{\mathrm{F}}^4 \operatorname{Tr}(\mathbf{M})}$$

$$+ \underbrace{\mathbb{E}\,\mathbf{X}_{\pi^\natural(i)}^\top \mathbf{X}_j \mathbf{X}_{\pi^\natural(j)}^\top \mathbf{M}\mathbf{X}_{\pi^\natural(i)}\mathbf{X}_i^\top \mathbf{X}_{\pi^\natural(i)}}_{\mathbb{1}_{i=j}\mathbb{E}\|\mathbf{X}_i\|_{\mathrm{F}}^2 \mathbf{X}_i^\top \mathbf{M}\mathbf{X}_i} + \underbrace{\mathbb{E}\,\mathbf{X}_{\pi^\natural(i)}^\top \mathbf{X}_j \mathbf{X}_{\pi^\natural(j)}^\top \mathbf{M}\mathbf{X}_{\pi^{\natural 2}(i)}\mathbf{X}_{\pi^\natural(i)}^\top \mathbf{X}_{\pi^\natural(i)}}_{\mathbb{1}_{i=j}\mathbb{1}_{i=\pi^{\natural 2}(i)}\mathbb{E}\|\mathbf{X}_i\|_{\mathrm{F}}^2 \mathbf{X}_i^\top \mathbf{M}\mathbf{X}_i}$$

$$+ \underbrace{\mathbb{E}\,\mathbf{X}_{\pi^\natural(i)}^\top \mathbf{X}_j \mathbf{X}_{\pi^\natural(j)}^\top \mathbf{M}\mathbf{X}_{\pi^\natural(j)}\mathbf{X}_j^\top \mathbf{X}_{\pi^\natural(i)}}_{\mathbb{1}_{i=j}\mathbb{E}\|\mathbf{X}_i\|_{\mathrm{F}}^2 \mathbf{X}_i^\top \mathbf{M}\mathbf{X}_i + \mathbb{1}_{i\neq j}p\operatorname{Tr}(\mathbf{M})} + \underbrace{\mathbb{E}\,\mathbf{X}_{\pi^\natural(i)}^\top \mathbf{X}_j \mathbf{X}_{\pi^\natural(j)}^\top \mathbf{M}\mathbf{X}_j \mathbf{X}_{\pi^{\natural -1}(j)}^\top \mathbf{X}_{\pi^\natural(i)}}_{\mathbb{1}_{j=\pi^{\natural 2}(j)}\left(\mathbb{1}_{i=j}\mathbb{E}\|\mathbf{X}_i\|_{\mathrm{F}}^2 \mathbf{X}_i^\top \mathbf{M}\mathbf{X}_i + \mathbb{1}_{i\neq j}\operatorname{Tr}(\mathbf{M})\right)}$$

$$+ \underbrace{\mathbb{E}\,\mathbf{X}_{\pi^\natural(i)}^\top \mathbf{X}_{\pi^{\natural -1}(j)}\mathbf{X}_j^\top \mathbf{M}\mathbf{X}_{\pi^\natural(i)}\mathbf{X}_i^\top \mathbf{X}_{\pi^\natural(i)}}_{\mathbb{1}_{i=j}\mathbb{1}_{i=\pi^{\natural 2}(i)}\mathbb{E}\|\mathbf{X}_i\|_{\mathrm{F}}^2 \mathbf{X}_i^\top \mathbf{M}\mathbf{X}_i} + \underbrace{\mathbb{E}\,\mathbf{X}_{\pi^\natural(i)}^\top \mathbf{X}_{\pi^{\natural -1}(j)}\mathbf{X}_j^\top \mathbf{M}\mathbf{X}_{\pi^{\natural 2}(i)}\mathbf{X}_{\pi^\natural(i)}^\top \mathbf{X}_{\pi^\natural(i)}}_{\mathbb{1}_{j=\pi^{\natural 2}(i)}\mathbb{E}\|\mathbf{X}_i\|_{\mathrm{F}}^4 \operatorname{Tr}(\mathbf{M})}$$

$$+ \underbrace{\mathbb{E}\,\mathbf{X}_{\pi^\natural(i)}^\top \mathbf{X}_{\pi^{\natural -1}(j)}\mathbf{X}_j^\top \mathbf{M}\mathbf{X}_{\pi^\natural(j)}\mathbf{X}_j^\top \mathbf{X}_{\pi^\natural(i)}}_{\mathbb{1}_{j=\pi^{\natural 2}(j)}\left[\mathbb{1}_{i=j}\mathbb{E}\|\mathbf{X}_i\|_{\mathrm{F}}^2 \mathbf{X}_i^\top \mathbf{M}\mathbf{X}_i + \mathbb{1}_{i\neq j}\operatorname{Tr}(\mathbf{M})\right]} + \underbrace{\mathbb{E}\,\mathbf{X}_{\pi^\natural(i)}^\top \mathbf{X}_{\pi^{\natural -1}(j)}\mathbf{X}_j^\top \mathbf{M}\mathbf{X}_j \mathbf{X}_{\pi^{\natural -1}(j)}^\top \mathbf{X}_{\pi^\natural(i)}}_{\mathbb{1}_{j=\pi^{\natural 2}(i)}\mathbb{E}\|\mathbf{X}_i\|_{\mathrm{F}}^4 \operatorname{Tr}(\mathbf{M}) + \mathbb{1}_{j\neq \pi^{\natural 2}(i)}p\operatorname{Tr}(\mathbf{M})}$$

$$= \left(p^2 + 5p + 2\right)\operatorname{Tr}(\mathbf{M}) + \mathbb{1}_{i=\pi^{\natural 2}(i)}2(p+2)\operatorname{Tr}(\mathbf{M}) + \mathbb{1}_{j=\pi^{\natural 2}(i)}\left(3p^2 + 5p\right)\operatorname{Tr}(\mathbf{M})$$

$$+ 2\mathbb{1}_{j=\pi^{\natural 2}(j)}\operatorname{Tr}(\mathbf{M}) + \mathbb{1}_{i=j}2(p+3)\operatorname{Tr}(\mathbf{M}) + \mathbb{1}_{i=j}\mathbb{1}_{i=\pi^{\natural 2}(i)}2(3p+5)\operatorname{Tr}(\mathbf{M}).$$

We consider $\Lambda_{3,2}$ as

$$\mathbb{E}\Lambda_{3,2} = \underbrace{\mathbb{E}\mathbf{X}_j^\top \mathbf{X}_i \mathbf{X}_{\pi^\natural(i)}^\top \mathbf{M}\mathbf{X}_{\pi^\natural(i)}\mathbf{X}_i^\top \mathbf{X}_j}_{\mathbb{1}_{i=j}\mathbb{E}\|\mathbf{X}_i\|_{\mathrm{F}}^4 \operatorname{Tr}(\mathbf{M}) + \mathbb{1}_{i\neq j}p\operatorname{Tr}(\mathbf{M})} + \underbrace{\mathbb{E}\mathbf{X}_j^\top \mathbf{X}_i \mathbf{X}_{\pi^\natural(i)}^\top \mathbf{M}\mathbf{X}_{\pi^{\natural 2}(i)}\mathbf{X}_{\pi^\natural(i)}^\top \mathbf{X}_j}_{\mathbb{1}_{i=\pi^{\natural 2}(i)}\left[\mathbb{1}_{i=j}\mathbb{E}\|\mathbf{X}_i\|_{\mathrm{F}}^2 \mathbf{X}_i^\top \mathbf{M}\mathbf{X}_i + \mathbb{1}_{i\neq j}\operatorname{Tr}(\mathbf{M})\right]}$$

$$
+ \underbrace{\mathbb{E}\mathbf{X}_j^\top \mathbf{X}_i \mathbf{X}_{\pi^\natural(i)}^\top \mathbf{M} \mathbf{X}_{\pi^\natural(j)} \mathbf{X}_j^\top \mathbf{X}_j}_{\mathbb{1}_{i=j}\mathbb{E}\|\mathbf{X}_i\|_{\mathrm F}^4 \operatorname{Tr}(\mathbf{M})} + \underbrace{\mathbb{E}\mathbf{X}_j^\top \mathbf{X}_i \mathbf{X}_{\pi^\natural(i)}^\top \mathbf{M} \mathbf{X}_j \mathbf{X}_{\pi^{\natural-1}(j)}^\top \mathbf{X}_j}_{\mathbb{1}_{i=j}\mathbb{1}_{i=\pi^{\natural 2}(i)}\mathbb{E}\|\mathbf{X}_i\|_{\mathrm F}^2 \mathbf{X}_i^\top \mathbf{M}\mathbf{X}_i}
$$

$$
+ \underbrace{\mathbb{E}\mathbf{X}_j^\top \mathbf{X}_{\pi^\natural(i)} \mathbf{X}_{\pi^{\natural 2}(i)}^\top \mathbf{M} \mathbf{X}_{\pi^\natural(i)} \mathbf{X}_i^\top \mathbf{X}_j}_{\mathbb{1}_{i=\pi^{\natural 2}(i)}\left[\mathbb{1}_{i=j}\mathbb{E}\|\mathbf{X}_i\|_{\mathrm F}^2 \mathbf{X}_i^\top \mathbf{M}\mathbf{X}_i + \mathbb{1}_{i\neq j}\operatorname{Tr}(\mathbf{M})\right]} + \underbrace{\mathbb{E}\mathbf{X}_j^\top \mathbf{X}_{\pi^\natural(i)} \mathbf{X}_{\pi^{\natural 2}(i)}^\top \mathbf{M} \mathbf{X}_{\pi^{\natural 2}(i)} \mathbf{X}_{\pi^\natural(i)}^\top \mathbf{X}_j}_{\mathbb{1}_{j=\pi^{\natural 2}(i)}\mathbb{E}\|\mathbf{X}_i\|_{\mathrm F}^2 \mathbf{X}_i^\top \mathbf{M}\mathbf{X}_i + \mathbb{1}_{j\neq\pi^{\natural 2}(i)}p\operatorname{Tr}(\mathbf{M})}
$$

$$
+ \underbrace{\mathbb{E}\mathbf{X}_j^\top \mathbf{X}_{\pi^\natural(i)} \mathbf{X}_{\pi^{\natural 2}(i)}^\top \mathbf{M} \mathbf{X}_{\pi^\natural(j)} \mathbf{X}_j^\top \mathbf{X}_j}_{\mathbb{1}_{i=j}\mathbb{1}_{i=\pi^{\natural 2}(i)}\mathbb{E}\|\mathbf{X}_i\|_{\mathrm F}^2 \mathbf{X}_i^\top \mathbf{M}\mathbf{X}_i} + \underbrace{\mathbb{E}\mathbf{X}_j^\top \mathbf{X}_{\pi^\natural(i)} \mathbf{X}_{\pi^{\natural 2}(i)}^\top \mathbf{M} \mathbf{X}_j \mathbf{X}_{\pi^{\natural-1}(j)}^\top \mathbf{X}_j}_{\mathbb{1}_{j=\pi^{\natural 2}(i)}\mathbb{E}\|\mathbf{X}_i\|_{\mathrm F}^2 \mathbf{X}_i^\top \mathbf{M}\mathbf{X}_i}
$$

$$
+ \underbrace{\mathbb{E}\mathbf{X}_j^\top \mathbf{X}_j \mathbf{X}_{\pi^\natural(j)}^\top \mathbf{M} \mathbf{X}_{\pi^\natural(i)} \mathbf{X}_i^\top \mathbf{X}_j}_{\mathbb{1}_{i=j}\mathbb{E}\|\mathbf{X}_i\|_{\mathrm F}^4 \operatorname{Tr}(\mathbf{M})} + \underbrace{\mathbb{E}\mathbf{X}_j^\top \mathbf{X}_j \mathbf{X}_{\pi^\natural(j)}^\top \mathbf{M} \mathbf{X}_{\pi^{\natural 2}(i)} \mathbf{X}_{\pi^\natural(i)}^\top \mathbf{X}_j}_{\mathbb{1}_{i=j}\mathbb{1}_{i=\pi^{\natural 2}(i)}\mathbb{E}\|\mathbf{X}_i\|_{\mathrm F}^2 \mathbf{X}_i^\top \mathbf{M}\mathbf{X}_i}
$$

$$
+ \underbrace{\mathbb{E}\mathbf{X}_j^\top \mathbf{X}_j \mathbf{X}_{\pi^\natural(j)}^\top \mathbf{M} \mathbf{X}_{\pi^\natural(j)} \mathbf{X}_j^\top \mathbf{X}_j}_{\mathbb{E}\|\mathbf{X}_i\|_{\mathrm F}^4 \operatorname{Tr}(\mathbf{M})} + \underbrace{\mathbb{E}\mathbf{X}_j^\top \mathbf{X}_j \mathbf{X}_{\pi^\natural(j)}^\top \mathbf{M} \mathbf{X}_j \mathbf{X}_{\pi^{\natural-1}(j)}^\top \mathbf{X}_j}_{\mathbb{1}_{j=\pi^{\natural 2}(j)}\mathbb{E}\|\mathbf{X}_i\|_{\mathrm F}^2 \mathbf{X}_i^\top \mathbf{M}\mathbf{X}_i}
$$

$$
+ \underbrace{\mathbb{E}\mathbf{X}_j^\top \mathbf{X}_{\pi^{\natural-1}(j)} \mathbf{X}_j^\top \mathbf{M} \mathbf{X}_{\pi^\natural(i)} \mathbf{X}_i^\top \mathbf{X}_j}_{\mathbb{1}_{i=j}\mathbb{1}_{i=\pi^{\natural 2}(i)}\mathbb{E}\|\mathbf{X}_i\|_{\mathrm F}^2 \mathbf{X}_i^\top \mathbf{M}\mathbf{X}_i} + \underbrace{\mathbb{E}\mathbf{X}_j^\top \mathbf{X}_{\pi^{\natural-1}(j)} \mathbf{X}_j^\top \mathbf{M} \mathbf{X}_{\pi^{\natural 2}(i)} \mathbf{X}_{\pi^\natural(i)}^\top \mathbf{X}_j}_{\mathbb{1}_{j=\pi^{\natural 2}(i)}\mathbb{E}\|\mathbf{X}_i\|_{\mathrm F}^2 \mathbf{X}_i^\top \mathbf{M}\mathbf{X}_i}
$$

$$
+ \underbrace{\mathbb{E}\mathbf{X}_j^\top \mathbf{X}_{\pi^{\natural-1}(j)} \mathbf{X}_j^\top \mathbf{M} \mathbf{X}_{\pi^\natural(j)} \mathbf{X}_j^\top \mathbf{X}_j}_{\mathbb{1}_{j=\pi^{\natural 2}(j)}\mathbb{E}\|\mathbf{X}_i\|_{\mathrm F}^2 \mathbf{X}_i^\top \mathbf{M}\mathbf{X}_i} + \underbrace{\mathbb{E}\mathbf{X}_j^\top \mathbf{X}_{\pi^{\natural-1}(j)} \mathbf{X}_j^\top \mathbf{M} \mathbf{X}_j \mathbf{X}_{\pi^{\natural-1}(j)}^\top \mathbf{X}_j}_{\mathbb{E}\|\mathbf{X}_i\|_{\mathrm F}^2 \mathbf{X}_i^\top \mathbf{M}\mathbf{X}_i}
$$

$$
= \left(p^2 + 5p + 2\right)\operatorname{Tr}(\mathbf{M}) + \mathbb{1}_{j=\pi^{\natural 2}(j)}2(p+2)\operatorname{Tr}(\mathbf{M}) + \mathbb{1}_{i=j}\left(3p^2 + 5p\right)\operatorname{Tr}(\mathbf{M})
$$
$$
+ 2\mathbb{1}_{i=\pi^{\natural 2}(i)}\operatorname{Tr}(\mathbf{M}) + \mathbb{1}_{j=\pi^{\natural 2}(i)}2(p+3)\operatorname{Tr}(\mathbf{M}) + \mathbb{1}_{i=j}\mathbb{1}_{i=\pi^{\natural 2}(i)}2\left(3p+5\right)\operatorname{Tr}(\mathbf{M}).
$$

We consider $\Lambda_{3,3}$ as

$$
\mathbb{E}\Lambda_{3,3} = \underbrace{\mathbb{E}\,\mathbf{X}_{\pi^\natural(i)}^\top \mathbf{X}_i \mathbf{X}_{\pi^\natural(i)}^\top \mathbf{M} \mathbf{X}_{\pi^\natural(i)} \mathbf{X}_i^\top \mathbf{X}_j}_{0} + \underbrace{\mathbb{E}\mathbf{X}_{\pi^\natural(i)}^\top \mathbf{X}_i \mathbf{X}_{\pi^\natural(i)}^\top \mathbf{M} \mathbf{X}_{\pi^{\natural 2}(i)} \mathbf{X}_{\pi^\natural(i)}^\top \mathbf{X}_j}_{0}
$$

$$
+ \underbrace{\mathbb{E}\,\mathbf{X}_{\pi^\natural(i)}^\top \mathbf{X}_i \mathbf{X}_{\pi^\natural(i)}^\top \mathbf{M} \mathbf{X}_{\pi^\natural(j)} \mathbf{X}_j^\top \mathbf{X}_j}_{\mathbb{1}_{i=\pi^\natural(j)}p\operatorname{Tr}(\mathbf{M})} + \underbrace{\mathbb{E}\,\mathbf{X}_{\pi^\natural(i)}^\top \mathbf{X}_i \mathbf{X}_{\pi^\natural(i)}^\top \mathbf{M} \mathbf{X}_j \mathbf{X}_{\pi^{\natural-1}(j)}^\top \mathbf{X}_j}_{0}
$$

$$
+ \underbrace{\mathbb{E}\,\mathbf{X}_{\pi^\natural(i)}^\top \mathbf{X}_{\pi^\natural(i)} \mathbf{X}_{\pi^{\natural 2}(i)}^\top \mathbf{M} \mathbf{X}_{\pi^\natural(i)} \mathbf{X}_i^\top \mathbf{X}_j}_{0} + \underbrace{\mathbb{E}\,\mathbf{X}_{\pi^\natural(i)}^\top \mathbf{X}_{\pi^\natural(i)} \mathbf{X}_{\pi^{\natural 2}(i)}^\top \mathbf{M} \mathbf{X}_{\pi^{\natural 2}(i)} \mathbf{X}_{\pi^\natural(i)}^\top \mathbf{X}_j}_{0}
$$

$$
+ \underbrace{\mathbb{E}\,\mathbf{X}_{\pi^\natural(i)}^\top \mathbf{X}_{\pi^\natural(i)} \mathbf{X}_{\pi^{\natural 2}(i)}^\top \mathbf{M} \mathbf{X}_{\pi^\natural(j)} \mathbf{X}_j^\top \mathbf{X}_j}_{0} + \underbrace{\mathbb{E}\,\mathbf{X}_{\pi^\natural(i)}^\top \mathbf{X}_{\pi^\natural(i)} \mathbf{X}_{\pi^{\natural 2}(i)}^\top \mathbf{M} \mathbf{X}_j \mathbf{X}_{\pi^{\natural-1}(j)}^\top \mathbf{X}_j}_{\mathbb{1}_{j=\pi^{\natural 3}(i)}p\operatorname{Tr}(\mathbf{M})}
$$

$$
+ \underbrace{\mathbb{E}\,\mathbf{X}_{\pi^\natural(i)}^\top \mathbf{X}_j \mathbf{X}_{\pi^\natural(j)}^\top \mathbf{M} \mathbf{X}_{\pi^\natural(i)} \mathbf{X}_i^\top \mathbf{X}_j}_{\mathbb{1}_{i=\pi^\natural(j)}\operatorname{Tr}(\mathbf{M})} + \underbrace{\mathbb{E}\,\mathbf{X}_{\pi^\natural(i)}^\top \mathbf{X}_j \mathbf{X}_{\pi^\natural(j)}^\top \mathbf{M} \mathbf{X}_{\pi^{\natural 2}(i)} \mathbf{X}_{\pi^\natural(i)}^\top \mathbf{X}_j}_{0}
$$

$$
+ \underbrace{\mathbb{E}\,\mathbf{X}_{\pi^\natural(i)}^\top \mathbf{X}_j \mathbf{X}_{\pi^\natural(j)}^\top \mathbf{M} \mathbf{X}_{\pi^\natural(j)} \mathbf{X}_j^\top \mathbf{X}_j}_{0} + \underbrace{\mathbb{E}\,\mathbf{X}_{\pi^\natural(i)}^\top \mathbf{X}_j \mathbf{X}_{\pi^\natural(j)}^\top \mathbf{M} \mathbf{X}_j \mathbf{X}_{\pi^{\natural-1}(j)}^\top \mathbf{X}_j}_{0}
$$

$$
+ \underbrace{\mathbb{E}\,\mathbf{X}_{\pi^\natural(i)}^\top \mathbf{X}_{\pi^{\natural-1}(j)} \mathbf{X}_j^\top \mathbf{M} \mathbf{X}_{\pi^\natural(i)} \mathbf{X}_i^\top \mathbf{X}_j}_{0} + \underbrace{\mathbb{E}\,\mathbf{X}_{\pi^\natural(i)}^\top \mathbf{X}_{\pi^{\natural-1}(j)} \mathbf{X}_j^\top \mathbf{M} \mathbf{X}_{\pi^{\natural 2}(i)} \mathbf{X}_{\pi^\natural(i)}^\top \mathbf{X}_j}_{\mathbb{1}_{j=\pi^{\natural 3}(i)}\operatorname{Tr}(\mathbf{M})}
$$

$$
+ \underbrace{\mathbb{E}\,\mathbf{X}_{\pi^\natural(i)}^\top \mathbf{X}_{\pi^{\natural-1}(j)} \mathbf{X}_j^\top \mathbf{M} \mathbf{X}_{\pi^\natural(j)} \mathbf{X}_j^\top \mathbf{X}_j}_{0} + \underbrace{\mathbb{E}\,\mathbf{X}_{\pi^\natural(i)}^\top \mathbf{X}_{\pi^{\natural-1}(j)} \mathbf{X}_j^\top \mathbf{M} \mathbf{X}_j \mathbf{X}_{\pi^{\natural-1}(j)}^\top \mathbf{X}_j}_{0}
$$

$$
= (p+1)\left[\mathbb{1}_{i=\pi^\natural(j)} + \mathbb{1}_{j=\pi^{\natural 3}(i)}\right]\operatorname{Tr}(\mathbf{M}).
$$

Then we consider $\Lambda_{3,4}$ as

$$
\mathbb{E}\Lambda_{3,4} = \underbrace{\mathbb{E}\mathbf{X}_j^\top \mathbf{X}_i \mathbf{X}_{\pi^\natural(i)}^\top \mathbf{M} \mathbf{X}_{\pi^\natural(i)} \mathbf{X}_i^\top \mathbf{X}_{\pi^\natural(i)}}_{0} + \underbrace{\mathbb{E}\mathbf{X}_j^\top \mathbf{X}_i \mathbf{X}_{\pi^\natural(i)}^\top \mathbf{M} \mathbf{X}_{\pi^{\natural 2}(i)} \mathbf{X}_{\pi^\natural(i)}^\top \mathbf{X}_{\pi^\natural(i)}}_{0}
$$

$$+ \underbrace{\mathbb{E}\mathbf{X}_j^\top \mathbf{X}_i \mathbf{X}_{\pi^\natural(j)}^\top \mathbf{M}\mathbf{X}_{\pi^\natural(j)} \mathbf{X}_j^\top \mathbf{X}_{\pi^\natural(i)}}_{\mathbb{1}_{i=\pi^\natural(j)}\,\mathrm{Tr}(\mathbf{M})} + \underbrace{\mathbb{E}\mathbf{X}_j^\top \mathbf{X}_i \mathbf{X}_{\pi^\natural(j)}^\top \mathbf{M}\mathbf{X}_j \mathbf{X}_{\pi^{\natural-1}(j)}^\top \mathbf{X}_{\pi^\natural(i)}}_{0}$$

$$+ \underbrace{\mathbb{E}\mathbf{X}_j^\top \mathbf{X}_{\pi^\natural(i)} \mathbf{X}_{\pi^{\natural 2}(i)}^\top \mathbf{M}\mathbf{X}_{\pi^\natural(i)} \mathbf{X}_i^\top \mathbf{X}_{\pi^\natural(i)}}_{0} + \underbrace{\mathbb{E}\mathbf{X}_j^\top \mathbf{X}_{\pi^\natural(i)} \mathbf{X}_{\pi^{\natural 2}(i)}^\top \mathbf{M}\mathbf{X}_{\pi^{\natural 2}(i)} \mathbf{X}_{\pi^\natural(i)}^\top \mathbf{X}_{\pi^\natural(i)}}_{0}$$

$$+ \underbrace{\mathbb{E}\mathbf{X}_j^\top \mathbf{X}_{\pi^\natural(i)} \mathbf{X}_{\pi^{\natural 2}(i)}^\top \mathbf{M}\mathbf{X}_{\pi^\natural(j)} \mathbf{X}_j^\top \mathbf{X}_{\pi^\natural(i)}}_{0} + \underbrace{\mathbb{E}\mathbf{X}_j^\top \mathbf{X}_{\pi^\natural(i)} \mathbf{X}_{\pi^{\natural 2}(i)}^\top \mathbf{M}\mathbf{X}_j \mathbf{X}_{\pi^{\natural-1}(j)}^\top \mathbf{X}_{\pi^\natural(i)}}_{\mathbb{1}_{j=\pi^{\natural 3}(i)}\,\mathrm{Tr}(\mathbf{M})}$$

$$+ \underbrace{\mathbb{E}\mathbf{X}_j^\top \mathbf{X}_j \mathbf{X}_{\pi^\natural(j)}^\top \mathbf{M}\mathbf{X}_{\pi^\natural(i)} \mathbf{X}_i^\top \mathbf{X}_{\pi^\natural(i)}}_{p\mathbb{1}_{i=\pi^\natural(j)}\,\mathrm{Tr}(\mathbf{M})} + \underbrace{\mathbb{E}\mathbf{X}_j^\top \mathbf{X}_j \mathbf{X}_{\pi^\natural(j)}^\top \mathbf{M}\mathbf{X}_{\pi^{\natural 2}(i)} \mathbf{X}_{\pi^\natural(i)}^\top \mathbf{X}_{\pi^\natural(i)}}_{0}$$

$$+ \underbrace{\mathbb{E}\mathbf{X}_j^\top \mathbf{X}_j \mathbf{X}_{\pi^\natural(j)}^\top \mathbf{M}\mathbf{X}_{\pi^\natural(j)} \mathbf{X}_j^\top \mathbf{X}_{\pi^\natural(i)}}_{0} + \underbrace{\mathbb{E}\mathbf{X}_j^\top \mathbf{X}_j \mathbf{X}_{\pi^\natural(j)}^\top \mathbf{M}\mathbf{X}_j \mathbf{X}_{\pi^{\natural-1}(j)}^\top \mathbf{X}_{\pi^\natural(i)}}_{0}$$

$$+ \underbrace{\mathbb{E}\mathbf{X}_j^\top \mathbf{X}_{\pi^{\natural-1}(j)} \mathbf{X}_j^\top \mathbf{M}\mathbf{X}_{\pi^\natural(i)} \mathbf{X}_i^\top \mathbf{X}_{\pi^\natural(i)}}_{0} + \underbrace{\mathbb{E}\mathbf{X}_j^\top \mathbf{X}_{\pi^{\natural-1}(j)} \mathbf{X}_j^\top \mathbf{M}\mathbf{X}_{\pi^{\natural 2}(i)} \mathbf{X}_{\pi^\natural(i)}^\top \mathbf{X}_{\pi^\natural(i)}}_{p\mathbb{1}_{j=\pi^{\natural 3}(i)}\,\mathrm{Tr}(\mathbf{M})}$$

$$+ \underbrace{\mathbb{E}\mathbf{X}_j^\top \mathbf{X}_{\pi^{\natural-1}(j)} \mathbf{X}_j^\top \mathbf{M}\mathbf{X}_{\pi^\natural(j)} \mathbf{X}_j^\top \mathbf{X}_{\pi^\natural(i)}}_{0} + \underbrace{\mathbb{E}\mathbf{X}_j^\top \mathbf{X}_{\pi^{\natural-1}(j)} \mathbf{X}_j^\top \mathbf{M}\mathbf{X}_j \mathbf{X}_{\pi^{\natural-1}(j)}^\top \mathbf{X}_{\pi^\natural(i)}}_{0}$$

$$= (p+1)\left(\mathbb{1}_{i=\pi^\natural(j)} + \mathbb{1}_{j=\pi^{\natural 3}(i)}\right)\mathrm{Tr}(\mathbf{M}).$$

In summary, we have

$$\begin{aligned}
\mathbb{E}\Lambda_3 =\ & 2\left(p^2 + 5p + 2\right)\mathrm{Tr}(\mathbf{M}) + 2(p+3)\left[\mathbb{1}_{i=\pi^{\natural 2}(i)} + \mathbb{1}_{j=\pi^{\natural 2}(j)}\right]\mathrm{Tr}(\mathbf{M}) \\
& + \left(3p^2 + 7p + 6\right)\left(\mathbb{1}_{i=j} + \mathbb{1}_{j=\pi^{\natural 2}(i)}\right)\mathrm{Tr}(\mathbf{M}) + \mathbb{1}_{i=j}\mathbb{1}_{i=\pi^{\natural 2}(i)}4\left(3p+5\right)\mathrm{Tr}(\mathbf{M}) \\
& - 2(p+1)\left[\mathbb{1}_{i=\pi^\natural(j)} + \mathbb{1}_{j=\pi^{\natural 3}(i)}\right]\mathrm{Tr}(\mathbf{M}) = 2p^2\,\mathrm{Tr}(\mathbf{M})\left[1 + o(1)\right].
\end{aligned} \tag{58}$$

Combining (55), (56), (57), and (58) then yields

$$\mathbb{E}\Lambda_3 = 2p^2\,\mathrm{Tr}(\mathbf{M})\left[1 + o(1)\right]. \tag{59}$$

The proof is then completed by (48), (49), (54), and (59).

$$\square$$

**Lemma 8.** *We have*

$$\begin{aligned}
\mathbb{E}\Xi_4^2 =\ & m(m+1)\left[p\left(p+2\right)\left(\mathbb{1}_{i=\pi^\natural(i)} + \mathbb{1}_{i=j}\right) + p\left(\mathbb{1}_{i\neq\pi^\natural(i)} + \mathbb{1}_{i\neq j}\right)\right] + 2mp(n+p+1) \\
=\ & \frac{(n-h)m^2 p^2}{n}\left[1 + o(1)\right],
\end{aligned}$$

*where $\Xi_4$ is defined in* (21).

*Proof.* For the conciseness of notation, we define $\boldsymbol{\Gamma}$ as $\mathbf{X}\left(\mathbf{X}_{\pi^\natural(i)} - \mathbf{X}_j\right)\left(\mathbf{X}_{\pi^\natural(i)} - \mathbf{X}_j\right)^\top \mathbf{X}^\top$ and hence have

$$\mathbb{E}\Xi_4^2 = \mathbb{E}\mathbf{W}_i^\top \mathbf{W}^\top \boldsymbol{\Gamma}\mathbf{W}\mathbf{W}_i.$$

We begin the discussion by expanding $\mathbf{W}\mathbf{W}_i$ as

$$\begin{bmatrix} \mathbf{W}_1^\top \\ \mathbf{W}_2^\top \\ \cdots \\ \mathbf{W}_n^\top \end{bmatrix}\mathbf{W}_i = \begin{bmatrix} \mathbf{W}_1^\top \mathbf{W}_i \\ \mathbf{W}_2^\top \mathbf{W}_i \\ \cdots \\ \mathbf{W}_n^\top \mathbf{W}_i \end{bmatrix}.$$

Then we obtain

$$\mathbb{E}\Xi_4^2 = \sum_{s=1}^n \sum_{t=1}^n \Gamma_{ij}\mathbb{E}\left[\left(\mathbf{W}_s^\top \mathbf{W}_i\right)\left(\mathbf{W}_t^\top \mathbf{W}_i\right)\right] = \Gamma_{ii}\mathbb{E}\left(\mathbf{W}_i^\top \mathbf{W}_i\right)^2 + \underbrace{\sum_{s\neq i}\sum_{t\neq i}\Gamma_{st}\mathbb{E}\left(\mathbf{W}_s^\top \mathbf{W}_i \mathbf{W}_t^\top \mathbf{W}_i\right)}_{\sum_{s\neq i}\Gamma_{ss}\mathbb{E}(\mathbf{W}_s^\top \mathbf{W}_i)^2}$$

$$
= \Gamma_{ii}\mathbb{E}\left(\sum_{j=1}^{m}W_{ij}^2\right)^2 + \sum_{s\neq i}\Gamma_{ss}\cdot m = m(m+1)\mathbb{E}\Gamma_{ii} + m\mathbb{E}\operatorname{Tr}(\mathbf{\Gamma}). \tag{60}
$$

We can thus complete the proof by separately computing $\mathbb{E}\operatorname{Tr}(\mathbf{M})$ and $\mathbb{E}\Gamma_{ii}$. First we compute $\mathbb{E}\Gamma_{ii}$, which proceeds as

$$
\begin{aligned}
\mathbb{E}\Gamma_{ii} &= \mathbb{E}\left(\mathbf{X}_i^\top \mathbf{X}_{\pi^\natural(i)}\right)^2 + \mathbb{E}\left(\mathbf{X}_i^\top \mathbf{X}_j\right)^2\\
&= \mathbb{1}_{i=\pi^\natural(i)}p(p+2) + \mathbb{1}_{i\neq\pi^\natural(i)}p + \mathbb{1}_{i=j}p(p+2) + \mathbb{1}_{i\neq j}p\\
&= p\,(p+2)\left[\mathbb{1}_{i=\pi^\natural(i)} + \mathbb{1}_{i=j}\right] + p\left[\mathbb{1}_{i\neq\pi^\natural(i)} + \mathbb{1}_{i\neq j}\right]. \tag{61}
\end{aligned}
$$

Then we turn to the computation of $\mathbb{E}\operatorname{Tr}(\mathbf{M})$, which proceeds as

$$
\begin{aligned}
\mathbb{E}\operatorname{Tr}(\mathbf{\Gamma}) &= \left\|\!\left|\mathbf{X}\left(\mathbf{X}_{\pi^\natural(i)} - \mathbf{X}_j\right)\right\|\!\right\|_{\mathrm{F}}^2\\
&= \mathbb{E}\left\|\mathbf{X}_{\pi^\natural(i)}^\top\left(\mathbf{X}_{\pi^\natural(i)} - \mathbf{X}_j\right)\right\|_2^2 + \mathbb{E}\left\|\mathbf{X}_j^\top\left(\mathbf{X}_{\pi^\natural(i)} - \mathbf{X}_j\right)\right\|_2^2 + \sum_{s\neq\pi^\natural(i),j}\mathbb{E}\left\|\mathbf{X}_s^\top\left(\mathbf{X}_{\pi^\natural(i)} - \mathbf{X}_j\right)\right\|_2^2\\
&= 2\mathbb{E}\left\|\mathbf{X}_{\pi^\natural(i)}\right\|_2^4 + 2\mathbb{E}\left(\mathbf{X}_{\pi^\natural(i)}^\top \mathbf{X}_j\right)^2 + 2\sum_{s\neq\pi^\natural(i),j}\mathbb{E}\|\mathbf{X}_s\|_2^2\\
&= 2p(p+3) + 2(n-2)p = 2p(n+p+1). \tag{62}
\end{aligned}
$$

The proof is thus completed by combing (60), (62), and (61). $\qquad\square$

**Lemma 9.** *We have*

$$
\mathbb{E}\Xi_1\Xi_4 = \frac{mp(n-h)(n+p-h)}{n}\left[1+o(1)\right]\operatorname{Tr}(\mathbf{M}),
$$

*where $\Xi_1$ and $\Xi_4$ are defined in (21).*

*Proof.* We have

$$
\mathbb{E}\Xi_1\Xi_4 = \mathbb{E}\underbrace{\mathbf{X}_{\pi^\natural(i)}^\top \mathbf{M}\mathbf{X}^\top\mathbf{\Pi}^{\natural\top}\mathbf{X}\left(\mathbf{X}_{\pi^\natural(i)} - \mathbf{X}_j\right)\left(\mathbf{X}_{\pi^\natural(i)} - \mathbf{X}_j\right)^\top \mathbf{X}^\top}_{\triangleq \boldsymbol{v}^\top}\mathbf{W}\mathbf{W}_i.
$$

First we conditional on $\mathbf{X}$. Expanding the product $\mathbf{W}\mathbf{W}_i$ as

$$
\mathbb{E}\begin{bmatrix}\mathbf{W}_1^\top \mathbf{W}_i\\ \mathbf{W}_2^\top \mathbf{W}_i\\ \cdots\\ \mathbf{W}_i^\top \mathbf{W}_i\\ \cdots\\ \mathbf{W}_n^\top \mathbf{W}_i\end{bmatrix} = \begin{bmatrix}0\\ 0\\ \cdots\\ m\\ \cdots\\ 0\end{bmatrix},
$$

we can compute $\mathbb{E}\Xi_1\Xi_2$ w.r.t. $\mathbf{W}$ as

$$
\mathbb{E}\left(\Xi_1\Xi_2\right) = \mathbb{E}\boldsymbol{v}^\top\mathbf{W}\mathbf{W}_i = m\mathbb{E}\boldsymbol{v}_i,
$$

where $\boldsymbol{v}_i$ denotes the $i$-th entry of $\boldsymbol{v}$ and can be written as

$$
\boldsymbol{v}_i = \underbrace{\mathbf{X}_i^\top\left(\mathbf{X}_{\pi^\natural(i)} - \mathbf{X}_j\right)\left(\mathbf{X}_{\pi^\natural(i)} - \mathbf{X}_j\right)^\top \mathbf{\Sigma}\mathbf{M}\mathbf{X}_{\pi^\natural(i)}}_{\Lambda_1} + \underbrace{\mathbf{X}_i^\top\left(\mathbf{X}_{\pi^\natural(i)} - \mathbf{X}_j\right)\left(\mathbf{X}_{\pi^\natural(i)} - \mathbf{X}_j\right)^\top \mathbf{\Delta}\mathbf{M}\mathbf{X}_{\pi^\natural(i)}}_{\Lambda_2}.
$$

For $\Lambda_1$, we conclude

$$
\begin{aligned}
\mathbb{E}\Lambda_1 &= \mathbb{E}\mathbf{X}_i^\top \mathbf{X}_{\pi^\natural(i)}\mathbf{X}_{\pi^\natural(i)}^\top \mathbf{\Sigma}\mathbf{M}\mathbf{X}_{\pi^\natural(i)} + \mathbb{E}\mathbf{X}_i^\top \mathbf{X}_j\mathbf{X}_j^\top \mathbf{\Sigma}\mathbf{M}\mathbf{X}_{\pi^\natural(i)}\\
&\quad - \mathbb{E}\mathbf{X}_i^\top \mathbf{X}_{\pi^\natural(i)}\mathbf{X}_j^\top \mathbf{\Sigma}\mathbf{M}\mathbf{X}_{\pi^\natural(i)} - \mathbb{E}\mathbf{X}_i^\top \mathbf{X}_j\mathbf{X}_{\pi^\natural(i)}^\top \mathbf{\Sigma}\mathbf{M}\mathbf{X}_{\pi^\natural(i)}\\
&= \mathbb{1}_{i=\pi^\natural(i)}(p+3)\mathbb{E}\operatorname{Tr}(\mathbf{\Sigma}\mathbf{M}) - \mathbb{1}_{i=j}(p+1)\mathbb{E}\operatorname{Tr}(\mathbf{\Sigma}\mathbf{M})
\end{aligned}
$$

$$= (n-h)\left(\mathbb{1}_{i=\pi^{\natural}(i)}(p+3) - \mathbb{1}_{i=j}(p+1)\right)\mathrm{Tr}(\mathbf{M}) = \frac{p(n-h)^2}{n}\left[1+o(1)\right]\mathrm{Tr}(\mathbf{M}).$$

Then we turn to $\mathbb{E}\Lambda_2$ and obtain

$$\mathbb{E}\Lambda_2 = \mathbb{E}\underbrace{\mathbf{X}_i^{\top}\mathbf{X}_{\pi^{\natural}(i)}\mathbf{X}_{\pi^{\natural}(i)}^{\top}\boldsymbol{\Delta}\mathbf{M}\mathbf{X}_{\pi^{\natural}(i)}}_{\Lambda_{2,1}} + \mathbb{E}\underbrace{\mathbf{X}_i^{\top}\mathbf{X}_j\mathbf{X}_j^{\top}\boldsymbol{\Delta}\mathbf{M}\mathbf{X}_{\pi^{\natural}(i)}}_{\Lambda_{2,2}}$$

$$- \mathbb{E}\underbrace{\mathbf{X}_i^{\top}\mathbf{X}_{\pi^{\natural}(i)}\mathbf{X}_j^{\top}\boldsymbol{\Delta}\mathbf{M}\mathbf{X}_{\pi^{\natural}(i)}}_{\Lambda_{2,3}} - \mathbb{E}\underbrace{\mathbf{X}_i^{\top}\mathbf{X}_j\mathbf{X}_{\pi^{\natural}(i)}^{\top}\boldsymbol{\Delta}\mathbf{M}\mathbf{X}_{\pi^{\natural}(i)}}_{\Lambda_{2,4}}.$$

We compute the value of $\mathbb{E}\Lambda_2$ under the four different cases.

**Case $(s,s)$:** $i = \pi^{\natural}(i)$ **and** $j = \pi^{\natural}(j)$**.** In this case, we have $\boldsymbol{\Delta}$ be

$$\boldsymbol{\Delta} = \boldsymbol{\Delta}^{(s,s)} = \mathbf{X}_i\mathbf{X}_i^{\top} + \mathbf{X}_j\mathbf{X}_j^{\top}.$$

We have

$$\mathbb{E}\Lambda_{2,1} = \mathbb{E}\mathbf{X}_i^{\top}\mathbf{X}_i\mathbf{X}_i^{\top}\left(\mathbf{X}_i\mathbf{X}_i^{\top} + \mathbf{X}_j\mathbf{X}_j^{\top}\right)\mathbf{M}\mathbf{X}_i = (p+2)(p+5)\,\mathrm{Tr}(\mathbf{M}),$$
$$\mathbb{E}\Lambda_{2,2} = \mathbb{E}\mathbf{X}_i^{\top}\mathbf{X}_j\mathbf{X}_j^{\top}\left(\mathbf{X}_i\mathbf{X}_i^{\top} + \mathbf{X}_j\mathbf{X}_j^{\top}\right)\mathbf{M}\mathbf{X}_i = 2(p+2)\,\mathrm{Tr}(\mathbf{M}),$$
$$\mathbb{E}\Lambda_{2,3} = \mathbb{E}\mathbf{X}_i^{\top}\mathbf{X}_i\mathbf{X}_j^{\top}\left(\mathbf{X}_i\mathbf{X}_i^{\top} + \mathbf{X}_j\mathbf{X}_j^{\top}\right)\mathbf{M}\mathbf{X}_i = 0,$$
$$\mathbb{E}\Lambda_{2,4} = \mathbb{E}\mathbf{X}_i^{\top}\mathbf{X}_j\mathbf{X}_i^{\top}\left(\mathbf{X}_i\mathbf{X}_i^{\top} + \mathbf{X}_j\mathbf{X}_j^{\top}\right)\mathbf{M}\mathbf{X}_i = 0,$$

which implies

$$\mathbb{E}\Lambda_2 = (p+2)(p+7)\,\mathrm{Tr}(\mathbf{M}).$$

**Case $(s,d)$:** $i = \pi^{\natural}(i)$ **and** $j \neq \pi^{\natural}(j)$**.** First we write $\boldsymbol{\Delta}$ as

$$\boldsymbol{\Delta}^{(s,d)} = \mathbf{X}_i\mathbf{X}_i^{\top} + \mathbf{X}_j\mathbf{X}_{\pi^{\natural}(j)}^{\top} + \mathbf{X}_{\pi^{\natural-1}(j)}\mathbf{X}_j^{\top}.$$

Then we conclude

$$\mathbb{E}\Lambda_{2,1} = \mathbb{E}\mathbf{X}_i^{\top}\mathbf{X}_i\mathbf{X}_i^{\top}\left(\mathbf{X}_i\mathbf{X}_i^{\top} + \mathbf{X}_j\mathbf{X}_{\pi^{\natural}(j)}^{\top} + \mathbf{X}_{\pi^{\natural-1}(j)}\mathbf{X}_j^{\top}\right)\mathbf{M}\mathbf{X}_i = (p+2)(p+4)\,\mathrm{Tr}(\mathbf{M}),$$

$$\mathbb{E}\Lambda_{2,2} = \mathbb{E}\mathbf{X}_i^{\top}\mathbf{X}_j\mathbf{X}_j^{\top}\left(\mathbf{X}_i\mathbf{X}_i^{\top} + \mathbf{X}_j\mathbf{X}_{\pi^{\natural}(j)}^{\top} + \mathbf{X}_{\pi^{\natural-1}(j)}\mathbf{X}_j^{\top}\right)\mathbf{M}\mathbf{X}_i = (p+2)\,\mathrm{Tr}(\mathbf{M}),$$

$$\mathbb{E}\Lambda_{2,3} = \mathbb{E}\mathbf{X}_i^{\top}\mathbf{X}_i\mathbf{X}_j^{\top}\left(\mathbf{X}_i\mathbf{X}_i^{\top} + \mathbf{X}_j\mathbf{X}_{\pi^{\natural}(j)}^{\top} + \mathbf{X}_{\pi^{\natural-1}(j)}\mathbf{X}_j^{\top}\right)\mathbf{M}\mathbf{X}_i = 0,$$

$$\mathbb{E}\Lambda_{2,4} = \mathbb{E}\mathbf{X}_i^{\top}\mathbf{X}_j\mathbf{X}_i^{\top}\left(\mathbf{X}_i\mathbf{X}_i^{\top} + \mathbf{X}_j\mathbf{X}_{\pi^{\natural}(j)}^{\top} + \mathbf{X}_{\pi^{\natural-1}(j)}\mathbf{X}_j^{\top}\right)\mathbf{M}\mathbf{X}_i = 0,$$

which suggests that

$$\mathbb{E}\Lambda_2 = (p+2)(p+5)\,\mathrm{Tr}(\mathbf{M}).$$

**Case $(d,s)$:** $i \neq \pi^{\natural}(i)$ **and** $j = \pi^{\natural}(j)$**.** In this case, $\boldsymbol{\Delta}$ reduces to

$$\boldsymbol{\Delta}^{(d,s)} = \mathbf{X}_i\mathbf{X}_{\pi^{\natural}(i)}^{\top} + \mathbf{X}_{\pi^{\natural}(i)}\mathbf{X}_{\pi^{\natural2}(i)}^{\top} + \mathbf{X}_j\mathbf{X}_j^{\top}.$$

We have

$$\mathbb{E}\Lambda_{2,1} = \underbrace{\mathbb{E}\mathbf{X}_i^{\top}\mathbf{X}_{\pi^{\natural}(i)}\mathbf{X}_{\pi^{\natural}(i)}^{\top}\mathbf{X}_i\mathbf{X}_{\pi^{\natural}(i)}^{\top}\mathbf{M}\mathbf{X}_{\pi^{\natural}(i)}}_{\mathbb{E}\|\mathbf{X}_i\|_2^2\mathbf{X}_i^{\top}\mathbf{M}\mathbf{X}_i} + \underbrace{\mathbb{E}\mathbf{X}_i^{\top}\mathbf{X}_{\pi^{\natural}(i)}\mathbf{X}_{\pi^{\natural}(i)}^{\top}\mathbf{X}_{\pi^{\natural}(i)}\mathbf{X}_{\pi^{\natural2}(i)}^{\top}\mathbf{M}\mathbf{X}_{\pi^{\natural}(i)}}_{\mathbb{1}_{i=\pi^{\natural2}(i)}\mathbb{E}\|\mathbf{X}_i\|_2^2\mathbf{X}_i^{\top}\mathbf{M}\mathbf{X}_i}$$

$$+ \underbrace{\mathbb{E}\mathbf{X}_i^{\top}\mathbf{X}_{\pi^{\natural}(i)}\mathbf{X}_{\pi^{\natural}(i)}^{\top}\mathbf{X}_j\mathbf{X}_j^{\top}\mathbf{M}\mathbf{X}_{\pi^{\natural}(i)}}_{0} = (p+2)\left[1 + \mathbb{1}_{i=\pi^{\natural2}(i)}\right]\mathrm{Tr}(\mathbf{M}),$$

$$\mathbb{E}\Lambda_{2,2} = \underbrace{\mathbb{E}\mathbf{X}_i^{\top}\mathbf{X}_j\mathbf{X}_j^{\top}\mathbf{X}_i\mathbf{X}_{\pi^{\natural}(i)}^{\top}\mathbf{M}\mathbf{X}_{\pi^{\natural}(i)}}_{p\,\mathrm{Tr}(\mathbf{M})} + \underbrace{\mathbb{E}\mathbf{X}_i^{\top}\mathbf{X}_j\mathbf{X}_j^{\top}\mathbf{X}_{\pi^{\natural}(i)}\mathbf{X}_{\pi^{\natural2}(i)}^{\top}\mathbf{M}\mathbf{X}_{\pi^{\natural}(i)}}_{\mathbb{1}_{i=\pi^{\natural2}(i)}\,\mathrm{Tr}(\mathbf{M})}$$

$$+ \underbrace{\mathbb{E}\mathbf{X}_i^\top \mathbf{X}_j \mathbf{X}_j^\top \mathbf{X}_j \mathbf{X}_j^\top \mathbf{M}\mathbf{X}_{\pi^\natural(i)}}_{0} = \left(p + \mathbb{1}_{i=\pi^{\natural 2}(i)}\right) \mathrm{Tr}(\mathbf{M}),$$

$$\mathbb{E}\Lambda_{2,3} = 0,$$
$$\mathbb{E}\Lambda_{2,4} = 0,$$

which suggests

$$\mathbb{E}\Lambda_2 = 2(p+1)\,\mathrm{Tr}(\mathbf{M}) + \mathbb{1}_{i=\pi^{\natural 2}(i)}\,(p+3)\,\mathrm{Tr}(\mathbf{M}).$$

**Case $(d,d)$:** $i \neq \pi^\natural(i)$ **and** $j \neq \pi^\natural(j)$. In this case, $\boldsymbol{\Delta}$ is written as

$$\boldsymbol{\Delta}^{(d,d)} = \mathbf{X}_i\mathbf{X}_{\pi^\natural(i)}^\top + \mathbf{X}_{\pi^\natural(i)}\mathbf{X}_{\pi^{\natural 2}(i)}^\top + \mathbf{X}_j\mathbf{X}_{\pi^\natural(j)}^\top + \mathbf{X}_{\pi^{\natural-1}(j)}\mathbf{X}_j^\top.$$

We have

$$\mathbb{E}\Lambda_{2,1} = \underbrace{\mathbb{E}\mathbf{X}_i^\top \mathbf{X}_{\pi^\natural(i)}\mathbf{X}_{\pi^\natural(i)}^\top \mathbf{X}_i\mathbf{X}_{\pi^\natural(i)}^\top \mathbf{M}\mathbf{X}_{\pi^\natural(i)}}_{\mathbb{E}\|\mathbf{X}_i\|_2^2 \mathbf{X}_i^\top \mathbf{M}\mathbf{X}_i} + \underbrace{\mathbb{E}\mathbf{X}_i^\top \mathbf{X}_{\pi^\natural(i)}\mathbf{X}_{\pi^\natural(i)}^\top \mathbf{X}_{\pi^\natural(i)}\mathbf{X}_{\pi^{\natural 2}(i)}^\top \mathbf{M}\mathbf{X}_{\pi^\natural(i)}}_{\mathbb{1}_{i=\pi^{\natural 2}(i)}\mathbb{E}\|\mathbf{X}_i\|_2^2 \mathbf{X}_i^\top \mathbf{M}\mathbf{X}_i}$$

$$+ \underbrace{\mathbb{E}\mathbf{X}_i^\top \mathbf{X}_{\pi^\natural(i)}\mathbf{X}_{\pi^\natural(i)}^\top \mathbf{X}_j\mathbf{X}_{\pi^\natural(j)}^\top \mathbf{M}\mathbf{X}_{\pi^\natural(i)}}_{\mathbb{1}_{i=j}\mathbb{E}\|\mathbf{X}_i\|_2^2 \mathbf{X}_i^\top \mathbf{M}\mathbf{X}_i} + \underbrace{\mathbb{E}\mathbf{X}_i^\top \mathbf{X}_{\pi^\natural(i)}\mathbf{X}_{\pi^\natural(i)}^\top \mathbf{X}_{\pi^{\natural-1}(j)}\mathbf{X}_j^\top \mathbf{M}\mathbf{X}_{\pi^\natural(i)}}_{\mathbb{1}_{i=j}\mathbb{1}_{i=\pi^{\natural 2}(i)}\mathbb{E}\|\mathbf{X}_i\|_2^2 \mathbf{X}_i^\top \mathbf{M}\mathbf{X}_i};$$

$$\mathbb{E}\Lambda_{2,2} = \underbrace{\mathbb{E}\mathbf{X}_i^\top \mathbf{X}_j\mathbf{X}_j^\top \mathbf{X}_i\mathbf{X}_{\pi^\natural(i)}^\top \mathbf{M}\mathbf{X}_{\pi^\natural(i)}}_{\mathbb{1}_{i=j}p(p+2)\,\mathrm{Tr}(\mathbf{M})+\mathbb{1}_{i\neq j}p\,\mathrm{Tr}(\mathbf{M})} + \underbrace{\mathbb{E}\mathbf{X}_i^\top \mathbf{X}_j\mathbf{X}_j^\top \mathbf{X}_{\pi^\natural(i)}\mathbf{X}_{\pi^{\natural 2}(i)}^\top \mathbf{M}\mathbf{X}_{\pi^\natural(i)}}_{\mathbb{1}_{i=\pi^{\natural 2}(i)}\left[\mathbb{1}_{i=j}\mathbb{E}\|\mathbf{X}_i\|_2^2 \mathbf{X}_i^\top \mathbf{M}\mathbf{X}_i+\mathbb{1}_{i\neq j}\,\mathrm{Tr}(\mathbf{M})\right]}$$

$$+ \underbrace{\mathbb{E}\mathbf{X}_i^\top \mathbf{X}_j\mathbf{X}_j^\top \mathbf{X}_j\mathbf{X}_{\pi^\natural(j)}^\top \mathbf{M}\mathbf{X}_{\pi^\natural(i)}}_{\mathbb{1}_{i=j}p(p+2)\,\mathrm{Tr}(\mathbf{M})} + \underbrace{\mathbb{E}\mathbf{X}_i^\top \mathbf{X}_j\mathbf{X}_j^\top \mathbf{X}_{\pi^{\natural-1}(j)}\mathbf{X}_j^\top \mathbf{M}\mathbf{X}_{\pi^\natural(i)}}_{\mathbb{1}_{i=j}\mathbb{1}_{j=\pi^{\natural 2}(i)}\mathbb{E}\|\mathbf{X}\|_2^2\mathbf{X}^\top \mathbf{M}\mathbf{X}};$$

$$\mathbb{E}\Lambda_{2,3} = \underbrace{\mathbb{E}\mathbf{X}_i^\top \mathbf{X}_{\pi^\natural(i)}\mathbf{X}_j^\top \mathbf{X}_i\mathbf{X}_{\pi^\natural(i)}^\top \mathbf{M}\mathbf{X}_{\pi^\natural(i)}}_{0} + \underbrace{\mathbb{E}\mathbf{X}_i^\top \mathbf{X}_{\pi^\natural(i)}\mathbf{X}_j^\top \mathbf{X}_{\pi^\natural(i)}\mathbf{X}_{\pi^{\natural 2}(i)}^\top \mathbf{M}\mathbf{X}_{\pi^\natural(i)}}_{0}$$

$$+ \underbrace{\mathbb{E}\mathbf{X}_i^\top \mathbf{X}_{\pi^\natural(i)}\mathbf{X}_j^\top \mathbf{X}_j\mathbf{X}_{\pi^\natural(j)}^\top \mathbf{M}\mathbf{X}_{\pi^\natural(i)}}_{\mathbb{1}_{i=\pi^\natural(j)}p\,\mathrm{Tr}(\mathbf{M})} + \underbrace{\mathbb{E}\mathbf{X}_i^\top \mathbf{X}_{\pi^\natural(i)}\mathbf{X}_j^\top \mathbf{X}_{\pi^{\natural-1}(j)}\mathbf{X}_j^\top \mathbf{M}\mathbf{X}_{\pi^\natural(i)}}_{0};$$

$$\mathbb{E}\Lambda_{2,4} = \underbrace{\mathbb{E}\mathbf{X}_i^\top \mathbf{X}_j\mathbf{X}_{\pi^\natural(i)}^\top \mathbf{X}_i\mathbf{X}_{\pi^\natural(i)}^\top \mathbf{M}\mathbf{X}_{\pi^\natural(i)}}_{0} + \underbrace{\mathbb{E}\mathbf{X}_i^\top \mathbf{X}_j\mathbf{X}_{\pi^\natural(i)}^\top \mathbf{X}_{\pi^\natural(i)}\mathbf{X}_{\pi^{\natural 2}(i)}^\top \mathbf{M}\mathbf{X}_{\pi^\natural(i)}}_{0}$$

$$+ \underbrace{\mathbb{E}\mathbf{X}_i^\top \mathbf{X}_j\mathbf{X}_{\pi^\natural(i)}^\top \mathbf{X}_j\mathbf{X}_{\pi^\natural(j)}^\top \mathbf{M}\mathbf{X}_{\pi^\natural(i)}}_{\mathbb{1}_{i=\pi^\natural(j)}\,\mathrm{Tr}(\mathbf{M})} + \underbrace{\mathbb{E}\mathbf{X}_i^\top \mathbf{X}_j\mathbf{X}_{\pi^\natural(i)}^\top \mathbf{X}_{\pi^{\natural-1}(j)}\mathbf{X}_j^\top \mathbf{M}\mathbf{X}_{\pi^\natural(i)}}_{0}.$$

Hence we conclude

$$\mathbb{E}\Lambda_2 = 2(p+1)\,\mathrm{Tr}(\mathbf{M}) + \mathbb{1}_{i=j}2\,(p+1)^2\,\mathrm{Tr}(\mathbf{M}) + \mathbb{1}_{i=\pi^\natural(j)}(p+1)\,\mathrm{Tr}(\mathbf{M})$$
$$+ \mathbb{1}_{i=\pi^{\natural 2}(i)}(p+3)\,\mathrm{Tr}(\mathbf{M}) + \mathbb{1}_{i=j}\mathbb{1}_{i=\pi^{\natural 2}(i)}(3p+5)\,\mathrm{Tr}(\mathbf{M}).$$

$\square$

**Lemma 10.** *We have*

$$\mathbb{E}\Xi_2\Xi_3 = \frac{p(n-h)(n+p-h)}{n}\,\mathrm{Tr}(\mathbf{M})\left[1+o(1)\right],$$

*where $\Xi_2$ and $\Xi_3$ are defined in* (21).

*Proof.* To start with, we write the expectation as $\mathbb{E}\Xi_2\Xi_3$

$$\mathbb{E}\Xi_2\Xi_3 = \mathbb{E}\underbrace{\mathbf{X}_{\pi^\natural(i)}^\top \mathbf{B}^\natural \mathbf{W}^\top}_{\boldsymbol{u}^\top} \underbrace{\mathbf{X}\left(\mathbf{X}_{\pi^\natural(i)} - \mathbf{X}_j\right)}_{\boldsymbol{p}\in\mathbb{R}^{n\times 1}} \mathbf{W}_i^\top \underbrace{\mathbf{B}^{\natural\top}\mathbf{X}^\top \boldsymbol{\Pi}^{\natural\top}\mathbf{X}\left(\mathbf{X}_{\pi^\natural(i)} - \mathbf{X}_j\right)}_{\boldsymbol{v}}$$

$$= \mathbb{E}\boldsymbol{u}^\top \mathbf{W}^\top \boldsymbol{p}\mathbf{W}_i^\top \boldsymbol{v} = \mathbb{E}\left\langle \mathbf{W}_i, \boldsymbol{u}^\top \mathbf{W}^\top \boldsymbol{p}\boldsymbol{v}\right\rangle.$$

Exploiting the independence among $\mathbf{X}$ and $\mathbf{W}$, we condition on $\mathbf{X}$ and have

$$\mathbb{E}_{\mathbf{W}} \left\langle \mathbf{W}_i, \boldsymbol{u}^\top \mathbf{W}^\top \boldsymbol{p}\boldsymbol{v} \right\rangle = \mathbb{E}_{\mathbf{W}} \operatorname{Tr} \left( \nabla_{\mathbf{W}_i} \boldsymbol{u}^\top \mathbf{W}^\top \boldsymbol{p}\boldsymbol{v} \right).$$

Note that only the diagonal entries of the Hessian matrix $\nabla_{\mathbf{W}_i} \boldsymbol{u}^\top \mathbf{W}^\top \boldsymbol{p}\boldsymbol{v}$ matters. For an arbitrary index $s$, we can compute the gradient of the $s$-th entry of $\boldsymbol{u}^\top \mathbf{W}^\top \boldsymbol{p}\boldsymbol{v}$ w.r.t. $\mathbf{W}_{i,s}$ as

$$\frac{d}{dW_{i,s}} \left( \boldsymbol{u}^\top \mathbf{W}^\top \boldsymbol{p}\boldsymbol{v}_s \right) = \boldsymbol{v}_s \frac{d}{dW_{i,s}} \left( \boldsymbol{u}^\top \mathbf{W}^\top \boldsymbol{p} \right) = \boldsymbol{v}_s \sum_{t=1}^{n} \frac{d}{dW_{i,s}} \left( \boldsymbol{p}_t \mathbf{W}_t^\top \boldsymbol{u} \right) = \boldsymbol{v}_s \frac{d}{dW_{i,s}} \boldsymbol{p}_i \mathbf{W}_i^\top \boldsymbol{u} = \boldsymbol{p}_i \boldsymbol{v}_s \boldsymbol{u}_s.$$

Invoking the definitions of $\boldsymbol{p}, \boldsymbol{v}$ and $\boldsymbol{u}$, we have

$$\mathbb{E}_{\mathbf{W},\mathbf{X}} \left\langle \mathbf{W}_i, \boldsymbol{u}^\top \mathbf{W}^\top \boldsymbol{p}\boldsymbol{v} \right\rangle = \mathbb{E}_{\mathbf{X}} \left( \mathbf{X}_{\pi^\natural(i)} - \mathbf{X}_j \right)^\top \mathbf{X}_i \sum_{s=1}^{m} \left[ \mathbf{X}_{\pi^\natural(i)}^\top \left( \mathbf{B}^{\natural\top} \right)_s \left( \mathbf{B}^{\natural\top} \right)_s^\top \mathbf{X}^\top \mathbf{\Pi}^{\natural\top} \mathbf{X} \left( \mathbf{X}_{\pi^\natural(i)} - \mathbf{X}_j \right) \right]$$

$$\overset{①}{=} \mathbb{E} \underbrace{\left[ \left( \mathbf{X}_{\pi^\natural(i)} - \mathbf{X}_j \right)^\top \mathbf{X}_i \mathbf{X}_{\pi^\natural(i)}^\top \mathbf{M}\mathbf{\Sigma}^\top \left( \mathbf{X}_{\pi^\natural(i)} - \mathbf{X}_j \right) \right]}_{\Lambda_1} + \mathbb{E} \underbrace{\left[ \left( \mathbf{X}_{\pi^\natural(i)} - \mathbf{X}_j \right)^\top \mathbf{X}_i \mathbf{X}_{\pi^\natural(i)}^\top \mathbf{M}\mathbf{\Delta}^\top \left( \mathbf{X}_{\pi^\natural(i)} - \mathbf{X}_j \right) \right]}_{\Lambda_2},$$

where in ① we use the relation $\sum_{s=1}^{m} \left( \mathbf{B}^{\natural\top} \right)_s \left( \mathbf{B}^{\natural\top} \right)_s^\top = \mathbf{B}^\natural \mathbf{B}^{\natural\top} = \mathbf{M}$.

For the first term $\Lambda_1$, we obtain

$$\mathbb{E}\Lambda_1 = \underbrace{\mathbb{E} \left( \mathbf{X}_{\pi^\natural(i)}^\top \mathbf{X}_i \mathbf{X}_{\pi^\natural(i)}^\top \mathbf{M}\mathbf{\Sigma}^\top \mathbf{X}_{\pi^\natural(i)} \right)}_{\mathbb{1}_{i=\pi^\natural(i)} \mathbb{E}\|\mathbf{X}_i\|_F^2 \mathbf{X}_i^\top \mathbf{M}\mathbf{\Sigma}^\top \mathbf{X}_i} + \underbrace{\mathbb{E} \left( \mathbf{X}_j^\top \mathbf{X}_i \mathbf{X}_{\pi^\natural(i)}^\top \mathbf{M}\mathbf{\Sigma}^\top \mathbf{X}_j \right)}_{\mathbb{1}_{i=\pi^\natural(i)} \mathbb{E}\mathbf{X}_i^\top \mathbf{M}\mathbf{\Sigma}^\top \mathbf{X}_i}$$

$$- \underbrace{\mathbb{E} \left( \mathbf{X}_{\pi^\natural(i)}^\top \mathbf{X}_i \mathbf{X}_{\pi^\natural(i)}^\top \mathbf{M}\mathbf{\Sigma}^\top \mathbf{X}_j \right)}_{\mathbb{1}_{i=j} \mathbb{1}_{i\neq\pi^\natural(i)} \mathbb{E}\mathbf{X}_{\pi^\natural(i)}^\top \mathbf{M}\mathbf{\Sigma}^\top \mathbf{X}_{\pi^\natural(i)}} - \underbrace{\mathbb{E} \left( \mathbf{X}_j^\top \mathbf{X}_i \mathbf{X}_{\pi^\natural(i)}^\top \mathbf{M}\mathbf{\Sigma}^\top \mathbf{X}_{\pi^\natural(i)} \right)}_{p\mathbb{1}_{i=j} \mathbb{1}_{i\neq\pi^\natural(i)} \mathbb{E}\mathbf{X}_{\pi^\natural(i)}^\top \mathbf{M}\mathbf{\Sigma}^\top \mathbf{X}_{\pi^\natural(i)}}$$

$$= (n - h) \left[ \mathbb{1}_{i=\pi^\natural(i)}(p + 3) - (p + 1)\mathbb{1}_{i=j} \mathbb{1}_{i\neq\pi^\natural(i)} \right] \operatorname{Tr}(\mathbf{M}).$$

Then we consider the second term $\Lambda_2$, which can be decomposed further into four sub-terms reading as

$$\mathbb{E}\Lambda_2 = \underbrace{\mathbb{E} \left( \mathbf{X}_{\pi^\natural(i)}^\top \mathbf{X}_i \mathbf{X}_{\pi^\natural(i)}^\top \mathbf{M}\mathbf{\Delta}^\top \mathbf{X}_{\pi^\natural(i)} \right)}_{\Lambda_{2,1}} + \underbrace{\mathbb{E} \left( \mathbf{X}_j^\top \mathbf{X}_i \mathbf{X}_{\pi^\natural(i)}^\top \mathbf{M}\mathbf{\Delta}^\top \mathbf{X}_j \right)}_{\Lambda_{2,2}}$$

$$- \underbrace{\mathbb{E} \left( \mathbf{X}_{\pi^\natural(i)}^\top \mathbf{X}_i \mathbf{X}_{\pi^\natural(i)}^\top \mathbf{M}\mathbf{\Delta}^\top \mathbf{X}_j \right)}_{\Lambda_{2,3}} - \underbrace{\mathbb{E} \left( \mathbf{X}_j^\top \mathbf{X}_i \mathbf{X}_{\pi^\natural(i)}^\top \mathbf{M}\mathbf{\Delta}^\top \mathbf{X}_{\pi^\natural(i)} \right)}_{\Lambda_{2,4}}.$$

**Case $(s, s)$: $i = \pi^\natural(i)$ and $j = \pi^\natural(j)$.** In this case, we have $\mathbf{\Delta}$ be

$$\mathbf{\Delta}^{(s,s)} = \mathbf{X}_i \mathbf{X}_i^\top + \mathbf{X}_j \mathbf{X}_j^\top.$$

Hence we conclude

$$\mathbb{E}\Lambda_{2,1} = \mathbb{E} \left[ \mathbf{X}_i^\top \mathbf{X}_i \mathbf{X}_i^\top \mathbf{M} \left( \mathbf{X}_i \mathbf{X}_i^\top + \mathbf{X}_j \mathbf{X}_j^\top \right) \mathbf{X}_i \right] = \mathbb{E}\|\mathbf{X}_i\|_F^4 \mathbf{X}_i^\top \mathbf{M}\mathbf{X}_i + \mathbb{E}\|\mathbf{X}_i\|_F^2 \mathbf{X}_i^\top \mathbf{M}\mathbf{X}_i$$

$$= (p + 2)(p + 4) \operatorname{Tr}(\mathbf{M}) + (p + 2) \operatorname{Tr}(\mathbf{M}),$$

$$\mathbb{E}\Lambda_{2,2} = \mathbb{E} \left[ \mathbf{X}_j^\top \mathbf{X}_i \mathbf{X}_i^\top \mathbf{M} \left( \mathbf{X}_i \mathbf{X}_i^\top + \mathbf{X}_j \mathbf{X}_j^\top \right) \mathbf{X}_j \right] = \mathbb{E}\|\mathbf{X}_i\|_F^2 \mathbf{X}_i^\top \mathbf{M}\mathbf{X}_i + \mathbb{E}\|\mathbf{X}_j\|_F^2 \mathbf{X}_j^\top \mathbf{M}\mathbf{X}_j$$

$$= 2\mathbb{E}\|\mathbf{X}_i\|_F^2 \mathbf{X}_i^\top \mathbf{M}\mathbf{X}_i = 2(p + 2) \operatorname{Tr}(\mathbf{M}),$$

$$\mathbb{E}\Lambda_{2,3} = \mathbb{E} \left[ \mathbf{X}_i^\top \mathbf{X}_i \mathbf{X}_i^\top \mathbf{M} \left( \mathbf{X}_i \mathbf{X}_i^\top + \mathbf{X}_j \mathbf{X}_j^\top \right) \mathbf{X}_j \right] = 0,$$

$$\mathbb{E}\Lambda_{2,4} = \mathbb{E} \left[ \mathbf{X}_j^\top \mathbf{X}_i \mathbf{X}_i^\top \mathbf{M} \left( \mathbf{X}_i \mathbf{X}_i^\top + \mathbf{X}_j \mathbf{X}_j^\top \right) \mathbf{X}_i \right] = 0,$$

which suggests $\mathbb{E}\Lambda_2 = (p + 2)(p + 7) \operatorname{Tr}(\mathbf{M})$.

**Case $(s, d)$: $i = \pi^\natural(i)$ and $j \neq \pi^\natural(j)$.** First we write $\mathbf{\Delta}$ as

$$\boldsymbol{\Delta}^{(s,d)\top} = \mathbf{X}_i\mathbf{X}_i^\top + \mathbf{X}_{\pi^\natural(j)}\mathbf{X}_j^\top + \mathbf{X}_j\mathbf{X}_{\pi^{\natural-1}(j)}^\top.$$

Then we conclude

$$\mathbb{E}\Lambda_{2,1} = \underbrace{\mathbb{E}\left(\mathbf{X}_i^\top\mathbf{X}_i\mathbf{X}_i^\top\mathbf{M}\mathbf{X}_i\mathbf{X}_i^\top\mathbf{X}_i\right)}_{\mathbb{E}\|\mathbf{X}_i\|_\mathrm{F}^4\mathbf{X}_i^\top\mathbf{M}\mathbf{X}_i} + \underbrace{\mathbb{E}\left(\mathbf{X}_i^\top\mathbf{X}_i\mathbf{X}_i^\top\mathbf{M}\mathbf{X}_{\pi^\natural(j)}\mathbf{X}_j^\top\mathbf{X}_i\right)}_{0}$$

$$+ \underbrace{\mathbb{E}\left(\mathbf{X}_i^\top\mathbf{X}_i\mathbf{X}_i^\top\mathbf{M}\mathbf{X}_j\mathbf{X}_{\pi^{\natural-1}(j)}^\top\mathbf{X}_i\right)}_{0} = \mathbb{E}\|\mathbf{X}_i\|_\mathrm{F}^4\mathbf{X}_i^\top\mathbf{M}\mathbf{X}_i = (p+2)(p+4)\operatorname{Tr}(\mathbf{M});$$

$$\mathbb{E}\Lambda_{2,2} = \underbrace{\mathbb{E}\left(\mathbf{X}_j^\top\mathbf{X}_i\mathbf{X}_i^\top\mathbf{M}\mathbf{X}_i\mathbf{X}_i^\top\mathbf{X}_j\right)}_{\mathbb{E}\|\mathbf{X}_i\|_\mathrm{F}^2\mathbf{X}_i^\top\mathbf{M}\mathbf{X}_i} + \underbrace{\mathbb{E}\left(\mathbf{X}_j^\top\mathbf{X}_i\mathbf{X}_i^\top\mathbf{M}\mathbf{X}_{\pi^\natural(j)}\mathbf{X}_j^\top\mathbf{X}_j\right)}_{0}$$

$$+ \underbrace{\mathbb{E}\left(\mathbf{X}_j^\top\mathbf{X}_i\mathbf{X}_i^\top\mathbf{M}\mathbf{X}_j\mathbf{X}_{\pi^{\natural-1}(j)}^\top\mathbf{X}_j\right)}_{0} = \mathbb{E}\|\mathbf{X}_i\|_\mathrm{F}^2\mathbf{X}_i^\top\mathbf{M}\mathbf{X}_i = (p+2)\operatorname{Tr}(\mathbf{M}),$$

$$\mathbb{E}\Lambda_{2,3} = \mathbb{E}\left[\mathbf{X}_i^\top\mathbf{X}_i\mathbf{X}_i^\top\mathbf{M}\left(\mathbf{X}_i\mathbf{X}_i^\top + \mathbf{X}_{\pi^\natural(j)}\mathbf{X}_j^\top + \mathbf{X}_j\mathbf{X}_{\pi^{\natural-1}(j)}^\top\right)\mathbf{X}_j\right] = 0,$$

$$\mathbb{E}\Lambda_{2,4} = \mathbb{E}\left[\mathbf{X}_j^\top\mathbf{X}_i\mathbf{X}_i^\top\mathbf{M}\left(\mathbf{X}_i\mathbf{X}_i^\top + \mathbf{X}_{\pi^\natural(j)}\mathbf{X}_j^\top + \mathbf{X}_j\mathbf{X}_{\pi^{\natural-1}(j)}^\top\right)\mathbf{X}_i\right] = 0,$$

which suggests $\mathbb{E}\Lambda_2 = (p+2)(p+5)\operatorname{Tr}(\mathbf{M})$.

**Case $(d,s)$:** $i \neq \pi^\natural(i)$ **and** $j = \pi^\natural(j)$**.** In this case, $\boldsymbol{\Delta}$ reduces to

$$\boldsymbol{\Delta}^{(d,s)\top} = \mathbf{X}_{\pi^\natural(i)}\mathbf{X}_i^\top + \mathbf{X}_{\pi^{\natural 2}(i)}\mathbf{X}_{\pi^\natural(i)}^\top + \mathbf{X}_j\mathbf{X}_j^\top.$$

Then we obtain

$$\mathbb{E}\Lambda_{2,1} = \underbrace{\mathbb{E}\left(\mathbf{X}_{\pi^\natural(i)}^\top\mathbf{X}_i\mathbf{X}_{\pi^\natural(i)}^\top\mathbf{M}\mathbf{X}_{\pi^\natural(i)}\mathbf{X}_i^\top\mathbf{X}_{\pi^\natural(i)}\right)}_{\mathbb{E}\|\mathbf{X}_i\|_\mathrm{F}^2\mathbf{X}_i^\top\mathbf{M}\mathbf{X}_i} + \underbrace{\mathbb{E}\left(\mathbf{X}_{\pi^\natural(i)}^\top\mathbf{X}_i\mathbf{X}_{\pi^\natural(i)}^\top\mathbf{M}\mathbf{X}_{\pi^{\natural 2}(i)}\mathbf{X}_{\pi^\natural(i)}^\top\mathbf{X}_{\pi^\natural(i)}\right)}_{\mathbb{1}_{i=\pi^{\natural 2}(i)}\mathbb{E}\|\mathbf{X}_i\|_\mathrm{F}^2\mathbf{X}_i^\top\mathbf{M}\mathbf{X}_i}$$

$$+ \underbrace{\mathbb{E}\left(\mathbf{X}_{\pi^\natural(i)}^\top\mathbf{X}_i\mathbf{X}_{\pi^\natural(i)}^\top\mathbf{M}\mathbf{X}_j\mathbf{X}_j^\top\mathbf{X}_{\pi^\natural(i)}\right)}_{0} = \left(1 + \mathbb{1}_{i=\pi^{\natural 2}(i)}\right)(p+2)\operatorname{Tr}(\mathbf{M}),$$

$$\mathbb{E}\Lambda_{2,2} = \underbrace{\mathbb{E}\left(\mathbf{X}_j^\top\mathbf{X}_i\mathbf{X}_{\pi^\natural(i)}^\top\mathbf{M}\mathbf{X}_{\pi^\natural(i)}\mathbf{X}_i^\top\mathbf{X}_j\right)}_{\mathbb{1}_{i=j}\mathbb{E}\|\mathbf{X}_i\|_\mathrm{F}^4\operatorname{Tr}(\mathbf{M})+\mathbb{1}_{i\neq j}p\operatorname{Tr}(\mathbf{M})} + \underbrace{\mathbb{E}\left(\mathbf{X}_j^\top\mathbf{X}_i\mathbf{X}_{\pi^\natural(i)}^\top\mathbf{M}\mathbf{X}_{\pi^{\natural 2}(i)}\mathbf{X}_{\pi^\natural(i)}^\top\mathbf{X}_j\right)}_{\mathbb{1}_{i=\pi^{\natural 2}(i)}\left[\mathbb{1}_{i=j}\mathbb{E}\|\mathbf{X}_i\|_\mathrm{F}^2\mathbf{X}_i^\top\mathbf{M}\mathbf{X}_i+\mathbb{1}_{i\neq j}\operatorname{Tr}(\mathbf{M})\right]}$$

$$+ \underbrace{\mathbb{E}\left(\mathbf{X}_j^\top\mathbf{X}_i\mathbf{X}_{\pi^\natural(i)}^\top\mathbf{M}\mathbf{X}_j\mathbf{X}_j^\top\mathbf{X}_j\right)}_{0} = \left(p + \mathbb{1}_{i=\pi^{\natural 2}(i)}\right)\operatorname{Tr}(\mathbf{M}),$$

$$\mathbb{E}\Lambda_{2,3} = \mathbb{E}\left[\mathbf{X}_{\pi^\natural(i)}^\top\mathbf{X}_i\mathbf{X}_{\pi^\natural(i)}^\top\mathbf{M}\left(\mathbf{X}_{\pi^\natural(i)}\mathbf{X}_i^\top + \mathbf{X}_{\pi^{\natural 2}(i)}\mathbf{X}_{\pi^\natural(i)}^\top + \mathbf{X}_j\mathbf{X}_j^\top\right)\mathbf{X}_j\right] = 0,$$

$$\mathbb{E}\Lambda_{2,4} = \mathbb{E}\left[\mathbf{X}_j^\top\mathbf{X}_i\mathbf{X}_{\pi^\natural(i)}^\top\mathbf{M}\left(\mathbf{X}_{\pi^\natural(i)}\mathbf{X}_i^\top + \mathbf{X}_{\pi^{\natural 2}(i)}\mathbf{X}_{\pi^\natural(i)}^\top + \mathbf{X}_j\mathbf{X}_j^\top\right)\mathbf{X}_{\pi^\natural(i)}\right] = 0,$$

which suggests

$$\mathbb{E}\Lambda_2 = 2(p+1)\operatorname{Tr}(\mathbf{M}) + (p+3)\mathbb{1}_{i=\pi^{\natural 2}(i)}\operatorname{Tr}(\mathbf{M}).$$

**Case $(d,d)$:** $i \neq \pi^\natural(i)$ **and** $j \neq \pi^\natural(j)$**.** In this case, $\boldsymbol{\Delta}$ is written as

$$\boldsymbol{\Delta}^{(d,d)\top} = \mathbf{X}_{\pi^\natural(i)}\mathbf{X}_i^\top + \mathbf{X}_{\pi^{\natural 2}(i)}\mathbf{X}_{\pi^\natural(i)}^\top + \mathbf{X}_{\pi^\natural(j)}\mathbf{X}_j^\top + \mathbf{X}_j\mathbf{X}_{\pi^{\natural-1}(j)}^\top.$$

Then we have

$$\mathbb{E}\Lambda_{2,1} = \underbrace{\mathbb{E}\left(\mathbf{X}_{\pi^\natural(i)}^\top\mathbf{X}_i\mathbf{X}_{\pi^\natural(i)}^\top\mathbf{M}\mathbf{X}_{\pi^\natural(i)}\mathbf{X}_i^\top\mathbf{X}_{\pi^\natural(i)}\right)}_{\mathbb{E}\|\mathbf{X}_i\|_\mathrm{F}^2\mathbf{X}_i^\top\mathbf{M}\mathbf{X}_i} + \underbrace{\mathbb{E}\left(\mathbf{X}_{\pi^\natural(i)}^\top\mathbf{X}_i\mathbf{X}_{\pi^\natural(i)}^\top\mathbf{M}\mathbf{X}_{\pi^{\natural 2}(i)}\mathbf{X}_{\pi^\natural(i)}^\top\mathbf{X}_{\pi^\natural(i)}\right)}_{\mathbb{1}_{i=\pi^{\natural 2}(i)}\mathbb{E}\|\mathbf{X}_i\|_\mathrm{F}^2\mathbf{X}_i^\top\mathbf{M}\mathbf{X}_i}$$

$$+ \underbrace{\mathbb{E}\left(\mathbf{X}_{\pi^{\natural}(i)}^{\top}\mathbf{X}_i\mathbf{X}_{\pi^{\natural}(i)}^{\top}\mathbf{M}\mathbf{X}_{\pi^{\natural}(j)}\mathbf{X}_j^{\top}\mathbf{X}_{\pi^{\natural}(i)}\right)}_{\mathbb{1}_{i=j}\mathbb{E}\|\mathbf{X}_i\|_{\mathrm{F}}^2\mathbf{X}_i^{\top}\mathbf{M}\mathbf{X}_i} + \underbrace{\mathbb{E}\left(\mathbf{X}_{\pi^{\natural}(i)}^{\top}\mathbf{X}_i\mathbf{X}_{\pi^{\natural}(i)}^{\top}\mathbf{M}\mathbf{X}_j\mathbf{X}_{\pi^{\natural-1}(j)}^{\top}\mathbf{X}_{\pi^{\natural}(i)}\right)}_{\mathbb{1}_{i=j}\mathbb{1}_{i=\pi^{\natural 2}(i)}\mathbb{E}\|\mathbf{X}_i\|_{\mathrm{F}}^2\mathbf{X}_i^{\top}\mathbf{M}\mathbf{X}_i}$$

$$= (p+2)\left[1 + \mathbb{1}_{i=\pi^{\natural 2}(i)} + \mathbb{1}_{i=j} + \mathbb{1}_{i=\pi^{\natural 2}(i)}\mathbb{1}_{i=j}\right]\mathrm{Tr}(\mathbf{M}),$$

$$\mathbb{E}\Lambda_{2,2} = \underbrace{\mathbb{E}\left(\mathbf{X}_j^{\top}\mathbf{X}_i\mathbf{X}_{\pi^{\natural}(i)}^{\top}\mathbf{M}\mathbf{X}_{\pi^{\natural}(i)}\mathbf{X}_i^{\top}\mathbf{X}_j\right)}_{\mathbb{1}_{i=j}\mathbb{E}\|\mathbf{X}_i\|_{\mathrm{F}}^4\mathrm{Tr}(\mathbf{M})+p\mathbb{1}_{i\neq j}\mathrm{Tr}(\mathbf{M})} + \underbrace{\mathbb{E}\left(\mathbf{X}_j^{\top}\mathbf{X}_i\mathbf{X}_{\pi^{\natural}(i)}^{\top}\mathbf{M}\mathbf{X}_{\pi^{\natural 2}(i)}\mathbf{X}_{\pi^{\natural}(i)}^{\top}\mathbf{X}_j\right)}_{\mathbb{1}_{i=\pi^{\natural 2}(i)}[\mathbb{1}_{i=j}(p+2)\mathrm{Tr}(\mathbf{M})+\mathbb{1}_{i\neq j}\mathrm{Tr}(\mathbf{M})]}$$

$$+ \underbrace{\mathbb{E}\left(\mathbf{X}_j^{\top}\mathbf{X}_i\mathbf{X}_{\pi^{\natural}(i)}^{\top}\mathbf{M}\mathbf{X}_{\pi^{\natural}(j)}\mathbf{X}_j^{\top}\mathbf{X}_j\right)}_{\mathbb{1}_{i=j}\mathbb{E}\|\mathbf{X}_i\|_{\mathrm{F}}^4\mathrm{Tr}(\mathbf{M})} + \underbrace{\mathbb{E}\left(\mathbf{X}_j^{\top}\mathbf{X}_i\mathbf{X}_{\pi^{\natural}(i)}^{\top}\mathbf{M}\mathbf{X}_j\mathbf{X}_{\pi^{\natural-1}(j)}^{\top}\mathbf{X}_j\right)}_{\mathbb{1}_{i=j}\mathbb{1}_{i=\pi^{\natural 2}(i)}\mathbb{E}\|\mathbf{X}_i\|_{\mathrm{F}}^2\mathbf{X}_i^{\top}\mathbf{M}\mathbf{X}_i}$$

$$= 2\mathbb{1}_{i=j}p(p+2)\mathrm{Tr}(\mathbf{M}) + p\mathbb{1}_{i\neq j}\mathrm{Tr}(\mathbf{M}) + \mathbb{1}_{i=\pi^{\natural 2}(i)}\left[\mathbb{1}_{i=j}2(p+2)\mathrm{Tr}(\mathbf{M}) + \mathbb{1}_{i\neq j}\mathrm{Tr}(\mathbf{M})\right],$$

$$\mathbb{E}\Lambda_{2,3} = \underbrace{\mathbb{E}\left(\mathbf{X}_{\pi^{\natural}(i)}^{\top}\mathbf{X}_i\mathbf{X}_{\pi^{\natural}(i)}^{\top}\mathbf{M}\mathbf{X}_{\pi^{\natural}(i)}\mathbf{X}_i^{\top}\mathbf{X}_j\right)}_{0} + \underbrace{\mathbb{E}\left(\mathbf{X}_{\pi^{\natural}(i)}^{\top}\mathbf{X}_i\mathbf{X}_{\pi^{\natural}(i)}^{\top}\mathbf{M}\mathbf{X}_{\pi^{\natural 2}(i)}\mathbf{X}_{\pi^{\natural}(i)}^{\top}\mathbf{X}_j\right)}_{0}$$

$$+ \underbrace{\mathbb{E}\left(\mathbf{X}_{\pi^{\natural}(i)}^{\top}\mathbf{X}_i\mathbf{X}_{\pi^{\natural}(i)}^{\top}\mathbf{M}\mathbf{X}_{\pi^{\natural}(j)}\mathbf{X}_j^{\top}\mathbf{X}_j\right)}_{p\mathbb{1}_{i=\pi^{\natural}(j)}\mathrm{Tr}(\mathbf{M})} + \underbrace{\mathbb{E}\left(\mathbf{X}_{\pi^{\natural}(i)}^{\top}\mathbf{X}_i\mathbf{X}_{\pi^{\natural}(i)}^{\top}\mathbf{M}\mathbf{X}_j\mathbf{X}_{\pi^{\natural-1}(j)}^{\top}\mathbf{X}_j\right)}_{0},$$

$$\mathbb{E}\Lambda_{2,4} = \underbrace{\mathbb{E}\left(\mathbf{X}_j^{\top}\mathbf{X}_i\mathbf{X}_{\pi^{\natural}(i)}^{\top}\mathbf{M}\mathbf{X}_{\pi^{\natural}(i)}\mathbf{X}_i^{\top}\mathbf{X}_{\pi^{\natural}(i)}\right)}_{0} + \underbrace{\mathbb{E}\left(\mathbf{X}_j^{\top}\mathbf{X}_i\mathbf{X}_{\pi^{\natural}(i)}^{\top}\mathbf{M}\mathbf{X}_{\pi^{\natural 2}(i)}\mathbf{X}_{\pi^{\natural}(i)}^{\top}\mathbf{X}_{\pi^{\natural}(i)}\right)}_{0}$$

$$+ \underbrace{\mathbb{E}\left(\mathbf{X}_j^{\top}\mathbf{X}_i\mathbf{X}_{\pi^{\natural}(i)}^{\top}\mathbf{M}\mathbf{X}_{\pi^{\natural}(j)}\mathbf{X}_j^{\top}\mathbf{X}_{\pi^{\natural}(i)}\right)}_{\mathbb{1}_{i=\pi^{\natural}(j)}\mathrm{Tr}(\mathbf{M})} + \underbrace{\mathbb{E}\left(\mathbf{X}_j^{\top}\mathbf{X}_i\mathbf{X}_{\pi^{\natural}(i)}^{\top}\mathbf{M}\mathbf{X}_j\mathbf{X}_{\pi^{\natural-1}(j)}^{\top}\mathbf{X}_{\pi^{\natural}(i)}\right)}_{0},$$

which gives

$$\mathbb{E}\Lambda_2 = (p+1)\left[2 + 2(p+1)\mathbb{1}_{i=j} + \mathbb{1}_{i=\pi^{\natural}(j)}\right]\mathrm{Tr}(\mathbf{M}) + \mathbb{1}_{i=\pi^{\natural 2}(i)}\left[p+3 + (3p+5)\mathbb{1}_{i=j}\right]\mathrm{Tr}(\mathbf{M}).$$

$\square$

### D.2.3 SUPPORTING LEMMAS

First, we study the higher order expectations of Gaussian random vectors' inner product, which hopefully will serve independent interests.

**Lemma 11.** *Assume $\boldsymbol{x} \in \mathbb{R}^p$ and $\boldsymbol{y} \in \mathbb{R}^p$ are Gaussian distributed random vectors whose entries follow the i.i.d. standard normal distribution, then we have*

$$\mathbb{E}\,\mathrm{Tr}\left(\boldsymbol{y}\boldsymbol{y}^{\top}\boldsymbol{x}\boldsymbol{x}^{\top}\mathbf{M}\right) = \mathrm{Tr}(\mathbf{M}), \tag{63}$$

$$\mathbb{E}\,\mathrm{Tr}\left(\boldsymbol{y}\boldsymbol{y}^{\top}\boldsymbol{x}^{\top}\mathbf{M}\boldsymbol{x}\right) = \mathbb{E}\|\boldsymbol{y}\|_2^2\,\mathrm{Tr}\left(\boldsymbol{x}^{\top}\mathbf{M}\boldsymbol{x}\right) = p\,\mathrm{Tr}(\mathbf{M}), \tag{64}$$

$$\mathbb{E}\left(\boldsymbol{x}^{\top}\mathbf{M}\boldsymbol{x}\right)^2 = [\mathrm{Tr}(\mathbf{M})]^2 + \mathrm{Tr}(\mathbf{M}\mathbf{M}) + \mathrm{Tr}\left(\mathbf{M}^{\top}\mathbf{M}\right), \tag{65}$$

$$\mathbb{E}\|\boldsymbol{x}\|_2^2(\boldsymbol{x}^{\top}\mathbf{M}\boldsymbol{x}) = (p+2)\,\mathrm{Tr}(\mathbf{M}), \tag{66}$$

$$\mathbb{E}\|\boldsymbol{x}\|_2^4(\boldsymbol{x}^{\top}\mathbf{M}\boldsymbol{x}) = (p+2)(p+4)\,\mathrm{Tr}(\mathbf{M}), \tag{67}$$

$$\mathbb{E}\|\boldsymbol{x}\|_2^2\left(\boldsymbol{x}^{\top}\mathbf{M}\boldsymbol{x}\right)^2 = (p+4)\left[(\mathrm{Tr}(\mathbf{M}))^2 + \mathrm{Tr}(\mathbf{M}\mathbf{M}) + \mathrm{Tr}\left(\mathbf{M}^{\top}\mathbf{M}\right)\right], \tag{68}$$

$$\mathbb{E}\|\boldsymbol{x}\|_2^4\left(\boldsymbol{x}^{\top}\mathbf{M}\boldsymbol{x}\right)^2 = (p+4)(p+6)\left[(\mathrm{Tr}(\mathbf{M}))^2 + \mathrm{Tr}(\mathbf{M}\mathbf{M}) + \mathrm{Tr}\left(\mathbf{M}^{\top}\mathbf{M}\right)\right], \tag{69}$$

$$\mathbb{E}(\boldsymbol{x}^{\top}\boldsymbol{y})^2\boldsymbol{y}^{\top}\mathbf{M}_1\boldsymbol{x}\boldsymbol{x}^{\top}\mathbf{M}_2\boldsymbol{y} = 2\,\mathrm{Tr}(\mathbf{M}_1)\,\mathrm{Tr}(\mathbf{M}_2) + (p+4)\,\mathrm{Tr}(\mathbf{M}_1\mathbf{M}_2) + 2\,\mathrm{Tr}(\mathbf{M}_1\mathbf{M}_2^{\top}), \tag{70}$$

*where $\mathbf{M} \in \mathbb{R}^{p\times p}$ is a fixed matrix.*

**Remark 2.** *If we assume $\mathbf{M} = \mathbf{I}_{p\times p}$, we can get $\mathbb{E}\|\boldsymbol{x}\|_2^4 = p(p+2)$, $\mathbb{E}\|\boldsymbol{x}\|_2^6 = p(p+2)(p+4)$, and $\mathbb{E}\|\boldsymbol{x}\|_2^8 = p(p+2)(p+4)(p+6)$.*

*Proof.* This lemma is proved by iteratively applying the Wick's theorem in Theorem 4, Stein's lemma in Lemma 20, and Lemma 19.

- **Proof of** (63) **and** (64). The proof can be conducted easily with the property such that $\text{Tr}(\boldsymbol{u}\boldsymbol{v}^\top) = \boldsymbol{u}^\top\boldsymbol{v} = \text{Tr}(\boldsymbol{v}\boldsymbol{u}^\top)$ holds for arbitrary vectors $\boldsymbol{u}$ and $\boldsymbol{v}$.

- **Proof of** (65). This property is a direct consequence of Neudecker & Wansbeek (1987) (Equation (3.2)), which is attached in Lemma 19 for the sake of self-containing.

- **Proof of** (66). Invoking the Stein's lemma, we have

$$\mathbb{E}\|\boldsymbol{x}\|_2^2(\boldsymbol{x}^\top\mathbf{M}\boldsymbol{x}) = \mathbb{E}\left[\nabla_{\boldsymbol{x}}(\boldsymbol{x}^\top\mathbf{M}\boldsymbol{x})\boldsymbol{x}\right].$$

Then our goal transforms to computing the trace of the Hessian matrix $\text{Tr}\left[\nabla_{\boldsymbol{x}}\text{Tr}(\boldsymbol{x}^\top\mathbf{M}\boldsymbol{x})\boldsymbol{x}\right]$. For the $i$-th entry of the gradient, we have

$$\frac{d}{dx_i}\boldsymbol{x}^\top\mathbf{M}\boldsymbol{x} = \langle\mathbf{M}_i, \boldsymbol{x}\rangle + \langle(\mathbf{M}^\top)_i, \boldsymbol{x}\rangle,$$

where $\mathbf{M}_i$ is the $i$-th row (or column) of $\mathbf{M}$. Then we obtain

$$\frac{d}{dx_i}\left[x_i\,\text{Tr}(\boldsymbol{x}^\top\mathbf{M}\boldsymbol{x})\right] = \boldsymbol{x}^\top\mathbf{M}\boldsymbol{x} + x_i\left[\langle\mathbf{M}_i, \boldsymbol{x}\rangle + \langle(\mathbf{M}^\top)_i, \boldsymbol{x}\rangle\right],$$

and hence

$$\mathbb{E}\|\boldsymbol{x}\|_2^2(\boldsymbol{x}^\top\mathbf{M}\boldsymbol{x}) = \sum_{i=1}^p \mathbb{E}(\boldsymbol{x}^\top\mathbf{M}\boldsymbol{x}) + \sum_{i=1}^p\mathbb{E}\left[x_i\left(\langle\mathbf{M}_i, \boldsymbol{x}\rangle + \langle(\mathbf{M}^\top)_i, \boldsymbol{x}\rangle\right)\right]$$

$$= p\,\text{Tr}(\mathbf{M}) + 2\sum_i\mathbf{M}_{ii} = (p+2)\,\text{Tr}(\mathbf{M}).$$

- **Proof of** (67). Following the same strategy as in proving (66), we have

$$\mathbb{E}\|\boldsymbol{x}\|_{\text{F}}^4(\boldsymbol{x}^\top\mathbf{M}\boldsymbol{x}) = \mathbb{E}\left[\nabla_{\boldsymbol{x}}\|\boldsymbol{x}\|_2^2(\boldsymbol{x}^\top\mathbf{M}\boldsymbol{x})\boldsymbol{x}\right].$$

Then our goal transforms to computing the trace of the Hessian matrix $\text{Tr}\left[\nabla_{\boldsymbol{x}}\|\boldsymbol{x}\|_2^2(\boldsymbol{x}^\top\mathbf{M}\boldsymbol{x})\boldsymbol{x}\right]$. For the $i$-th entry of the gradient, we obtain

$$\frac{d}{dx_i}\left[x_i\|\boldsymbol{x}\|_2^2(\boldsymbol{x}^\top\mathbf{M}\boldsymbol{x})\right] = \|\boldsymbol{x}\|_2^2\cdot(\boldsymbol{x}^\top\mathbf{M}\boldsymbol{x}) + x_i\frac{d}{dx_i}\left[\|\boldsymbol{x}\|_2^2(\boldsymbol{x}^\top\mathbf{M}\boldsymbol{x})\right]$$

$$= \|\boldsymbol{x}\|_2^2\cdot(\boldsymbol{x}^\top\mathbf{M}\boldsymbol{x}) + 2x_i^2(\boldsymbol{x}^\top\mathbf{M}\boldsymbol{x}) + x_i\|\boldsymbol{x}\|_2^2\left[\langle\mathbf{M}_i, \boldsymbol{x}\rangle + \langle(\mathbf{M}^\top)_i, \boldsymbol{x}\rangle\right],$$

whose expectation reads as

$$\mathbb{E}\|\boldsymbol{x}\|_2^2\cdot(\boldsymbol{x}^\top\mathbf{M}\boldsymbol{x}) + 2\mathbb{E}\left[x_i^2(\boldsymbol{x}^\top\mathbf{M}\boldsymbol{x})\right] + \mathbb{E}x_i\|\boldsymbol{x}\|_2^2\left[\langle\mathbf{M}_i, \boldsymbol{x}\rangle + \langle(\mathbf{M}^\top)_i, \boldsymbol{x}\rangle\right]$$

$$= (p+2)\,\text{Tr}(\mathbf{M}) + 2\mathbb{E}\left[x_i^2 M_{ii}x_i^2 + x_i^2\left(\sum_{j\neq i}M_{jj}x_j^2\right)\right] + 2M_{ii}\mathbb{E}\left[x_i^2\|\boldsymbol{x}\|_2^2\right]$$

$$= (p+2)\,\text{Tr}(\mathbf{M}) + 2M_{ii}\left(\mathbb{E}x_i^4\right) + 2\sum_{j\neq i}M_{jj}(\mathbb{E}x_i^2)(\mathbb{E}x_j^2) + 2M_{ii}\left[\mathbb{E}(x_i^4) + \sum_{j\neq i}(\mathbb{E}x_i^2)(\mathbb{E}x_j^2)\right]$$

$$= (p+2)\,\text{Tr}(\mathbf{M}) + 6M_{ii} + 2\sum_{j\neq i}M_{jj} + 2M_{ii}(3+p-1)$$

$$= (p+4)\,\text{Tr}(\mathbf{M}) + 2(p+4)M_{ii}.$$

Then we conclude

$$\mathbb{E}\,\text{Tr}\left[\nabla_{\boldsymbol{x}}\|\boldsymbol{x}\|_2^2(\boldsymbol{x}^\top\mathbf{M}\boldsymbol{x})\boldsymbol{x}\right] = p(p+2)\,\text{Tr}(\mathbf{M}) + 2(p+4)\sum_i M_{ii} + 2p\,\text{Tr}(\mathbf{M})$$

$$= (p+2)(p+4)\,\text{Tr}(\mathbf{M}).$$

- **Proof of** (68). Invoking the Stein's lemma, we have

$$\mathbb{E}\|\boldsymbol{x}\|_2^2\left(\boldsymbol{x}^\top\mathbf{M}\boldsymbol{x}\right)^2 = \sum_i \frac{d}{dx_i}\left[x_i\left(\boldsymbol{x}^\top\mathbf{M}\boldsymbol{x}\right)^2\right] = p\left(\boldsymbol{x}^\top\mathbf{M}\boldsymbol{x}\right)^2 + 4\sum_i x_i\left(\boldsymbol{x}^\top\mathbf{M}\boldsymbol{x}\right)\left\langle\mathbf{M}_i^{(\mathsf{sym})},\boldsymbol{x}\right\rangle.$$

The proof is then completed by invoking Lemma 12.

- **Proof of** (69). Following the same strategy as in proving (68), we consider the $i$-th gradient w.r.t $x_i$, which can be written as

$$\mathbb{E}\|\boldsymbol{x}\|_2^4\left(\boldsymbol{x}^\top\mathbf{M}\boldsymbol{x}\right)^2 = \sum_i \frac{d}{dx_i}\left[x_i\|\boldsymbol{x}\|_2^2\left(\boldsymbol{x}^\top\mathbf{M}\boldsymbol{x}\right)^2\right]$$

$$= \sum_i\|\boldsymbol{x}\|_2^2\left(\boldsymbol{x}^\top\mathbf{M}\boldsymbol{x}\right)^2 + 2\sum_i x_i^2\left(\boldsymbol{x}^\top\mathbf{M}\boldsymbol{x}\right)^2 + 4\sum_i x_i\|\boldsymbol{x}\|_2^2(\boldsymbol{x}^\top\mathbf{M}\boldsymbol{x})\left\langle\mathbf{M}^{(\mathsf{sym})},\boldsymbol{x}\right\rangle$$

$$= \sum_i\|\boldsymbol{x}\|_2^2\left(\boldsymbol{x}^\top\mathbf{M}\boldsymbol{x}\right)^2 + 2\sum_i\mathbb{E}\frac{d}{dx_i}\left[x_i\left(\boldsymbol{x}^\top\mathbf{M}\boldsymbol{x}\right)^2\right] + 4\sum_i\mathbb{E}\frac{d}{dx_i}\left[\|\boldsymbol{x}\|_2^2(\boldsymbol{x}^\top\mathbf{M}\boldsymbol{x})\left\langle\mathbf{M}_i^{(\mathsf{sym})},\boldsymbol{x}\right\rangle\right]. \tag{71}$$

Noticing the following relations

$$\frac{d}{dx_i}\left[x_i\left(\boldsymbol{x}^\top\mathbf{M}\boldsymbol{x}\right)^2\right] = \left(\boldsymbol{x}^\top\mathbf{M}\boldsymbol{x}\right)^2 + 4x_i\left(\boldsymbol{x}^\top\mathbf{M}\boldsymbol{x}\right)\left\langle\mathbf{M}_i^{(\mathsf{sym})},\boldsymbol{x}\right\rangle, \tag{72}$$

$$\frac{d}{dx_i}\left[\|\boldsymbol{x}\|_2^2\mathrm{Tr}(\boldsymbol{x}^\top\mathbf{M}\boldsymbol{x})\left\langle\mathbf{M}_i^{(\mathsf{sym})},\boldsymbol{x}\right\rangle\right] = 2x_i\left(\boldsymbol{x}^\top\mathbf{M}\boldsymbol{x}\right)\left\langle\mathbf{M}_i^{(\mathsf{sym})},\boldsymbol{x}\right\rangle$$

$$+ 2\|\boldsymbol{x}\|_2^2\left\langle\mathbf{M}_i^{(\mathsf{sym})},\boldsymbol{x}\right\rangle^2 + M_{ii}\|\boldsymbol{x}\|_2^2\left(\boldsymbol{x}^\top\mathbf{M}\boldsymbol{x}\right), \tag{73}$$

we can conclude the proof by combining (68), (71), (72), (73), and Lemma 12.

- **Proof of** (70). Due to the independence between $\boldsymbol{x}$ and $\boldsymbol{y}$, we first condition on $\boldsymbol{x}$ and have

$$\mathbb{E}(\boldsymbol{x}^\top\boldsymbol{y})^2\boldsymbol{y}^\top\mathbf{M}_1\boldsymbol{x}\boldsymbol{x}^\top\mathbf{M}_2\boldsymbol{y} = \mathbb{E}_{\boldsymbol{x}}\mathbb{E}_{\boldsymbol{y}}\boldsymbol{y}^\top\boldsymbol{x}\boldsymbol{x}^\top\boldsymbol{y}\boldsymbol{y}^\top\mathbf{M}_1\boldsymbol{x}\boldsymbol{x}^\top\mathbf{M}_2\boldsymbol{y}$$

$$\overset{\text{①}}{=} \mathbb{E}_{\boldsymbol{x}}\mathrm{Tr}\left(\boldsymbol{x}\boldsymbol{x}^\top\right)\mathrm{Tr}\left(\mathbf{M}_1\boldsymbol{x}\boldsymbol{x}^\top\mathbf{M}_2\right) + \mathbb{E}_{\boldsymbol{x}}\mathrm{Tr}\left(\boldsymbol{x}\boldsymbol{x}^\top\mathbf{M}_1\boldsymbol{x}\boldsymbol{x}^\top\mathbf{M}_2\right) + \mathbb{E}_{\boldsymbol{x}}\mathrm{Tr}\left(\boldsymbol{x}\boldsymbol{x}^\top\mathbf{M}_2^\top\boldsymbol{x}\boldsymbol{x}\mathbf{M}_1^\top\right)$$

$$= \mathbb{E}_{\boldsymbol{x}}\|\boldsymbol{x}\|_2^2\boldsymbol{x}^\top\mathbf{M}_2\mathbf{M}_1\boldsymbol{x} + \mathbb{E}_{\boldsymbol{x}}\boldsymbol{x}^\top\mathbf{M}_1\boldsymbol{x}\boldsymbol{x}^\top\mathbf{M}_2\boldsymbol{x} + \mathbb{E}_{\boldsymbol{x}}\boldsymbol{x}^\top\mathbf{M}_1^\top\boldsymbol{x}\boldsymbol{x}^\top\mathbf{M}_2^\top\boldsymbol{x}$$

$$\overset{\text{②}}{=} 2\mathrm{Tr}(\mathbf{M}_1)\mathrm{Tr}(\mathbf{M}_2) + (p+4)\mathrm{Tr}(\mathbf{M}_1\mathbf{M}_2) + 2\mathrm{Tr}(\mathbf{M}_1\mathbf{M}_2^\top),$$

where in ① and ② we both use Lemma 19.

$\square$

**Lemma 12.** *For a fixed matrix* $\mathbf{M}\in\mathbb{R}^{p\times p}$, *we associate it with a symmetric matrix* $\mathbf{M}^{(\mathsf{sym})}$ *defined as* $(\mathbf{M}+\mathbf{M}^\top)/2$. *Consider the Gaussian distributed random vector* $\boldsymbol{x}\sim\mathsf{N}(\mathbf{0},\mathbf{I})$, *we have*

$$\mathbb{E}\sum_i x_i(\boldsymbol{x}^\top\mathbf{M}\boldsymbol{x})\left\langle\mathbf{M}_i^{(\mathsf{sym})},\boldsymbol{x}\right\rangle = (\mathrm{Tr}(\mathbf{M}))^2 + \|\mathbf{M}\|_\mathrm{F}^2 + \mathrm{Tr}(\mathbf{M}\mathbf{M}).$$

*Proof.* This lemma is a direct application of Wick's theorem, which is completed by showing

$$\mathbb{E}x_i\left(\boldsymbol{x}^\top\mathbf{M}\boldsymbol{x}\right)\left\langle\mathbf{M}_i^{(\mathsf{sym})},\boldsymbol{x}\right\rangle = \mathbb{E}\sum_j\sum_{\ell_1,\ell_2} M_{i,j}^{(\mathsf{sym})}M_{\ell_1,\ell_2}x_ix_jx_{\ell_1}x_{\ell_2}$$

$$= \underbrace{\mathbb{E}\sum_j\sum_{\ell_1,\ell_2}\mathbb{1}_{\ell_1=i}\mathbb{1}_{\ell_2=j}M_{i,j}^{(\mathsf{sym})}M_{\ell_1,\ell_2}x_ix_jx_{\ell_1}x_{\ell_2}}_{\sum_j M_{i,j}^{(\mathsf{sym})}M_{i,j}} + \underbrace{\mathbb{E}\sum_j\sum_{\ell_1,\ell_2}\mathbb{1}_{\ell_2=i}\mathbb{1}_{\ell_1=j}M_{i,j}^{(\mathsf{sym})}M_{\ell_1,\ell_2}x_ix_jx_{\ell_1}x_{\ell_2}}_{\sum_j M_{i,j}^{(\mathsf{sym})}M_{j,i}}$$

$$+ \underbrace{\mathbb{E}\sum_j\sum_{\ell_1,\ell_2}\mathbb{1}_{i=j}\mathbb{1}_{\ell_1=\ell_2}M_{i,j}^{(\mathsf{sym})}M_{\ell_1,\ell_2}x_ix_jx_{\ell_1}x_{\ell_2}}_{\sum_\ell M_{i,i}^{(\mathsf{sym})}M_{\ell,\ell}} = \sum_j 2\left[M_{i,j}^{(\mathsf{sym})}\right]^2 + M_{i,i}\mathrm{Tr}(\mathbf{M}),$$

where $\mathbf{M}^{(\mathsf{sym})}$ is defined as $\left(\mathbf{M}+\mathbf{M}^\top\right)/2$. $\square$

Then we study the properties of $\mathbf{\Sigma}$, which is defined as $\mathbf{X}^\top \mathbf{\Pi}^\natural \mathbf{X} - \mathbf{\Delta}$.

**Lemma 13.** *For a fixed matrix* $\mathbf{M}$, *we have*

$$\mathbb{E}\operatorname{Tr}\left(\mathbf{\Sigma}\mathbf{M}\mathbf{\Sigma}^\top\right) = n^2\left[\frac{p}{n} + \left(1 - \frac{h}{n}\right)^2 + o(1)\right]\operatorname{Tr}(\mathbf{M}),$$

*where matrix* $\mathbf{\Sigma}$ *is defined in* (22).

*Proof.* We conclude the proof by showing

$$
\begin{aligned}
\mathbb{E}\operatorname{Tr}\left(\mathbf{\Sigma}\mathbf{M}\mathbf{\Sigma}^\top\right) &\overset{\textcircled{1}}{=} \sum_{\ell_1,\ell_2\in\mathcal{S}} \mathbb{E}\operatorname{Tr}\left[\mathbf{X}_{\ell_1}\mathbf{X}_{\ell_1}^\top \mathbf{M}\mathbf{X}_{\ell_2}\mathbf{X}_{\ell_2}^\top\right] + \sum_{\ell_1,\ell_2\in\mathcal{D}} \mathbb{E}\operatorname{Tr}\left[\mathbf{X}_{\ell_1}\mathbf{X}_{\pi^\natural(\ell_1)}^\top \mathbf{M}\mathbf{X}_{\pi^\natural(\ell_2)}\mathbf{X}_{\ell_2}^\top\right] \\
&= \sum_{\ell\in\mathcal{S}} \mathbb{E}\operatorname{Tr}\left(\mathbf{X}_\ell\mathbf{X}_\ell^\top \mathbf{M}\mathbf{X}_\ell\mathbf{X}_\ell^\top\right) + \sum_{\ell_1,\ell_2\in\mathcal{S},\ell_1\neq\ell_2} \mathbb{E}\operatorname{Tr}\left[\mathbf{X}_{\ell_1}\mathbf{X}_{\ell_1}^\top \mathbf{M}\mathbf{X}_{\ell_2}\mathbf{X}_{\ell_2}^\top\right] \\
&\quad + \sum_{\ell\in\mathcal{D}} \underbrace{\mathbb{E}\operatorname{Tr}\left[\mathbf{X}_\ell\mathbf{X}_{\pi^\natural(\ell)}^\top \mathbf{M}\mathbf{X}_{\pi^\natural(\ell)}\mathbf{X}_\ell^\top\right]}_{p\operatorname{Tr}(\mathbf{M})} + \sum_{(\ell_1,\ell_2)\in\mathcal{D}_{\text{pair}}} \underbrace{\mathbb{E}\operatorname{Tr}\left[\mathbf{X}_{\ell_1}\mathbf{X}_{\ell_2}^\top \mathbf{M}\mathbf{X}_{\ell_1}\mathbf{X}_{\ell_2}^\top\right]}_{\operatorname{Tr}(\mathbf{M})} \\
&= (n-h)(p+2)\operatorname{Tr}(\mathbf{M}) + (n-h)(n-h-1)\operatorname{Tr}(\mathbf{M}) + hp\operatorname{Tr}(\mathbf{M}) + |\mathcal{D}_{\text{pair}}|\operatorname{Tr}(\mathbf{M}) \\
&\overset{\textcircled{2}}{=} n^2\left[\frac{p}{n} + \left(1 - \frac{h}{n}\right)^2 + o(1)\right]\operatorname{Tr}(\mathbf{M}),
\end{aligned}
$$

where $\textcircled{1}$ is due to the definitions of index sets $\mathcal{S}$ and $\mathcal{D}$ (Equation (27) and in Equation (28)), and $\textcircled{2}$ is because $|\mathcal{D}_{\text{pair}}| \leq h$. $\qquad\square$

**Lemma 14.** *For a fixed matrix* $\mathbf{M}$, *we have*

$$\mathbb{E}\operatorname{Tr}(\mathbf{\Sigma}\mathbf{M}\mathbf{\Sigma}\mathbf{M}) = (n - h + |\mathcal{D}_{\text{pair}}|)\left[\operatorname{Tr}\left(\mathbf{M}\right)\right]^2 + (n-h)^2\operatorname{Tr}\left(\mathbf{M}\mathbf{M}\right) + n\operatorname{Tr}\left(\mathbf{M}\mathbf{M}^\top\right)$$

*Proof.* Following the same strategy as in proving Lemma 13, we complete the proof by showing

$$
\begin{aligned}
\mathbb{E}\operatorname{Tr}(\mathbf{\Sigma}\mathbf{M}\mathbf{\Sigma}\mathbf{M}) &= \sum_{\ell=\pi^\natural(\ell)} \mathbb{E}\operatorname{Tr}\left(\mathbf{X}_\ell^\top \mathbf{M}\mathbf{X}_\ell\mathbf{X}_\ell^\top \mathbf{M}\mathbf{X}_\ell\right) + \sum_{\ell_1,\ell_2\in\mathcal{S},\ell_1\neq\ell_2} \mathbb{E}\operatorname{Tr}\left(\mathbf{X}_{\ell_1}\mathbf{X}_{\ell_1}^\top \mathbf{M}\mathbf{X}_{\ell_2}\mathbf{X}_{\ell_2}^\top \mathbf{M}\right) \\
&\quad + \sum_{\ell\in\mathcal{D}} \mathbb{E}\operatorname{Tr}\left(\mathbf{X}_\ell\mathbf{X}_{\pi^\natural(\ell)}^\top \mathbf{M}\mathbf{X}_\ell\mathbf{X}_{\pi^\natural(\ell)}^\top \mathbf{M}\right) + \sum_{\ell\in\mathcal{D}_{\text{pair}}} \mathbb{E}\operatorname{Tr}\left(\mathbf{X}_\ell\mathbf{X}_{\pi^\natural(\ell)}^\top \mathbf{M}\mathbf{X}_{\pi^\natural(\ell)}\mathbf{X}_\ell^\top \mathbf{M}\right) \\
&= (n - h + |\mathcal{D}_{\text{pair}}|)\left[\operatorname{Tr}\left(\mathbf{M}\right)\right]^2 + (n-h)^2\operatorname{Tr}\left(\mathbf{M}\mathbf{M}\right) + n\operatorname{Tr}\left(\mathbf{M}\mathbf{M}^\top\right).
\end{aligned}
$$

$\qquad\square$

**Lemma 15.** *For a fixed matrix* $\mathbf{M}$, *we have*

$$\mathbb{E}\left[\operatorname{Tr}(\mathbf{\Sigma}\mathbf{M})\right]^2 = (n-h)^2\left[\operatorname{Tr}(\mathbf{M})\right]^2 + n\operatorname{Tr}\left(\mathbf{M}^\top\mathbf{M}\right) + (n - h + |\mathcal{D}_{\text{pair}}|)\operatorname{Tr}(\mathbf{M}\mathbf{M}).$$

*Proof.* We complete the proof by showing

$$
\begin{aligned}
\mathbb{E}\left(\operatorname{Tr}(\mathbf{\Sigma}\mathbf{M})\right)^2 &= \sum_{\ell\in\mathcal{S}} \mathbb{E}\left(\mathbf{X}_\ell^\top \mathbf{M}\mathbf{X}_\ell\right)^2 + \sum_{\ell_1,\ell_2\in\mathcal{S},\ell_1\neq\ell_2} (\operatorname{Tr}(\mathbf{M}))^2 + \sum_{\ell\in\mathcal{D}} \underbrace{\mathbb{E}\mathbf{X}_{\pi^\natural(\ell)}^\top \mathbf{M}\mathbf{X}_\ell\mathbf{X}_\ell^\top \mathbf{M}^\top \mathbf{X}_{\pi^\natural(\ell)}}_{\operatorname{Tr}(\mathbf{M}^\top\mathbf{M})} \\
&\quad + \sum_{\ell\in\mathcal{D}_{\text{pair}}} \underbrace{\mathbb{E}\mathbf{X}_{\pi^\natural(\ell)}^\top \mathbf{M}\mathbf{X}_\ell\mathbf{X}_\ell^\top \mathbf{M}\mathbf{X}_{\pi^\natural(\ell)}}_{\operatorname{Tr}(\mathbf{M}\mathbf{M})} \\
&= (n-h)^2\left[\operatorname{Tr}(\mathbf{M})\right]^2 + n\operatorname{Tr}\left(\mathbf{M}^\top\mathbf{M}\right) + (n - h + |\mathcal{D}_{\text{pair}}|)\operatorname{Tr}(\mathbf{M}\mathbf{M}).
\end{aligned}
$$

$\qquad\square$

**Lemma 16.** *For a fixed matrix* $\mathbf{M}$*, we have*

$$\mathbb{E} \sum_{\ell=\pi^\natural(\ell)} \mathbf{X}_{\pi^\natural(i)}^\top \mathbf{M} \mathbf{X}_\ell \mathbf{X}_\ell^\top \mathbf{X}_{\pi^\natural(i)} = (n-h) \operatorname{Tr}(\mathbf{M}) + (p+1) \mathbb{1}_{i=\pi^\natural(i)} \operatorname{Tr}(\mathbf{M}).$$

*Proof.* Provided that $i = \pi^\natural(i)$, we have

$$\mathbb{E}\Xi_{1,1} = \mathbb{E}\mathbf{X}_i^\top \mathbf{M}\mathbf{X}_i\mathbf{X}_i^\top \mathbf{X}_i + \sum_{\ell \neq i, \ \ell=\pi^\natural(\ell)} \mathbb{E}\mathbf{X}_i^\top \mathbf{M}\mathbf{X}_\ell\mathbf{X}_\ell^\top \mathbf{X}_i$$

$$= (p+2) \operatorname{Tr}(\mathbf{M}) + \sum_{\ell \neq i, \ \ell=\pi^\natural(\ell)} \operatorname{Tr}(\mathbf{M}) = (n-h+p+1) \operatorname{Tr}(\mathbf{M}) \mathbb{1}_{i=\pi^\natural(i)}. \tag{74}$$

Provided that $i \neq \pi^\natural(i)$, we have

$$\mathbb{E}\Xi_{1,1} = \sum_{\ell=\pi^\natural(\ell)} \mathbb{E}\mathbf{X}_{\pi^\natural(i)}^\top \mathbf{M}\mathbf{X}_\ell\mathbf{X}_\ell^\top \mathbf{X}_{\pi^\natural(i)} = (n-h) \operatorname{Tr}(\mathbf{M}) \mathbb{1}_{i\neq\pi^\natural(i)}. \tag{75}$$

Combining (74) and (75) then completes the proof. $\qquad\square$

**Lemma 17.** *For a fixed* $\mathbf{M}$*, we have*

$$\mathbb{E}\sum_\ell \mathbf{X}_{\pi^\natural(i)}^\top \mathbf{M}\mathbf{X}_{\pi^\natural(\ell)}\mathbf{X}_\ell^\top \mathbf{X}_j = \left(p\mathbb{1}_{i=j}\mathbb{1}_{i\neq\pi^\natural(i)} + \mathbb{1}_{j=\pi^{\natural 2}(i)}\right) \operatorname{Tr}(\mathbf{M}).$$

We omit its proof as it is a direct application of Wick's theorem (Theorem 4).

**Lemma 18.** *We have*

$$\mathbb{E}\mathbb{1}(i=\pi^\natural(i)) = \frac{n-h}{n}(1+o_P(1)), \qquad \mathbb{E}\mathbb{1}_{i=j} = \frac{h}{n^2}(1+o_P(1)),$$

$$\mathbb{E}\mathbb{1}_{j=\pi^{\natural 2}(i)} = \frac{h}{n^2}(1+o_P(1)), \qquad \mathbb{E}\mathbb{1}_{i=j}\mathbb{1}_{i=\pi^{\natural 2}(i)} = \frac{|\mathcal{D}_{\text{pair}}|}{n^2}(1+o_P(1)).$$

This lemma can be easily proved by assuming the indices $i$, $j$, $\pi^\natural(i)$, and $\pi^\natural(j)$ are uniformly sampled from the set $\{1, 2, \cdots, n\}$

# E USEFUL FACTS

This section collects some useful facts for the sake of self-containing.

**Theorem 4** (Wick's theorem (Theorem 1.28 in Janson (1997))). *Considering the centered jointly normal variables* $g_1, g_2, \cdots, g_n$*, we conclude*

$$\mathbb{E}(g_1 g_2 \cdots g_n) = \sum_{\substack{\text{all possible disjoint} \\ \text{pairs } (i_k, j_k) \in \{1,2,\cdots,n\}}} \prod_k \mathbb{E}(g_{i_k} g_{j_k}).$$

With Wick's theorem, we can reduce the computation of high-order Gaussian moments to calculating the expectations of a series of low-order Gaussian moments.

**Lemma 19** (Equation (3.2) in Neudecker & Wansbeek (1987)). *For a normally distributed random matrix* $\mathbf{G} \in \mathbb{R}^{n \times p}$ *which satisfies* $\mathbb{E}\mathbf{G} = \mathbf{0}$ *and* $\mathbb{E}\operatorname{vec}(\mathbf{G})\operatorname{vec}(\mathbf{G})^\top = \mathbf{U} \otimes \mathbf{V}$*, we have*

$$\mathbb{E}\left(\mathbf{G}^\top \mathbf{A}\mathbf{G}\mathbf{C}\mathbf{G}^\top \mathbf{B}\mathbf{G}\right) = \operatorname{Tr}(\mathbf{A}\mathbf{U})\operatorname{Tr}(\mathbf{B}\mathbf{U})\mathbf{V}\mathbf{C}\mathbf{V} + \operatorname{Tr}\left(\mathbf{A}\mathbf{U}\mathbf{B}^\top \mathbf{U}\right)\mathbf{V}\mathbf{C}^\top \mathbf{V}$$

$$+ \operatorname{Tr}(\mathbf{A}\mathbf{U}\mathbf{B}\mathbf{U})\operatorname{Tr}(\mathbf{C}\mathbf{V})\mathbf{V},$$

*where* $\operatorname{vec}(\cdot)$ *is the vector operation;* $\otimes$ *is the Kronecker product (Horn & Johnson, 1990); and* $\mathbf{A}, \mathbf{B}$ *and* $\mathbf{C}$ *are arbitrary fixed matrices.*

**Lemma 20** (Stein's Lemma (cf. Section 1.3 in Talagrand (2010))). *Let* $g \sim \mathsf{N}(0,1)$*. Then for any differentiable function* $f : \mathbb{R} \mapsto \mathbb{R}$ *we have*

$$\mathbb{E}[gf(g)] = \mathbb{E}f^{'}(g),$$

*where* $\lim_{\|g\|\to\infty} f(g)e^{-a\|g\|_2^2} = 0$ *for any* $a > 0$.