# OpenReview forum: "The Phase Transition Phenomenon of Shuffled Regression"
_ICLR.cc/2026/Conference — Submitted to ICLR 2026_

### Official Review · Reviewer_D68h · 2025-11-01

**Soundness:** 2
**Presentation:** 2
**Contribution:** 2
**Rating:** 4
**Confidence:** 3

**Summary:**

This paper studies shuffled regression problem, the goal of which is to find permutation matrix and coefficient matrix. In particular, the authors study this problem using message passing algorithm and provide a framework to identify the locations of phase transition thresholds. With a deep theoretical study on this problem, the paper provide some level of insight into the shuffled regression problem.

**Strengths:**

The paper provides novel frame work to find phase transition point of shuffled regression problem. The analysis relates the recoverable fraction of shuffled rows to dimension ratios, which I think is informative. Overall, I think the paper conducts rigorous theoretical study for phase-transition point in shuffled regression.

**Weaknesses:**

As far as I understood, the Gaussian approximation is a core step to make the analysis ease, which might cause approximation error. Moreover, I am doubt that oracle cases exist in practice. Moreover, it is quite difficult to see that if the proposed non-oracle case approach is practically useful as the experimental results are very limited. I understand this work is theoretical one, but the implication and application of the proposed approach should be discussed.

**Questions:**

1. Under what explicit conditions on $X$ and noise is the Gaussian approximation provably accurate for the MP potentials? How do the predicted thresholds shift when those conditions are violated?
2. Is there any realistic experiments or application of the shuffled regression? It would be more convincing how the proposed phase-transition is used in some applications.

---

### Official Review · Reviewer_Nmx8 · 2025-11-01

**Soundness:** 2
**Presentation:** 2
**Contribution:** 3
**Rating:** 4
**Confidence:** 3

**Summary:**

This paper studies the shuffled regression problem, where the features are permuted before regression. It is empirically known that, in this problem, when a certain parameter exceeds a threshold, the estimation error rapidly converges to zero, showing a phase transition phenomenon. The authors aim to theoretically prove this phenomenon. To this end, they reformulate the problem using a probabilistic graphical model and connect the message passing technique to a branching random walk process, thereby explaining the relationship between the signal-to-noise ratio (SNR) and the phase transition point. They investigate both the oracle case, where the hidden parameter $B$ is known, and the non-oracle case, where $B$ is unknown to the learning agent.

**Strengths:**

Although this reviewer is not familiar with the shuffled regression problem itself, as someone who often studies theoretical aspects of applied linear regression problems, I found this problem setting highly interesting, and the phase transition phenomenon particularly appealing. I would like to read the paper more deeply if I had more time. If, as the authors claim, this paper provides the first result on phase transition for $m > 1$, the contribution seems quite solid. How incremental the contribution is would be better assessed by an expert in this specific field.

**Weaknesses:**

1. Equation (3) seems important, but there appear to be issues with notation.
- 1-1) What does the indicator mean? The indicator function is usually defined on a set, returning 1 if the element belongs to it and 0 otherwise. This should be clarified.
- 1-2) Since $\Pi$ is a permutation matrix, $\sum_j \Pi_{ij} = 1$, so doesn’t $\mu(\Pi) = 0$ just hold trivially?

2. Clarity issues.
- 2-1) The definition of $E$ is confusing. Is $E$ the random matrix introduced in line 77, or the fixed matrix in Eq. (2)?
- 2-2) Why do edges suddenly appear in lines 135–136?
- 2-3) What is $\Omega$? There is no explanation.
- 2-4) Eq. (19) appears for the first time without prior introduction, yet plays a crucial role. This lack of explanation is problematic.
- 2-5) There is insufficient explanation for why the permutation recovery transition point satisfies $H + H' > \Omega$, and no guidance on where to look in the Appendix for this.
- 2-6) What is $\pi(i)$ in line 206?
- 2-7) In Proposition 2, why does $\text{snr}{\text{oracle}}$ appear on both sides? Couldn’t it be written in closed form as $\text{snr}{\text{oracle}} = \dots$? Also, does Proposition 2 indeed represent the theoretical SNR?
- 2-8) The definition of threshold also needs clarification. Does “sharply drops to zero” mean that the error probability literally becomes zero after the threshold, or that it rapidly converges to zero as SNR increases?

There are too many places where explanations are missing, making it very difficult for this reviewer to understand the paper’s contribution. While domain experts may be able to follow the arguments quickly, for readers with partial background knowledge, the paper feels quite unfriendly. At minimum, the authors should provide Appendix references or guidance for readers unfamiliar with the topic. Overall, the writing feels quite rushed.

3. The paper describes the phase transition only under settings where all variables grow linearly proportionally. However, there can be other asymptotic regimes where $n$ increases at a slower rate than the number of samples, e.g., $n \propto \sqrt{m}$, while still diverging. The current results appear limited to a restrictive setting where $m, n, p$ satisfy fixed linear ratios.


4. Assumption 1 introduces an independence assumption that seems questionable. Even if the numerical results do not show issues, can such independence really be assumed freely? For instance, since the same $W_i$ contributes linearly to all $E_{ij}^{\text{oracle}}$ with the same $i$, these random variables are almost certainly dependent.

5. The non-oracle case appears much more interesting and practical than the oracle one, but Section 4 does not seem to provide any theoretical results for SNR. Only the mean and variance of $\Xi$ are presented. Does Section 4 only describe how the error probability is measured? Are there no theoretical results regarding the SNR?

**Minor Comments**

1. In the abstract (lines 17–19), “message passing” is repeated too soon. It might be better to end line 17 with “identify the locations of the phase transition points” and mention message passing later when describing the analysis.
- 1-1) Additionally, the abstract feels unusually long.

2. The paper cites Lufkin et al., 2024 as “published this February,” but it is listed as a 2024 paper. Perhaps the timing is mixed up, or this line was copied from an earlier draft.

3. In line 377, it would be clearer to add parentheses around $E\Xi$ and $\text{Var},\Xi$, e.g., write them as $E[\Xi]$ and $\text{Var}[\Xi]$.

**Questions:**

Please check the weaknesses above.

---

### Official Review · Reviewer_S647 · 2025-11-01

**Soundness:** 2
**Presentation:** 1
**Contribution:** 2
**Rating:** 2
**Confidence:** 4

**Summary:**

This paper studies the shuffled regression problem (linear regression with an unknown permutation of the outputs) and investigates the phase transition phenomenon in permutation recovery. The authors model the problem using a probabilistic graphical framework and analyze it via a message passing (MP) algorithm to determine when perfect recovery becomes possible. They derive the signal-to-noise ratio (SNR) threshold at which this transition occurs.

**Strengths:**

The paper addresses an interesting and important problem: recovering correspondences in shuffled regression. Explaining the empirically observed phase transition through the lens of message passing algorithms and graphical models is a novel approach that could offer fresh insights.

The authors derive an analytical characterization of the phase transition threshold (critical SNR) using a Gaussian approximation method.
However, since the paper was hard to parse I couldn't verify the correctness of the theoretical claims.

The paper also includes numerical experiments that appear to support the theory.

**Weaknesses:**

In the non-oracle case, the authors adopt an estimator from prior work (Zhang & Li, 2020) to sidestep the hardness of the quadratic assignment problem. This means the analysis no longer corresponds to the true maximum-likelihood estimator but to a specific algorithmic surrogate. While the authors justify this choice by noting that the estimator is statistically optimal in many regimes and exhibits similar phase transition behavior, the paper does not clarify how tight or general these results are.

The paper overall is poorly written and poorly organized, which severely limits readability and comprehension. Many explanations are terse or scattered. As a result, I found it difficult to fully understand or evaluate the contributions. Some specific issues include:
1. The application of the problem is mentioned in just a single line, with no elaboration or context. Expanding this would help readers understand its practical relevance.

2. The authors emphasize deriving the phase transition threshold in both the Abstract and Introduction, yet the definition of SNR appears much later (at the bottom of page 2).

3. The assumptions on matrix X (being random with i.i.d. Gaussian entries) are not introduced until line 190 in Section 3. This assumption is fundamental and should be stated clearly in the problem setup.

4. In Equation (3), the notation for the indicator random variable is incorrect or unclear.

5. Section 2.1 is particularly hard to read. The paper jumps into heavy technical derivations without first providing an intuitive overview of the approach or a clear explanation.

**Questions:**

Gaussian Approximation: It would be helpful to add a brief discussion (perhaps in Section 3.3 or the Conclusion) explaining why the Gaussian approximation is expected to be accurate and what intuition underlies this assumption.

---

### Official Review · Reviewer_EyNj · 2025-11-01

**Soundness:** 3
**Presentation:** 2
**Contribution:** 3
**Rating:** 4
**Confidence:** 2

**Summary:**

The authors derive an approximate algorithm that can predict the phase transition in permutated linear regression problems and is the first significant work beyond the single observation setting.

They accomplish this by extending message passing techniques, which have well-established use in the permutated linear regression literature, along with a sequence of bounds and approximations.

**Strengths:**

The problem of permuted linear regression is very important to a small community, and the techniques developed here are potentially impactful, even outside of the community.

**Weaknesses:**

My primary concern is that I think this paper belongs in a statistical journal, not ICRL. Most of the contributions are technical and the permuted regression probably may not gather enough interest at this conference.

For an ICRL audience, which will largely not able to appreciate the technical innovations, the justification for your proposed method is weak. There are no sub-optimality bounds and the empirical results lack convincing baselines. Is essence, Figure 1 does not do a good job of showing the phase transition predictions for the general case are good. Are there any baselines you can compare to? Some naive approach, perhaps? Also, the x-axis are not aligned, which makes them much more work to read. Zooming out the x-axis would also emphasize your point.

There are quite a few approximations made to make your algorithm computable, and I would encourage the authors to provide some sense of how much error results.

**Questions:**

I’m also worried that Figure 1 only covers very trivial cases. How does it perform on something closer to an application?

Can you provide concrete evidence that your approximations aren’t too damaging on problems with non-trivial structure (e.g. beyond what is included in figure 1?). I know there are many approximations you use, but an argument for each independently or for the procedure end-to-end would be sufficient.

Why did you choose ICRL as a venue?

What does the approx equal sign in (6) mean?

Without pointing me to appendix, can you provide some intuition for how the translation between your problem and a message passing algorithm works?

You mentioned that you felt your techniques could have applications for different problems. Can you give any examples?

---

### Meta-Review · Area_Chair_KmNp · 2025-12-08

**Summary:**

All reviewers have quite significant concerns, including on the clarity, writing, and correctness.

**Reviewer Concerns:**

No response was given

**Reviewer Scores:**

No response was given, hence no change in scores

---

### Decision · Program_Chairs · 2026-01-26

Reject